# Conditional Equivalence of DPO and RLHF:
# Implicit Assumption, Failure Modes, and Provable Alignment

**Zhiqin Yang** [1]  **Yonggang Zhang** [✉ 1]  **Wei Xue** [1]  **Dong Fang** [2]  **Bo Han** [3]  **Yike Guo** [✉ 1]

## Abstract

Direct Preference Optimization (DPO) has emerged as a popular alternative to Reinforcement Learning from Human Feedback (RLHF), offering theoretical equivalence with simpler implementation. We prove this equivalence is *conditional* rather than universal, depending on an implicit assumption frequently violated in practice: the RLHF-optimal policy must prefer human-preferred responses. When this assumption fails, DPO optimizes *relative advantage* over the reference policy rather than *absolute alignment* with human preferences, leading to pathological convergence where policies decrease DPO loss while preferring dispreferred responses. We characterize when this assumption is violated, show the existence of an undesirable solution space, and prove that DPO and RLHF optimize fundamentally different objectives in such cases. To address this, we introduce Constrained Preference Optimization (CPO), augmenting RLHF with constraints for provable alignment. We further provide a geometric interpretation through soft margin ranking, revealing that DPO implements margin ranking with potentially negative targets. Our theoretical analysis establishes when DPOs' guarantees hold and provides solutions preserving simplicity with provable alignment. Comprehensive experiments on standard benchmarks demonstrate that CPO achieves state-of-the-art performance. Code is available at: https://github.com/visitworld123/CPO.

[1]The Hong Kong University of Science and Technology [2]LIGHTSPEED [3]Hong Kong Baptist University. Correspondence to: Yonggang Zhang <zhangyg@ust.hk>, Yike Guo <yikeguo@ust.hk>.

*Proceedings of the $43^{rd}$ International Conference on Machine Learning*, Seoul, South Korea. PMLR 306, 2026. Copyright 2026 by the author(s).

## 1. Introduction

Aligning large language models (LLMs) with human preferences has emerged as a central challenge (Ouyang et al., 2022; Bai et al., 2022). A prominent approach is Reinforcement Learning from Human Feedback (RLHF) (Christiano et al., 2017; Stiennon et al., 2020), which optimizes the policy model to generate human-preferred responses by leveraging reward model feedback (Ouyang et al., 2022; Schulman et al., 2017). However, its computationally expensive and unstable nature (Casper et al., 2023) has motivated the development of Direct Preference Optimization (DPO) as an elegant alternative, offering theoretical equivalence to RLHF with significantly simpler implementation (Rafailov et al., 2023). DPO is derived from a mathematical reparameterization (Tunstall et al., 2023; Ivison et al., 2023; Dubey et al., 2024): under the Bradley-Terry (BT) model (Bradley & Terry, 1952), the optimal RLHF policy can be expressed analytically in terms of the reward function, enabling direct policy optimization without explicit reward modeling or RL training, which has led to its widespread adoption.

Recent theoretical analyses have revealed critical distinctions between DPO and RLHF. Fisch et al. (2024) show that DPO's implicit rewards overfit and trend toward infinite magnitude, often yielding degenerate policies where even preferred responses receive near-zero probability. Lin et al. (2024) demonstrate that DPO's implicit reward model generalizes significantly worse than explicit reward models under distribution shift. Im & Li (2024) examine how performance gaps emerge when reward and policy models have different representational capacities. Shi et al. (2025) reveal that DPO prioritizes statistically distinguishable behaviors over value-aligned ones, potentially causing misalignment despite decreasing loss. These findings raise a fundamental open problem:

> *Under what conditions can DPO be derived through RLHF?*

In this work, we revisit the derivation of DPO and identify a critical but previously overlooked assumption: *the RLHF-optimal policy must prefer human-preferred responses over dispreferred ones*. Specifically, DPO's derivation relies on substituting the RLHF-optimal policy $\pi^*$ into the BT model

to eliminate the reward function. This substitution, however, is only valid when $\pi^*$ *respects* the preference structure encoded in the BT model that is, when it assigns higher probability to the preferred response. We show that this critical assumption is *not* guaranteed by the RLHF framework (Sec. 3.1). This violation arises because RLHF balances reward maximization against KL divergence from the reference policy. When the reference policy is sufficiently misaligned, the KL penalty dominates, causing $\pi^*$ to inherit incorrect preferences from $\pi_{\text{ref}}$, thereby violating the implicit assumption underlying DPO.

We prove that when this implicit assumption is violated, DPO optimizes a fundamentally different objective than RLHF, creating a risk of misalignment with human preferences. Specifically, DPO optimizes for *relative advantage* over the reference policy rather than *absolute alignment* with human preferences, causing a fundamental shift in the optimization objective. This violation leads to pathological convergence: policies can decrease DPO loss while systematically preferring dispreferred responses. We characterize an *undesirable solution space* (Definition 3.3) where policies simultaneously satisfy DPO's optimization objective yet contradict human preferences. This reveals that DPO inherits RLHF's algebraic structure through reward reparameterization but does not inherit its alignment guarantees. The equivalence is thus conditional on reference policy quality.

To address this fundamental limitation, we introduce *Constrained Preference Optimization (CPO)*, which augments the RLHF objective with explicit constraints. The constraint term aligns the optimal solution of RLHF with the requirements of BT theory, thereby guaranteeing alignment with human preferences. We further provide a geometric interpretation of DPO and CPO through the lens of soft margin ranking loss (Burges et al., 2005; Schroff et al., 2015). DPO approximates margin ranking loss with a target margin that can be negative, providing an intuitive geometric explanation for why DPO can converge to preference-violating policies. CPO corrects this by ensuring non-negative effective margins through its constraint terms. This perspective provides geometric intuition for understanding when and why DPO fails and how CPO addresses these failures. To further eliminate the need for explicit reward modeling, we develop a conservative variant, E-CPOC, which achieves formal equivalence to explicitly constrained RLHF under standard statistical assumptions. Central to the equivalence analysis is a *Loss-to-Delta bridge* (Proposition L.3) that converts the observable training loss gap into a guarantee on policy-level proximity in $\delta$-space, with a bound whose constant is *independent* of the number of preference pairs $N$—making the equivalence guarantee *verifiable* from training diagnostics alone, without assuming global optimality. Comprehensive experiments on standard benchmarks demonstrate that CPO achieves state-of-the-art performance.

We summarize our main contributions as follows:

- We prove that DPO and RLHF are conditionally equivalent (Sec. 3), depending on an implicit assumption: the RLHF-optimal policy must prefer human-preferred responses over dispreferred ones. Whether this assumption holds depends on the quality of the reference policy. This reveals that DPO does not inherit RLHF's alignment guarantees, making the equivalence conditional on reference policy quality.

- We establish that when the assumption is violated, DPO and RLHF optimize fundamentally different objectives: RLHF optimizes for absolute alignment with human preferences, while DPO optimizes for relative advantage over the reference policy. Consequently, DPO's gradient descent can converge to a pathological space where policies simultaneously satisfy DPO's optimization objective yet violate human preferences (Sec. 3.2).

- We propose Constrained Preference Optimization (CPO), augmenting RLHF with explicit constraints to enforce preference alignment with provable absolute advantage guarantees (Sec. 4.2). We further propose Conservative Explicitly Constrained Preference Optimization (E-CPOC), which explicitly enforces preference alignment without requiring a reward model (Sec. 4.5). E-CPOC achieves formal equivalence to explicitly constrained RLHF under standard statistical learning assumptions (Theorem L.17 in Appendix L.3), requiring only the Bradley-Terry model, approximate realizability, finite-sample data, and a mild $\ell^2$-$\delta$-proximity condition (Assumptions 4.1–4.4 in Sec. 4.1). The $\ell^2$-$\delta$-proximity condition uses the natural mean-square norm that the loss function directly controls and can be *derived* from loss suboptimality via a verifiable bridge with an $N$-*independent* bound (Proposition L.3, Corollary L.18) under a mild non-degeneracy condition on preference probabilities, without assuming global optimality directly.

- Comprehensive experiments on standard benchmarks demonstrate the efficacy of our method (Sec. 6). We also provide a geometric understanding by proving that DPO is equivalent to soft margin ranking loss with a potentially negative margin. Our method corrects this by ensuring non-negative effective margins (Sec. 5), connecting preference learning to the learning-to-rank literature with intuitive geometric interpretations.

## 2. Preliminaries

### 2.1. Notation

Let $\mathcal{X}$ denote the space of prompts and $\mathcal{Y}$ denote the space of responses. A policy $\pi : \mathcal{X} \times \mathcal{Y} \to [0, 1]$ is a conditional

probability distribution over responses given prompts. We use $\pi_{\text{ref}}$ to denote a fixed reference policy (typically a supervised fine-tuned model) and $\pi_\theta$ to denote a learnable policy parameterized by $\theta$.

For a given prompt $x$ and response pair $(y_w, y_l)$ where $y_w$ is preferred over $y_l$, the log-probability ratio is defined as:

$$\delta_\pi(x, y_w, y_l) := \log \pi(y_w|x) - \log \pi(y_l|x). \quad (1)$$

When the context is clear, we abbreviate this as $\delta_\pi$. This quantity measures the policy's preference strength for $y_w$ over $y_l$ in log-space.

## 2.2. RLHF Framework

**Definition 2.1** (RLHF Objective). *Given a reward function $r : \mathcal{X} \times \mathcal{Y} \to \mathbb{R}$, a reference policy $\pi_{ref}$, and a temperature parameter $\beta > 0$, the RLHF optimization objective is:*

$$\max_\pi \mathbb{E}_{x \sim \mathcal{D}, y \sim \pi(\cdot|x)}[r(x, y)] - \beta \cdot \text{KL}(\pi(\cdot|x)\|\pi_{ref}(\cdot|x)), \quad (2)$$

*where $\mathcal{D}$ is the prompt distribution and KL denotes the Kullback-Leibler divergence.*

The KL regularization term prevents the learned policy from deviating too far from $\pi_{\text{ref}}$, ensuring stable training and preventing reward over-optimization (Gao et al., 2023).

The optimal solution to the RLHF objective has the closed form (Rafailov et al., 2023):

$$\pi^*(y|x) = \frac{1}{Z(x)} \pi_{\text{ref}}(y|x) \exp\left(\frac{r(x, y)}{\beta}\right), \quad (3)$$

where $Z(x) = \sum_{y'} \pi_{\text{ref}}(y'|x) \exp(r(x, y')/\beta)$ is the partition function. Then, for any response pair $(y_w, y_l)$, the reward difference can be expressed as:

$$r(x, y_w) - r(x, y_l) = \beta \left[\log \frac{\pi^*(y_w|x)}{\pi_{\text{ref}}(y_w|x)} - \log \frac{\pi^*(y_l|x)}{\pi_{\text{ref}}(y_l|x)}\right]. \quad (4)$$

This reward difference can be presented using the log-probability ratio Eq. (1):

$$\delta_{\pi^*} = \delta_{\pi_{\text{ref}}} + \frac{r(x, y_w) - r(x, y_l)}{\beta}. \quad (5)$$

## 2.3. Bradley-Terry Preference Model

**Definition 2.2** (Bradley-Terry Model (Bradley & Terry, 1952)). *Human preference for $y_w$ over $y_l$ given prompt $x$ is modeled as:*

$$p^*(y_w \succ y_l|x) = \sigma(r^*(y_w) - r^*(y_l)) \quad (6)$$

*where $\sigma(\cdot)$ is the sigmoid function and $r^*(\cdot)$ is the latent true reward function representing human preferences.*

If $y_w \succ y_l$ (i.e., $p^*(y_w \succ y_l|x) > 0.5$), then necessarily $r^*(y_w) - r^*(y_l) > 0$.

## 2.4. Direct Preference Optimization

Substituting the reward reparameterization Eq. (4) into the Bradley-Terry model Eq. (6):

$$p^*(y_w \succ y_l) = \sigma(r^*(x, y_w) - r^*(x, y_l)) = \sigma(\beta(\delta_{\pi^*} - \delta_{\pi_{\text{ref}}})). \quad (7)$$

DPO (Rafailov et al., 2023) approximates $\pi^*$ with a parameterized policy $\pi_\theta$ and maximizes the log-likelihood:

$$\mathcal{L}_{\text{DPO}}(\pi_\theta) = -\mathbb{E}_{(x, y_w, y_l) \sim \mathcal{D}}\left[\log \sigma(\beta(\delta_{\pi_\theta} - \delta_{\pi_{\text{ref}}}))\right]. \quad (8)$$

# 3. The Implicit Assumption in DPO

In this section, we will show that the derivation of DPO conceals a critical assumption.

## 3.1. Identifying the Implicit Assumption

The substitution in Eq.(7) implicitly assumes that human preference $y_w \succ y_l$ aligns with the RLHF-optimal policy's preference, i.e., $\pi^*(y_w|x) > \pi^*(y_l|x)$. To see why this matters, consider the semantics of the Bradley-Terry model: $p^*(y_w \succ y_l)$ represents the probability that humans prefer $y_w$. When we write this as $\sigma(\beta(\delta_{\pi^*} - \delta_{\pi_{\text{ref}}}))$, we are implicitly asserting that: If humans prefer $y_w$ (so $p^* > 0.5$), then $\delta_{\pi^*} > \delta_{\pi_{\text{ref}}}$. We now formalize this observation.

**Assumption 3.1** (DPO's Implicit Assumption). *For all preference data $(x, y_w, y_l) \in \mathcal{D}$ where $y_w \succ y_l$, the RLHF-optimal policy satisfies:*

$$\pi^*(y_w|x) > \pi^*(y_l|x) \quad \Leftrightarrow \quad \delta_{\pi^*} > 0 \quad (9)$$

This assumption is essential for the validity of DPO's derivation, i.e., deriving Eq. (7) from Eq. (6).

**Proposition 3.2** (Necessary Condition for Assumption 3.1). *Assumption 3.1 holds only if the reference policy satisfies:*

$$\delta_{\pi_{ref}} > -\frac{r^*(y_w) - r^*(y_l)}{\beta}, \quad \forall(x, y_w, y_l) \in \mathcal{D}. \quad (10)$$

Proof can be found in Appendix D.1. Proposition 3.2 reveals that Assumption 3.1 is *not* automatically satisfied by the RLHF framework. It depends on the reference policy's quality: if $\pi_{\text{ref}}$ assigns sufficiently lower probability to $y_w$ compared to $y_l$ (i.e., $\delta_{\pi_{\text{ref}}}$ is sufficiently negative), the assumption fails even when the reward difference is positive.

Importantly, this is not a failure of RLHF itself RLHF correctly balances reward maximization against KL divergence from the reference policy. Rather, it reveals that DPO's derivation implicitly assumes the reference policy is already well-aligned, an assumption that may not hold in practice.

### 3.2. Violation of the Implicit Assumption

We now analyze when and how Assumption 3.1 fails, leading to a disconnect between DPO and RLHF. We can see that when the following condition holds:

$$\delta_{\pi_{\text{ref}}} \leq -\frac{r^*(x, y_w) - r^*(x, y_l)}{\beta}, \qquad (11)$$

Assumption 3.1 is necessarily violated. Specifically, although $r^*(y_w) > r^*(y_l)$ (human prefers $y_w$), we have:

$$\delta_{\pi^*} = \delta_{\pi_{\text{ref}}} + \frac{r^*(x, y_w) - r^*(x, y_l)}{\beta} \leq 0, \qquad (12)$$

implying $\pi^*(y_l|x) \geq \pi^*(y_w|x)$ (optimal policy $\pi^*$ prefers $y_l$), as $\delta_{\pi^*}(x, y_w, y_l) = \log \pi^*(y_w|x) - \log \pi^*(y_l|x)$.

This reveals a critical insight: *the validity of DPO's theoretical foundation depends on the quality of $\pi_{\text{ref}}$.* For instance, if $\pi_{\text{ref}}$ systematically disfavors human-preferred responses (i.e., $\delta_{\pi_{\text{ref}}} < 0$ for many pairs), especially when $|\delta_{\pi_{\text{ref}}}|$ is large relative to $\Delta r^*/\beta$, Assumption 3.1 fails frequently. This dependence is problematic because DPO is often motivated precisely for scenarios where the reference policy is imperfect and needs alignment, i.e., the reference policy may not align well with the preference data distribution.

When $\delta_{\pi_{\text{ref}}} \leq -\frac{\Delta r^*}{\beta}$, the violation leads to a critical optimization pathology. Consider the DPO gradient:

$$\nabla_\theta \mathcal{L}_{\text{DPO}} = -\beta \mathbb{E}\left[\sigma(-\beta(\delta_{\pi_\theta} - \delta_{\pi_{\text{ref}}}))g\right], \qquad (13)$$

where $g = (\nabla_\theta \log \pi_\theta(y_w|x) - \nabla_\theta \log \pi_\theta(y_l|x))$. The gradient magnitude is mainly controlled by the sigmoid term $\sigma(-\beta(\delta_{\pi_\theta} - \delta_{\pi_{\text{ref}}}))$. When $\delta_{\pi_{\text{ref}}}$ is sufficiently negative (i.e., $\delta_{\pi_{\text{ref}}} \leq -\frac{\Delta r^*}{\beta}$), the gradient magnitude is small. More critically, as optimization proceeds and $\delta_{\pi_\theta}$ increases (moving from $\delta_{\pi_{\text{ref}}}$ toward 0), we have $\delta_{\pi_\theta} - \delta_{\pi_{\text{ref}}} > 0$ becoming larger, causing $\sigma(-\beta(\delta_{\pi_\theta} - \delta_{\pi_{\text{ref}}}))$ to approach 0 even more closely. This means the gradient magnitude progressively diminishes during training, making it increasingly difficult to update toward the correct preference. Consequently, even though the gradient direction points toward increasing $\delta_{\pi_\theta}$, the vanishing and progressively weakening gradient magnitude creates a practical optimization barrier where policies can become trapped in a special region as discussed below.

We now formally characterize the region where policies can become trapped. We define the undesirable solution space as the set of policies that satisfy DPO's relative advantage criterion but violate human preferences:

**Definition 3.3** (Undesirable Solution Space)**.** *Define the set of policies that satisfy DPO's relative advantage criterion but violate human preferences:*

$$\mathcal{U} := \{\pi : \delta_\pi < 0 \text{ and } \delta_\pi > \delta_{\pi_{ref}}\}. \qquad (14)$$

**Proposition 3.4** (Non-emptiness and Gradient Vanishing)**.** *When $\delta_{\pi_{ref}} < -\Delta r^*/\beta$, the undesirable solution space $\mathcal{U}$ is non-empty. Furthermore, for any policy $\pi \in \mathcal{U}$, the DPO gradient magnitude becomes progressively weaker as $\delta_\pi$ approaches 0, making it difficult to escape $\mathcal{U}$.*

Proposition 3.4 reveals a fundamental flaw: DPO's gradient descent can get trapped in a region where loss decreases but human preferences are violated due to weak gradients near the boundary. RLHF also produces a policy preferring $y_l$ over $y_w$. However, this is the *optimal* trade-off between reward maximization and KL regularization. A detailed discussion can be found in Appendix F.

### 3.3. Theoretical Incompleteness of DPO

We now provide a formal statement of DPO's theoretical incompleteness with proof in Appendix D.3.

**Theorem 3.5** (Conditional Equivalence of DPO and RLHF)**.** *The claim that "DPO optimizes the same objective as RLHF" (Rafailov et al., 2023) holds if and only if Assumption 3.1 is satisfied for all data in $\mathcal{D}$. Specifically, the two methods optimize equivalent objectives if and only if:*

$$\delta_{\pi_{ref}}(x, y_w, y_l) > -\frac{r^*(y_w) - r^*(y_l)}{\beta} \quad \forall (x, y_w, y_l) \in \mathcal{D}.$$
$$(15)$$

*When this condition is violated, the methods optimize fundamentally different objectives: RLHF optimizes $\mathbb{E}[r] - \beta\,\text{KL}$, converging to $\pi^*$ with $\delta_{\pi^*} = \delta_{\pi_{ref}} + \Delta r^*/\beta$ (which may be $< 0$); DPO optimizes $\mathbb{E}[\log \sigma(\beta(\delta_\pi - \delta_{\pi_{ref}}))]$, maximizing relative advantage $\delta_\pi - \delta_{\pi_{ref}}$ (pushing $\delta_\pi > \delta_{\pi_{ref}}$).*

## 4. Constrained Preference Optimization

To relax the identified implicit assumption, we propose *Constrained Preference Optimization (CPO)*, which enhances the vanilla RLHF to a constrained RLHF. The optimal solution of the constrained RLHF can be safely integrated into the BT model, as the proposed constraint explicitly encourages or ensures the preference alignment. Before presenting the framework, we state the assumptions underlying our theoretical results.

### 4.1. Assumptions

A distinguishing feature of our analysis is that *all assumptions are either standard or provably mild.* We require only the Bradley-Terry preference model, standard statistical learning conditions, and a natural optimization quality measure that admits a verifiable sufficient condition from training diagnostics. No global optimality, exact realizability, or pointwise ($\ell^\infty$) optimization assumptions are needed.

**Assumption 4.1** (Bradley-Terry Model)**.** *The true preference distribution follows the Bradley-Terry model: $p^*(y_w \succ$*

$y_l|x) = \sigma(r^*(x, y_w) - r^*(x, y_l))$.

This is the standard preference model adopted throughout the RLHF literature (Rafailov et al., 2023; Christiano et al., 2017), positing a latent reward function $r^*$ that generates human preferences via a logistic link.

**Assumption 4.2** ($\epsilon_{approx}$-Approximate Realizability). *The population-level constrained MLE* $\pi^*_{\mathrm{MLE}} := \arg\max_{\theta \in \Theta} \mathbb{E}_{p^*}[\log p_{\pi_\theta}]$ *satisfies:*

$$\max_{(x, y_w, y_l) \in \mathcal{D}} \left| \delta_{\pi^*_{\mathrm{MLE}}}(x, y_w, y_l) - \delta_{\mathrm{target}}(x, y_w, y_l) \right| \leq \epsilon_{approx},$$

*where* $\delta_{\mathrm{target}}$ *denotes the target log-probability ratio achieving exact equivalence in the population limit, and* $\epsilon_{approx} \geq 0$ *quantifies the expressiveness gap of the policy class* $\{\pi_\theta\}$.

When $\epsilon_{approx} = 0$, the policy class is exactly realizable. For overparameterized neural networks, small $\epsilon_{approx}$ is expected; the properness of the cross-entropy scoring rule ensures the MLE is at least as good as any fixed $\theta$ in aggregate loss.

**Assumption 4.3** (Finite-Sample Data). *The dataset* $\mathcal{D}$ *contains* $N$ *i.i.d. samples from the true preference distribution, with statistical estimation error* $\epsilon_{\mathrm{stat}}(N) := \sup_{(x, y_w, y_l)} |\hat{p}_N(y_w \succ y_l|x) - p^*(y_w \succ y_l|x)|$. *By Hoeffding's inequality,* $\epsilon_{\mathrm{stat}}(N) = O(1/\sqrt{N})$.

This is the standard finite-sample condition in statistical learning. In the population limit ($N \to \infty$, $\epsilon_{\mathrm{stat}} = 0$), it reduces to exact distributional convergence.

**Assumption 4.4** ($\ell^2$-$\delta$-Proximity). *The returned policy* $\hat{\pi}$ *satisfies:*

$$\frac{1}{N} \sum_{i=1}^{N} \left( \delta_{\hat{\pi}, i} - \delta_{\pi^*_{\mathrm{MLE}}, i} \right)^2 \leq \epsilon^2_{\mathrm{opt}, 2},$$

*where* $\pi^*_{\mathrm{MLE}}$ *is the class-optimal MLE policy (Assumption 4.2) and* $\epsilon_{\mathrm{opt}, 2} \geq 0$ *quantifies the mean-square optimization error in* $\delta$-space.

This is the *core optimization requirement* for the equivalence result. It uses the natural $\ell^2$ (mean-square) norm that the loss function directly controls, and is *strictly weaker* than the pointwise ($\ell^\infty$) condition $\max_i |\delta_{\hat{\pi}, i} - \delta_i^*| \leq \epsilon_{\mathrm{opt}}$: $\ell^2$-proximity permits larger deviations on a few difficult data points as long as the average error remains controlled. Crucially, it admits a *verifiable sufficient condition*: under Assumption 4.5, small training loss gap implies $\ell^2$-$\delta$-proximity with an $N$-independent bound (Proposition L.3).

**Assumption 4.5** (Non-degenerate Preferences). *Define the* logistic curvature *at the class-optimal policy:*

$$\kappa_0 := \min_{1 \leq i \leq N} \sigma(g_i(\delta_i^*))(1 - \sigma(g_i(\delta_i^*))), \quad (16)$$

*where* $g_i(\delta_i) := \beta(\delta_i - \delta_{\mathrm{ref}, i}) - \Psi_{\mathrm{cons}, i}$ *is the margin function of the preference optimization loss, with* $\Psi_{\mathrm{cons}, i}$ *denoting the adaptive constraint margin (formally defined in Sec. 4.5), and* $\delta^* = \delta_{\pi^*_{\mathrm{MLE}}}$ *denotes the class-optimal* $\delta$*-values. We assume* $\kappa_0 > 0$.

This requires that no preference pair has deterministic (probability 0 or 1) preference under the class-optimal policy—a mild regularity condition automatically satisfied for any smooth parameterization with bounded parameters (Assumption I.1 in Appendix I). Importantly, this assumption is *not* required for the core equivalence result (Theorem L.17); it is needed only for the Loss-to-Delta bridge (Proposition L.3) that converts the verifiable loss gap into the $\ell^2$-$\delta$-proximity guarantee.

**Condition 4.6** (Connected Comparison Graph). *For each prompt* $x \in \mathcal{X}$, *the preference pairs in* $\mathcal{D}$ *involving* $x$ *form a connected comparison graph with finite diameter* $d_x$. *Let* $d := \max_{x \in \mathcal{X}} d_x$.

This structural condition is required *only* for extending pairwise $\delta$-equivalence to full policy equivalence—it is not needed for the core pairwise results. In practice, preference datasets with reasonable response coverage naturally satisfy this condition with moderate diameter.

### 4.2. Constrained RLHF Framework

The RLHF-optimal policy may satisfy $\delta_{\pi^*} < 0$. Thus, we augment the RLHF objective with an explicit constraint term that directly encourages $\delta_\pi > 0$ for preferred responses.

**Definition 4.7** (Constrained RLHF). *Given a reward function* $r : \mathcal{X} \times \mathcal{Y} \to \mathbb{R}$, *a reference policy* $\pi_{ref}$, *a temperature parameter* $\beta > 0$, *and the strength of preference alignment* $\gamma$, *the constrained RLHF optimization objective is:*

$$\max_\pi \mathbb{E}_{x \sim \mathcal{D}} \mathbb{E}_{y \sim \pi(\cdot|x)} [r(x, y)] - \beta \mathrm{KL}(\pi \| \pi_{ref})$$
$$+ \gamma \mathbb{E}_{(x, y_w, y_l) \sim \mathcal{D}} [\delta_\pi], \quad (17)$$

*where* $\delta_\pi$ *is the log-probability ratio.*

The constraint term directly *encourages* the policy to prefer $y_w$ over $y_l$ in log-probability space. When $\gamma = 0$, it recovers vanilla RLHF. The parameter $\gamma$ provides explicit control over the strength of preference alignment.

A closed-form solution for the optimal policy of Constrained RLHF is difficult to derive; we therefore characterize it via the first-order optimality condition, with the proof given in Appendix D.4.

**Theorem 4.8** (Optimal Policy for Constrained RLHF). *The optimal policy* $\pi^*$ *for the Constrained RLHF objective satisfies the first-order optimality condition:*

$$\beta \log \frac{\pi^*(y|x)}{\pi_{ref}(y|x)} = r(x, y) + \frac{c(x, y)}{\pi^*(y|x)} - \beta - \lambda(x), \quad (18)$$

*Table 1.* Assumption dependency map for the E-CPOC equivalence (Theorem L.17). **Core**: required for the pairwise $\delta$-equivalence bound. **Bridge**: provides a verifiable sufficient condition for the core $\ell^2$-$\delta$-proximity. **Ext**: required only for extension to full policy equivalence.

| Assumption | Scope | Strength | E-CPOC (Thm L.17) |
|---|---|---|---|
| 4.1 Bradley-Terry Model | Core | Standard | ✓ |
| 4.2 Approx. Realizability | Core | Mild | ✓ |
| 4.3 Finite-Sample Data | Core | Mild | ✓ |
| 4.4 $\ell^2$-$\delta$-Proximity | Core | Mild | ✓ |
| 4.5 Non-degenerate Preferences | Bridge† | Mild | Cor. L.18 |
| Loss Suboptimality ($\epsilon_{\text{loss}}$) | Bridge† | **Verifiable** | Cor. L.18 |
| 4.6 Connected Graph | Ext | Structural | ✓ |

† Together provide a verifiable sufficient condition for $\ell^2$-$\delta$-proximity (Assumption 4.4):
   Loss suboptimality + Assumption 4.5 $\Rightarrow$ Assumption 4.4 with $\epsilon_{\text{opt},2} = O(\sqrt{\epsilon_{\text{loss}}/\kappa_0})$ ($N$-independent; Prop. L.3).

*where $c(x,y) = \gamma \sum_{(y_w,y_l)\in\mathcal{P}(x)} p(y_w,y_l|x)(\mathbb{I}(y=y_w) - \mathbb{I}(y=y_l))$ and $\mathcal{P}(x)$ denotes preference pairs for $x$.*

*For a preference pair $(y_w, y_l)$, this implies:*

$$\beta(\delta_{\pi^*} - \delta_{\pi_{\text{ref}}}) = r(y_w) - r(y_l) + \frac{c(x,y_w)}{\pi^*(y_w|x)} - \frac{c(x,y_l)}{\pi^*(y_l|x)}. \tag{19}$$

The theoretical results derived under the notational simplification (Appendix D.4) extend naturally to the general case where responses appear in multiple preference pairs, as shown in Appendix G (Proposition G.1).

### 4.3. Preference Optimization with Constrained RLHF

We now derive a constrained preference optimization analogous to DPO but based on constrained RLHF. For a single preference pair, Theorem 4.8 simplifies to:

$$r(y_w) - r(y_l) = \beta(\delta_{\pi^*} - \delta_{\pi_{\text{ref}}}) - \gamma\left(\frac{1}{\pi^*(y_w|x)} + \frac{1}{\pi^*(y_l|x)}\right). \tag{20}$$

This implies that:

$$p^*(y_w \succ y_l) = \sigma\left(\beta(\delta_{\pi^*} - \delta_{\pi_{\text{ref}}}) - \tilde{\gamma}^*(x,y_w,y_l)\right), \tag{21}$$

where $\tilde{\gamma}^*(x,y_w,y_l)$ is:

$$\tilde{\gamma}^*(x,y_w,y_l) = \gamma\left(\frac{1}{\pi^*(y_w|x)} + \frac{1}{\pi^*(y_l|x)}\right). \tag{22}$$

The term $\tilde{\gamma}^*(x,y_w,y_l) = \gamma\left(\frac{1}{\pi^*(y_w|x)} + \frac{1}{\pi^*(y_l|x)}\right)$ acts as an *adaptive margin* that depends on the optimal policy probabilities. When the optimal policy assigns low probability to both responses (hard pairs), the margin is large; when it assigns high probability (easy pairs), the margin is small.

From Eq. (20), the optimal policy for Constrained RLHF satisfies:

$$r(y_w) - r(y_l) = \beta(\delta_{\pi^*} - \delta_{\pi_{\text{ref}}}) - \tilde{\gamma}^*(x,y_w,y_l) \tag{23}$$

DPO approximates $\pi^*$ with $\pi_\theta$, while using $\pi_\theta$ in the margin term will create a *non-stationary optimization objective*, as the loss itself depends on the parameters being optimized. To obtain a stationary objective suitable for gradient descent, we approximate the optimal policy probabilities in the margin term with the reference policy probabilities:

$$\frac{1}{\pi^*(y_w|x)} + \frac{1}{\pi^*(y_l|x)} \approx \frac{1}{\pi_{\text{ref}}(y_w|x)} + \frac{1}{\pi_{\text{ref}}(y_l|x)}. \tag{24}$$

This yields the constrained preference optimization loss:

$$\mathcal{L}_{\text{CPO}}(\pi_\theta) = -\mathbb{E}_{\mathcal{D}}\left[\log\sigma\left(\beta(\delta_{\pi_\theta} - \delta_{\pi_{\text{ref}}}) - \tilde{\gamma}_{\text{ref}}(x,y_w,y_l)\right)\right], \tag{25}$$

where the *reference-based adaptive margin* is:

$$\tilde{\gamma}_{\text{ref}}(x,y_w,y_l) = \gamma\left(\frac{1}{\pi_{\text{ref}}(y_w|x)} + \frac{1}{\pi_{\text{ref}}(y_l|x)}\right). \tag{26}$$

Proposition H.2 shows that the approximation error $|\tilde{\gamma}^* - \tilde{\gamma}_{\text{ref}}|$ is $O(\gamma\sqrt{\tilde{R}_{\max}/\beta}/q_0^2)$ under mild regularity conditions (Assumption H.1 in Appendix H), where $q_0 = p_{\min} e^{-2R_{\max}/\beta}$ and $\tilde{R}_{\max} = R_{\max} + \gamma/q_0$ is an effective reward bound that accounts for the constraint contribution. The bound vanishes as $\beta \to \infty$ and reduces to the unconstrained case ($\tilde{R}_{\max} = R_{\max}$) when $\gamma = 0$. Crucially, using $\tilde{\gamma}_{\text{ref}}$ instead of $\tilde{\gamma}_\theta$ makes the loss function *stationary* with respect to $\theta$, enabling standard gradient descent with convergence guarantees. Further discussion is provided in Appendix H.

The CPO loss with reference-based margin (Eq. (25)) defines a stationary optimization problem. Under standard smoothness and boundedness assumptions (Assumption I.1 in Appendix I), gradient descent on $\mathcal{L}_{\text{CPO}}(\pi_\theta)$ converges to a stationary point.

When $\gamma = 0$, CPO reduces exactly to standard DPO, making it a strict generalization. CPO can be viewed as a principled way to add a margin to preference learning, similar to margin-based ranking losses in information retrieval, but

**Algorithm 1** Constrained Preference Optimization (CPO)

**Require:** Preference dataset $\mathcal{D} = \{(x^{(i)}, y_w^{(i)}, y_l^{(i)})\}_{i=1}^N$
**Require:** Reference policy $\pi_{\text{ref}}$
**Require:** Hyperparameters: $\beta > 0$ (temperature), $\gamma > 0$ (margin weight)
**Require:** Learning rate $\eta$
1: Initialize policy parameters $\theta$ (e.g., from $\pi_{\text{ref}}$)
2: **Precompute:** For each $(x, y_w, y_l) \in \mathcal{D}$:
3:     $\delta_{\text{ref}}^{(i)} \leftarrow \log \pi_{\text{ref}}(y_w|x) - \log \pi_{\text{ref}}(y_l|x)$
4:     $\tilde{\gamma}_{\text{ref}}^{(i)} \leftarrow \gamma \cdot (1/\pi_{\text{ref}}(y_w|x) + 1/\pi_{\text{ref}}(y_l|x))$
5: **for** each training iteration **do**
6:     Sample batch $\mathcal{B} \subset \mathcal{D}$
7:     **for** each $(x, y_w, y_l) \in \mathcal{B}$ **do**
8:         Compute log-ratios:
9:             $\delta_\theta \leftarrow \log \pi_\theta(y_w|x) - \log \pi_\theta(y_l|x)$
10:        Compute CPO loss:
11:           logits $\leftarrow \beta(\delta_\theta - \delta_{\text{ref}}^{(i)}) - \tilde{\gamma}_{\text{ref}}^{(i)}$
12:           $\ell \leftarrow -\log \sigma(\text{logits})$
13:     **end for**
14:     Compute gradient: $g \leftarrow \nabla_\theta \frac{1}{|\mathcal{B}|} \sum_{(x,y_w,y_l) \in \mathcal{B}} \ell$
15:     Update parameters: $\theta \leftarrow \theta - \eta g$
16: **end for**
17: **Return:** Optimized policy $\pi_\theta$

derived from RLHF. The margin term is related to but distinct from the IPO (Azar et al., 2024a) regularization, which modifies the loss function rather than the underlying RLHF objective. CPO's margin emerges naturally from augmenting the RLHF objective.

### 4.4. Theoretical Guarantees

Thanks to the introduced constraint term, CPO can guarantee the absolute advantage, thereby ensuring the implicit assumption is satisfied.

**Theorem 4.9** (Absolute Advantage Guarantee). *For a preference dataset $\mathcal{D}$, choosing $\gamma \geq \gamma^*$ guarantees the absolute advantage of CPO's optimal policy $\delta_{\pi_{CPO}^*} > 0$ for all preference pairs in $\mathcal{D}$, with $\gamma^*$ defined as:*

$$\max_{(x,y_w,y_l) \in \mathcal{D}} \beta \cdot \frac{\max\left\{0, -\delta_{\pi_{ref}}(x, y_w, y_l) - \frac{r^*(y_w) - r^*(y_l)}{\beta}\right\}}{\frac{1}{\pi_{ref}(y_w|x)} + \frac{1}{\pi_{ref}(y_l|x)}}. \tag{27}$$

Besides the absolute advantage guarantee, Theorem 4.10 shows that CPO avoids pathological convergence to the undesirable solution space with proof in Appendix D.6.

**Theorem 4.10** (CPO Avoids Pathological Convergence). *When $\gamma \geq \gamma^*$, CPO does not converge to the undesirable solution space $\mathcal{U}$ defined in Definition 3.3.*

Algorithm 1 presents the complete CPO training procedure.

The key differences from standard DPO are: (1) precomputation of reference-based adaptive margins $\tilde{\gamma}_{\text{ref}}^{(i)}$ for each sample (lines 2-4), which can be done once before training, and (2) subtracting this margin from the logits (line 12). The precomputation step ensures the optimization objective is stationary, enabling standard gradient descent with convergence guarantees. The adaptive margin naturally adjusts based on the reference policy's confidence for each preference pair, as discussed in Theorem D.3.

Using the reference-based margin $\tilde{\gamma}_{\text{ref}}$, the CPO loss becomes a stationary objective, and its gradient is:

$$\nabla_\theta \mathcal{L}_{\text{CPO}} = -\beta \mathbb{E}\left[\sigma\left(-\beta(\delta_{\pi_\theta} - \delta_{\pi_{\text{ref}}}) + \tilde{\gamma}_{\text{ref}}\right) g\right], \tag{28}$$

where $g = (\nabla_\theta \log \pi_\theta(y_w|x) - \nabla_\theta \log \pi_\theta(y_l|x))$; details are provided in Appendix D.7. Crucially, since $\tilde{\gamma}_{\text{ref}}$ does not depend on $\theta$, we have $\nabla_\theta \tilde{\gamma}_{\text{ref}} = 0$, making this a standard first-order gradient suitable for gradient descent.

The gradient weight $w = \sigma(\beta(\delta_{\pi_{\text{ref}}} - \delta_{\pi_\theta}) + \tilde{\gamma}_{\text{ref}})$ has an intuitive interpretation: 1) When $\delta_{\pi_\theta}$ is small (policy not yet preferring $y_w$), the weight is large, providing strong gradient signal, 2) When $\delta_{\pi_\theta}$ is large (policy already strongly prefers $y_w$), the weight is small, reducing unnecessary updates, and 3) The margin term $\tilde{\gamma}_{\text{ref}}$ shifts the weighting function, ensuring that even when $\delta_{\pi_{\text{ref}}}$ is negative (reference policy misaligned), the gradient remains strong enough to push $\delta_{\pi_\theta}$ toward positive values. Building on these observations, CPO connects preference optimization to constrained RLHF through the adaptive margin $\tilde{\gamma}_{\text{ref}}$, providing a principled framework for margin-based preference learning.

### 4.5. Explicitly Constrained Preference Optimization

While CPO provides a principled framework, it relies on the selection of the hyper-parameter $\gamma$ and uses a soft penalty that *encourages* $\pi_\theta$ to prefer $y_w$ over $y_l$ instead of *ensuring* the preference. We now introduce *Conservative Explicitly Constrained Preference Optimization* (E-CPOC), which explicitly enforces preference alignment through hard constraints without requiring a reward model.

**Definition 4.11** (Explicitly Constrained RLHF). *We formulate the preference-aligned RLHF objective as:*

$$\max_\pi \mathbb{E}_{x \sim \mathcal{D}} \mathbb{E}_{y \sim \pi(\cdot|x)}[r(x,y)] - \beta \text{KL}(\pi \| \pi_{ref}) \tag{29}$$

$$s.t. \ \delta_\pi(x, y_w, y_l) \geq \gamma(x, y_w, y_l), \quad \forall (x, y_w, y_l) \in \mathcal{D} \tag{30}$$

*where $\gamma(x, y_w, y_l) > 0$ is a minimum required preference margin for each pair.*

The constraint directly ensures that the learned policy must prefer $y_w$ over $y_l$ with at least margin $\gamma$ in log-probability space. This is the condition needed to guarantee absolute preference alignment (Assumption 3.1). Similar to CPO,

we first give the log-probability ratio of the optimal policy $\delta_{\pi^*}$ in Theorem 4.12 with details and proofs in Appendix J.

**Theorem 4.12** (Log-probability Ratio of Optimal Policy). *The optimal policy for constrained RLHF satisfies:*

$$\delta_{\pi^*} = \delta_{\pi_{ref}} + \frac{\Delta r}{\beta} + \Phi(\delta_{\pi_{ref}}, \Delta r; \gamma, \tau), \qquad (31)$$

*where $\Phi(\delta_{\pi_{ref}}, \Delta r; \gamma, \tau)$ is defined as:*

$$\frac{1}{\tau} \log \left( 1 + \exp \left( \tau \left( \gamma - \delta_{\pi_{ref}} - \frac{\Delta r}{\beta} \right) \right) \right), \qquad (32)$$

*with $\tau > 0$ controlling smoothness.*

We now derive the E-CPOC loss from the explicitly constrained RLHF. From Theorem 4.12, the optimal policy satisfies $\delta_{\pi^*} = \delta_{\pi_{ref}} + \frac{\Delta r}{\beta} + \Phi(\delta_{\pi_{ref}}, \Delta r)$. The first-order optimality condition (Appendix J) reveals that the effective margin contribution of the Lagrange multipliers equals $\beta\Phi$ exactly (Proposition J.6). A key structural insight is that while the individual multiplier $\mu^*$ depends on $\pi^*$, the effective margin $\mu^*(1/\pi^*(y_w|x) + 1/\pi^*(y_l|x)) = \beta\Phi$ admits a closed form independent of $\pi^*$. Unlike CPO's margin $\tilde{\gamma}_{ref} = \gamma(1/\pi_{ref}(y_w|x) + 1/\pi_{ref}(y_l|x))$ which arises from approximating $1/\pi^*$ with $1/\pi_{ref}$, the margin $\beta\Phi$ absorbs the $1/\pi^*$ factors through the endogenous Lagrange multipliers of the KKT conditions, eliminating the approximation error entirely.

The general adaptive margin $\Phi(\delta_{\pi_{ref}}, \Delta r)$ depends on the true reward difference $\Delta r$, which is typically unknown. Rather than introducing a separate reward model to estimate $\Delta r$, we derive a reward-model-free formulation with provable guarantees by exploiting a key monotonicity property: $\Phi(\delta_{\pi_{ref}}, \Delta r)$ is monotone non-increasing in $\Delta r$ (Proposition E.1). Since preference data satisfies $\Delta r > 0$ by the Bradley-Terry model, the maximum value of $\Phi$ is achieved at $\Delta r \to 0^+$, yielding the conservative upper bound $\Phi_{cons}(\delta_{\pi_{ref}}) := \Phi(\delta_{\pi_{ref}}, 0) \geq \Phi(\delta_{\pi_{ref}}, \Delta r^*)$ for all $\Delta r^* > 0$. Applying the Bradley-Terry model, we obtain the E-CPOC loss (detailed derivation in Appendix K.1, Algorithm 2 in Appendix E):

$$\mathcal{L}_{\text{E-CPOC}}(\pi_\theta) = -\mathbb{E}_{\mathcal{D}} \left[ \log \sigma \left( \beta \left( \delta_{\pi_\theta} - \delta_{\pi_{ref}} \right) - \beta\Phi_{cons}(\delta_{\pi_{ref}}) \right) \right],$$
$$(33)$$

where $\Phi_{cons}(\delta_{\pi_{ref}}) = \Phi(\delta_{\pi_{ref}}, 0) = \frac{1}{\tau} \log \left( 1 + \exp \left( \tau(\gamma - \delta_{\pi_{ref}}) \right) \right)$.

E-CPOC requires no reward model while providing provable alignment guarantees. The adaptive margin function $\Phi_{cons}(\delta_{\pi_{ref}})$ provides stronger correction for difficult samples (where $\delta_{\pi_{ref}} \ll \gamma$) and minimal correction for easy samples (where $\delta_{\pi_{ref}} \gg \gamma$), implementing sample-adaptive weighting that emerges naturally from the constrained optimization framework. Key properties include: (1) monotonicity in $\delta_{\pi_{ref}}$, ensuring consistent behavior; (2) automatic

gradient weighting that focuses optimization on difficult pairs; and (3) interpretability as measuring constraint violation degree (Propositions E.4, E.5, E.6). The complete derivation, algorithm, theoretical analysis, and detailed property proofs are provided in Appendix E. E-CPOC is provably equivalent to explicitly constrained RLHF in the sense that $\delta_{\pi^*_{\text{E-CPOC}}} \geq \delta_{\pi^*_{\text{EC-RLHF}}}(\Delta r^*)$ for any true reward difference $\Delta r^* > 0$ (Theorem L.17 in Appendix L.3). This equivalence requires only standard statistical learning assumptions (Assumptions 4.1–4.4), without requiring a reward model. The $\ell^2$-$\delta$-proximity condition can be verified in practice through the training loss gap via a bridge lemma with an $N$-independent bound (Proposition L.3; Corollary L.18).

## 5. Preference Learning as Reranking

We provide an intuitive understanding of what will happen to DPO when the implicit Assumption 3.1 is violated, and how CPO and E-CPOC correct this failure.

The standard margin ranking loss for preference learning is:

$$\mathcal{L}_{\text{hinge}}(s_w, s_l; m) = \max(0, m - (s_w - s_l)) \qquad (34)$$

where $s_w, s_l$ are scores for the preferred and rejected responses, and $m \geq 0$ is the target margin.

We can see that DPO is the hinge loss with target margin $m = \delta_{\pi_{ref}}$. Proof can be found in Appendix M.1.

**Proposition 5.1** (DPO as Soft Margin Ranking). *DPO is a smooth approximation to margin ranking loss. In the high-temperature limit $\beta \to \infty$, for any fixed $(\delta_{\pi_\theta}, \delta_{\pi_{ref}}) \in \mathbb{R}^2$:*

$$\lim_{\beta \to \infty} \frac{1}{\beta} \mathcal{L}_{DPO}(\pi_\theta) = \max(0, \delta_{\pi_{ref}} - \delta_{\pi_\theta}). \qquad (35)$$

Through the geometric view, hinge loss leads to sharp corner at $\delta_{\pi_\theta} = \delta_{\pi_{ref}}$ and DPO loss results in smooth transition around $\delta_{\pi_\theta} = \delta_{\pi_{ref}}$ where $\beta$ controls sharpness of transition (larger $\beta \Rightarrow$ sharper corner, closer to hard hinge). When $\delta_{\pi_{ref}} < 0$ (reference policy disfavors $y_w$), DPO implements a *negative target margin*. The loss becomes zero when $\delta_{\pi_\theta} > \delta_{\pi_{ref}}$, which may still correspond to $\delta_{\pi_\theta} < 0$. In contrast, CPO provides guaranteed positive margin, as shown in Theorem 5.2, with the proof in Appendix M.2.

**Theorem 5.2** (CPO as Corrected Soft Margin Ranking). *The CPO loss function is a smooth approximation to margin ranking loss:*

$$\lim_{\beta \to \infty} \frac{1}{\beta} \mathcal{L}_{CPO}(\pi_\theta) = \max \left( 0, \delta_{\pi_{ref}} + \frac{2\gamma}{\beta} - \delta_{\pi_\theta} \right), \quad (36)$$

*with guaranteed non-negative margin:*

$$m^*_{eff} = \delta_{\pi_{ref}} + \frac{2\gamma^*}{\beta} \geq 0, \qquad (37)$$

*Where $\gamma$ is chosen according to Corollary 4.9.*

*Table 2.* Performance Comparison on Llama-3-8B-Instruct. "WR" denotes the win rate, and "LC" represents the length-controlled win rate with the average answer length. We report the 90% confidence interval for the Arena-Hard benchmark.

| Method | AlpacaEval 2 | | | Arena-Hard | |
|---|---|---|---|---|---|
| | WR(%) | LC(%) | Avg Length | WR (%) | 90% CI |
| SFT-Base (Dubey et al., 2024) | 14.22 | 13.47 | 1972 | 19.2 | $(-1.4 / +1.5)$ |
| SLiC-HF (Zhao et al., 2023) | 16.33 | 15.06 | 1998 | 22.9 | $(-1.4 / +1.6)$ |
| Contrastive-PO (Xu et al., 2024) | 18.94 | 15.95 | 2169 | 26.6 | $(-1.6 / +1.8)$ |
| DPO (Rafailov et al., 2023) | 24.60 | 25.09 | 1896 | 28.9 | $(-1.7 / +1.5)$ |
| RRHF (Yuan et al., 2023) | 15.53 | 16.40 | 1830 | 20.5 | $(-1.5 / +1.6)$ |
| RDPO (Park et al., 2024) | 24.25 | 25.01 | 1895 | 28.5 | $(-1.9 / +1.5)$ |
| ORPO (Hong et al., 2024) | 17.00 | 17.29 | 1876 | 21.7 | $(-1.1 / +1.3)$ |
| KTO (Ethayarajh et al., 2024) | 17.72 | 17.40 | 1925 | 23.1 | $(-1.0 / +1.1)$ |
| IPO (Azar et al., 2024b) | 21.48 | 20.41 | 1993 | 26.9 | $(-1.6 / +1.6)$ |
| SimPO (Meng et al., 2024) | 23.48 | 25.91 | 1810 | 30.0 | $(-2.7 / +2.3)$ |
| **CPO (Ours)** | **25.15** | **26.57** | 1879 | **32.6** | $(-1.9 / +2.4)$ |

Meanwhile, E-CPOC also provides sample-adaptive margin, as shown in Theorem 5.3, with the proof in Appendix M.3.

**Theorem 5.3** (E-CPOC as Adaptive Margin Ranking). *The E-CPOC loss implements an adaptive margin ranking loss:*

$$\lim_{\beta \to \infty} \frac{1}{\beta} \mathcal{L}_{E\text{-}CPOC}(\pi_\theta) = \max(0, \delta_{\pi_{ref}} + \Phi_{cons}(\delta_{\pi_{ref}}) - \delta_{\pi_\theta}),$$
(38)

*with guaranteed non-negative margin:*

$$m^*(\delta_{\pi_{ref}}) = \delta_{\pi_{ref}} + \Phi_{cons}(\delta_{\pi_{ref}}).$$
(39)

The margin ranking perspective provides three points: 1) DPO's equivalence to margin ranking loss with target margin $\delta_{\pi_{ref}}$ reveals that when $\delta_{\pi_{ref}} < 0$, DPO optimizes toward a *negative target*, allowing policies that prefer $y_l$ over $y_w$ to achieve low loss. This provides an intuitive explanation for the pathological behavior in Sec. 3.1: 1) CPO and E-CPOC can be understood as implementing soft margin ranking loss with guaranteed non-negative margins; 2) The softplus function $\log(1 + e^x)$ provides a smooth, differentiable approximation to the hard max operation in hinge loss. This enables gradient-based optimization while preserving the essential margin-based structure; 3) The parameter $\beta$ controls how closely the soft margin loss approximates the hard hinge loss. Larger $\beta$ yields sharper transitions and behavior closer to hard margin ranking.

## 6. Experiments

**Experimental Setup.** Following previous work (Meng et al., 2024), we use Llama-3-8B-Instruct (Dubey et al., 2024), and the princeton-nlp/llama3-ultrafeedback-armorm to conduct preference alignment. We then compare our methods with many baselines listed in Table 2 and the base model without any alignment process. Following previous work, we select AlpacaEval 2 (Li et al., 2023) and Arena-Hard (Li et al., 2024) to evaluate our method, which both evaluate conversational skills based on real-life queries.

**Main Results.** As shown in Table 2, all considered alignment methods yield consistent improvements over the SFT-Base on both AlpacaEval 2 and Arena-Hard, confirming the effectiveness of post-training preference optimization. Our proposed CPO establishes new SOTA performance among the reported methods. On AlpacaEval 2, CPO achieves the highest win rate of 25.15% (outperforming DPO at 24.60% by +0.55%) and the strongest length-controlled win rate of 26.57% (surpassing SimPO's 25.91% by +0.66%), while maintaining a competitive average response length of 1879 tokens similar to strong baselines like DPO and RDPO, without exhibiting excessive verbosity. The advantage is particularly pronounced on Arena-Hard, where CPO reaches 32.6% WR with a 90% confidence interval. This represents a +2.6% gain over SimPO (the runner-up at 30.0%) and an even larger +3.7% over DPO, highlighting CPO's superior ability to handle difficult, discriminative prompts where length bias and subtle preference distinctions matter most.

## 7. Conclusion

We prove DPO and RLHF are conditionally equal. By augmenting RLHF with explicit constraints, we propose CPO and E-CPOC which address DPO's failure modes by explicitly ensuring the preference. E-CPOC achieves provable equivalence to explicitly constrained RLHF under only standard statistical learning assumptions, requiring no reward model while providing guaranteed alignment. Our experiments demonstrate that CPO achieves SOTA performance.

**Limitations:** This paper needs to be verified on a larger scale and with a larger model to demonstrate the effectiveness of our method. In addition, the performance of E-CPOC should also be validated in experiments beyond the theoretical aspects, and training dynamics visualizations (e.g., loss curves, preference accuracy, and fraction of pairs in $\mathcal{U}$ over training steps) would further complement the theoretical characterization.

## Acknowledgement

We thank all the reviewers for their constructive suggestions and dedication to this paper. Zhiqin Yang, Yonggang Zhang, Wei Xue, and Yike Guo were supported by Hong Kong Generative AI Research & Development Center. Bo Han was supported by NSFC Major Research Plan No. 92570109 and NSFC General Program No. 62376235.

## Impact Statement

This paper presents work whose goal is to advance the field of Machine Learning. By enforcing stronger adherence to preference data, CPO may amplify biases present in the training data. This concern is shared across all preference learning methods (DPO, SimPO, IPO, RLHF) whose impact depends on data quality. Notably, CPO's $\gamma$ provides a controllable lever: smaller $\gamma$ reduces enforcement (recovering DPO at $\gamma = 0$), allowing practitioners to calibrate adherence based on data confidence.

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

# A. More Experimental Results

## A.1. Measurement of violation frequency

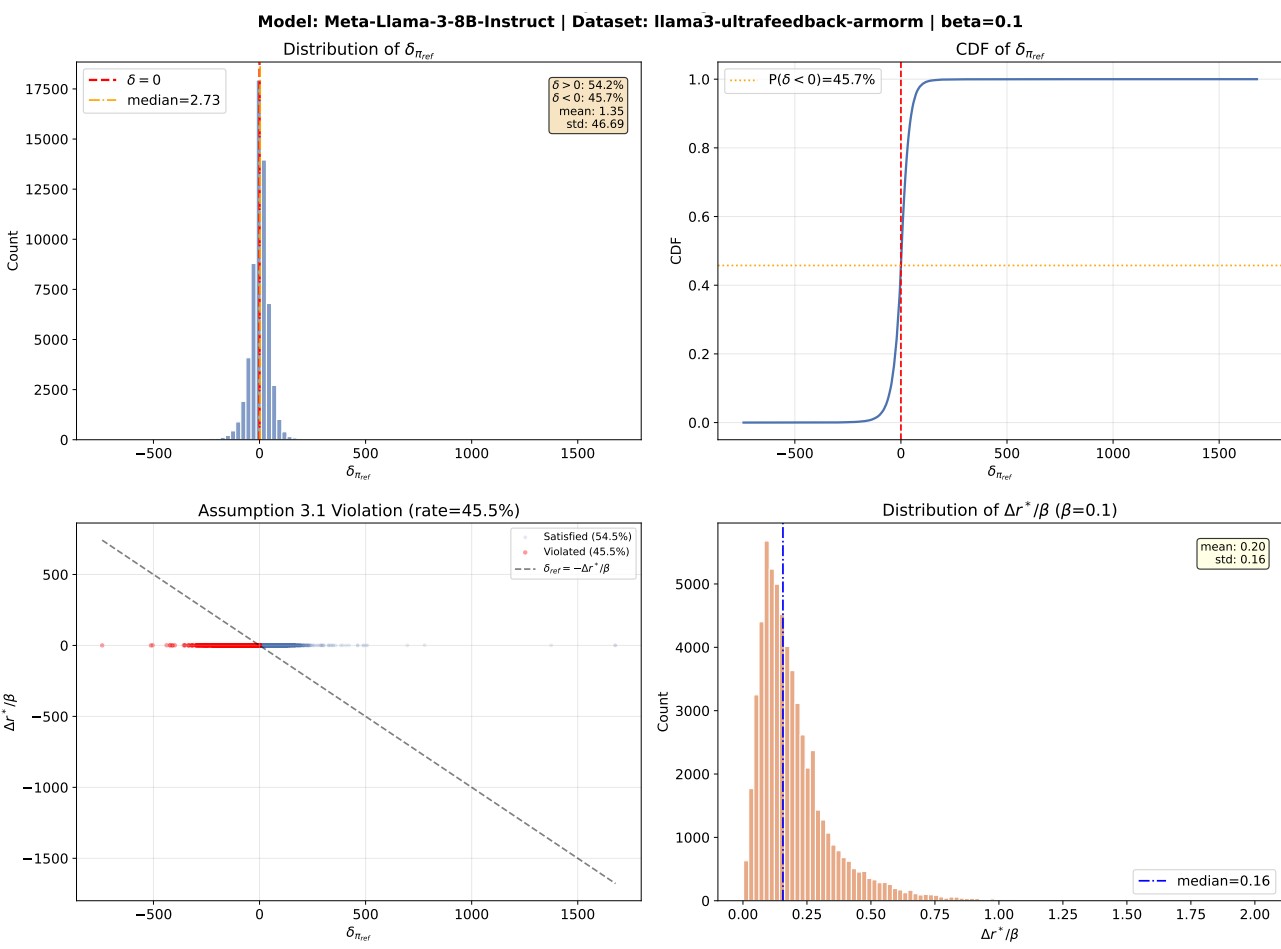

*Figure 1.* Measurement of violation frequency on Llama-3-8B-Instruct under Llama3 ultrafeedback armorn.

We compute the violation statistics. As shown in Figure 1 that Assumption 3.1 is violated for 45.5% of preference pairs** (Llama-3-8B-Instruct, $\beta = 0.1$). The reward correction $\Delta r^*/\beta$ is small (mean=0.20) relative to the large spread of $\delta_{\pi_{\text{ref}}}$ (std=46.69), meaning the reward signal often cannot compensate for the reference policy's misalignment—placing nearly half of all pairs in the regime where DPO optimizes a fundamentally different objective than RLHF (Theorem 3.5). A 45.5% violation rate on an *instruction-tuned* model confirms the pathology is far from a corner case, strongly motivating CPO's margin correction (Theorem. 4.9 and Theorem 4.10).

## A.2. Varying Reference Policy Quality

In this section, we systematically vary reference policy quality to directly validation.

**Constructing misaligned references.** From the dataset, we extract a fraction $R \in \{0.2, 0.3, 0.4\}$ of the data and use the rejected responses to SFT Llama-3-8B-Instruct (1 epoch, lr=$2 \times 10^{-5}$), forcing the model to learn to generate low-quality responses as a misaligned reference. The remaining $(1 - R)$ fraction retains original preference ordering for DPO/CPO training.

**Verifying misalignment.** We compute the $\delta_{\pi_{\text{ref}}}$ distribution of each misaligned reference on the original preference data. As shown in Table 3, as $R$ increases, the Assumption 3.1 violation rate grows correspondingly, confirming the effectiveness of misalignment.

*Table 3.* Misalignment statistics under different corruption ratios.

| Corruption Ratio | $\delta_{\pi_{\mathrm{ref}}} < 0$ (%) | Assumption 3.1 Violated (%) |
|---|---|---|
| $R = 0.2$ | 53.2 | 52.9 |
| $R = 0.3$ | 56.9 | 56.8 |
| $R = 0.4$ | 60.1 | 60.0 |

**Results.** Under each corruption ratio, we train DPO and CPO starting from the misaligned reference on the clean data. Results on AlpacaEval 2 are shown in Table 4. Note that the original evaluator `gpt-4-1106-preview` has been deprecated; we re-evaluate using `gpt-4.1` as the annotator.

*Table 4.* AlpacaEval 2 performance under misaligned reference policies.

| Corruption Ratio | Method | WR(%) | LC(%) | Avg Length |
|---|---|---|---|---|
| $R = 0.2$ | DPO | 16.82 | 17.23 | 1958 |
| | CPO | **22.47** | **27.60** | 1699 |
| $R = 0.3$ | DPO | 14.99 | 15.48 | 1894 |
| | CPO | **22.91** | **27.35** | 1686 |
| $R = 0.4$ | DPO | 15.43 | 15.98 | 1907 |
| | CPO | **20.54** | **24.34** | 1714 |

**Analysis.** **1. DPO degrades under a misaligned reference.** DPO's LC WR drops across all three corruption ratios, with stronger corruption leading to worse performance. This is consistent with Proposition 3.4.

**2. CPO remains robust under the same conditions.** CPO achieves LC of 27.60% and 27.35% at $R = 0.2$ and $R = 0.3$ respectively, remaining stable. This validates the core role of the margin term $\tilde{\gamma}_{\mathrm{ref}}$ (Eq. 27): even when $\delta_{\pi_{\mathrm{ref}}}$ is negative, the margin preserves gradient strength, enabling the policy to push $\delta_{\pi_\theta}$ past 0 and escape $\mathcal{U}$.

**3. Fraction in $\mathcal{U}$ directly validates the theory.** The fraction in $\mathcal{U}$ trajectory during training exhibits a characteristic three-phase pattern that directly reflects the theoretical mechanism:

- **Phase 1 (Initialization):** At step 0, the policy is identical to the reference (frac in $\mathcal{U}$ = 0). Since $\delta_{\pi_\theta} = \delta_{\pi_{\mathrm{ref}}}$ for all samples, the condition $\delta_{\pi_\theta} > \delta_{\pi_{\mathrm{ref}}}$ is not satisfied, so no samples fall in $\mathcal{U}$.

- **Phase 2 (Entry into $\mathcal{U}$):** As training begins, gradients push $\delta_{\pi_\theta}$ upward. For misaligned samples where $\delta_{\pi_{\mathrm{ref}}} < 0$, the policy improves relative to the reference but has not yet crossed 0, resulting in $\delta_{\pi_{\mathrm{ref}}} < \delta_{\pi_\theta} < 0$, exactly the $\mathcal{U}$ region. This causes frac in $\mathcal{U}$ to rise.

- **Phase 3 (Escape attempt):** As $\delta_{\pi_\theta}$ continues to increase and some samples cross 0 ($\delta_{\pi_\theta} > 0$), they exit $\mathcal{U}$, causing frac in $\mathcal{U}$ to decrease. The critical divergence emerges:
    - **DPO:** The gradient weight $\sigma(-\beta(\delta_{\pi_\theta} - \delta_{\pi_{\mathrm{ref}}}))$ weakens as $\delta_{\pi_\theta}$ approaches 0 (Proposition 3.4). Many samples get stuck at $\delta_{\pi_\theta} \approx 0^-$, unable to escape.
    - **CPO:** The margin term $\tilde{\gamma}_{\mathrm{ref}}$ shifts the gradient weighting function, ensuring strong gradient signal even near the $\mathcal{U}$ boundary. Most samples successfully push past $\delta_{\pi_\theta} = 0$, and frac in $\mathcal{U}$ rapidly drops.

The training dynamics of frac in $\mathcal{U}$ are visualized in Figure 2. The difference in frac in $\mathcal{U}$ directly corresponds to the theoretical contrast between DPO's weak gradient near the $\mathcal{U}$ boundary (Proposition 3.4) and CPO's margin-corrected gradients (Theorem 5.2).

### A.3. Sensitivity Analysis of $\gamma$

We conduct a sensitivity analysis of the hyperparameter $\gamma$ on AlpacaEval 2. Note that the original evaluator `gpt-4-1106-preview` has been deprecated; we re-evaluate using `gpt-4.1` as the annotator. Results are shown

*Figure 2.* Fraction of training samples in the undesirable solution space $\mathcal{U}$ (Definition 3.3) over training steps under different corruption ratios $R \in \{0.2, 0.3, 0.4\}$.

in Table 5. CPO performs robustly across $\gamma \in [0.2, 0.4]$ (WR 26–28%, LC 31–34%), with peak performance at $\gamma = 0.25$. Performance drops notably below $0.2$, where the margin correction becomes insufficient to address the assumption violation. In all experiments reported in the main paper, we use $\gamma = 0.25$.

*Table 5.* Sensitivity analysis of $\gamma$ on AlpacaEval 2 using Llama-3-8B-Instruct.

| $\gamma$ | WR(%) | LC(%) | Avg Length |
|---|---|---|---|
| 0.10 | 20.45 | 26.46 | 1605 |
| 0.15 | 20.87 | 26.32 | 1635 |
| 0.20 | 27.08 | 32.56 | 1705 |
| **0.25** | **28.36** | **33.97** | 1702 |
| 0.30 | 26.31 | 31.40 | 1700 |
| 0.35 | 26.10 | 30.92 | 1727 |
| 0.40 | 27.53 | 32.87 | 1729 |

### A.4. Evaluation on IFEval

Following the reviewer's suggestion, we add IFEval (Zhou et al., 2023) as an additional benchmark to evaluate instruction-following capability. Results are shown in Table 6. CPO achieves the highest performance on both strict and loose accuracy, confirming that the improvement from CPO extends beyond conversational benchmarks to instruction-following tasks.

### A.5. Comparison with Clipped-Reference Baseline

To isolate the effect of CPO's adaptive margin from generic margin regularization, we compare with a clipped-reference baseline that clips $\delta_{\pi_{\text{ref}}}$ to be non-negative before applying the standard DPO loss. Results on AlpacaEval 2 are shown in Table 7. CPO substantially outperforms the clipped-reference baseline, demonstrating that the adaptive margin $\tilde{\gamma}_{\text{ref}}$ provides benefits beyond simply preventing negative margins

*Table 6.* IFEval results on Llama-3-8B-Instruct.

| Method | Strict Acc | Loose Acc |
|---|---|---|
| Llama-8B-Instruct | 32.35% | 39.56% |
| Llama-8B-Instruct-DPO | 34.01% | 40.67% |
| Llama-8B-Instruct-RDPO | 34.57% | 43.62% |
| Llama-8B-Instruct-SimPO | 33.83% | 42.81% |
| Llama-8B-Instruct-CPO | **35.12%** | **43.99%** |

*Table 7.* Comparison with clipped-reference baseline on AlpacaEval 2.

| Method | WR(%) | LC(%) | Avg Length |
|---|---|---|---|
| Llama-8B-Instruct-clipped-ref | 17.91% | 23.86% | 1586 |
| Llama-8B-Instruct-CPO | **28.36%** | **33.97%** | 1702 |

# B. Related Work

RLHF has become the standard approach for aligning LLMs with human preferences (Christiano et al., 2017; Stiennon et al., 2020; Ouyang et al., 2022). The framework typically uses the Bradley-Terry model (Bradley & Terry, 1952) to learn reward functions from pairwise preference data, followed by policy optimization with KL regularization to prevent reward over-optimization (Ziegler et al., 2019; Gao et al., 2023). While effective, RLHF requires training a separate reward model and applying RL algorithms like PPO (Schulman et al., 2017), making it computationally expensive and potentially unstable.

DPO (Rafailov et al., 2023) proposes to bypass explicit reward modeling by directly optimizing policies on preference data, claiming theoretical equivalence to RLHF. This simplicity has led to widespread adoption and numerous variants. IPO (Azar et al., 2024a) is proposed for better calibration and robustness to noise. KTO (Ethayarajh et al., 2024) extends to binary feedback rather than pairwise comparisons. ORPO (Hong et al., 2024) combines preference optimization with supervised fine-tuning. However, all these variants retain structural similarities to DPO and do not address the conditional nature of the DPO-RLHF equivalence that we identify.

Recent works have begun examining theoretical properties of preference-based methods. Azar et al. (2024a) analyze the Nash equilibrium properties of preference learning and propose regularization for improved sample efficiency. Munos et al. (2024) study the game-theoretic foundations of RLHF. While these works provide valuable theoretical insights, none have systematically characterized the conditions under which DPO-RLHF equivalence holds or fails. Our work is the first to identify the implicit assumption, prove the equivalence is conditional, characterize precise failure conditions, and provide methods with provable alignment guarantees.

**Reference-free methods.** SimPO (Meng et al., 2024) replaces the reward reparameterization $r(x, y) = \beta \log \frac{\pi_\theta(y|x)}{\pi_{\text{ref}}(y|x)}$ with length-normalized log-probability as an implicit reward, a heuristic not derived from any RLHF objective or Bradley-Terry model. By removing the reference policy entirely, SimPO sidesteps the assumption violation identified in this work (Assumption 3.1), but at the cost of abandoning the RLHF-BT framework: it cannot claim equivalence to any reward-maximizing objective with KL regularization. In contrast, CPO and E-CPOC retain the full RLHF-BT chain with formal guarantees: absolute advantage (Theorem 4.9), avoidance of $\mathcal{U}$ (Theorem 4.10), and provable equivalence to constrained RLHF (Theorem L.17). Our goal is to understand *why* DPO's RLHF equivalence breaks and *how* to fix it with guarantees, which is fundamentally different from designing reference-free heuristics. Nonetheless, investigating how reference-free methods relate to the margin ranking perspective (Section 5) is an interesting direction for future work.

# C. Explanation about Preference Learning as Reranking

The margin ranking loss is a fundamental tool in learning-to-rank (Burges et al., 2005; Cao et al., 2007), where it ensures that relevant items are ranked above irrelevant ones with sufficient margin. Our analysis shows that preference learning can be viewed as a reranking problem in log-probability space, where the margin ensures that preferred responses are ranked above dispreferred ones. This unified view through margin ranking loss not only provides an intuitive understanding of DPO's failure modes and our corrections, but also connects preference learning to a rich body of existing theory and practice

in ranking and margin-based learning.

# D. Proofs.

**Clarification on Assumption 3.1.** The algebraic substitution in Eq. 7 holds regardless of the sign of $\delta_{\pi^*}$. Assumption 3.1 is not needed for the substitution itself, but for the equivalence of objectives between DPO and RLHF. The DPO loss $-\log \sigma(\beta(\delta_{\pi_\theta} - \delta_{\pi_{\text{ref}}}))$ is monotonically decreasing in $\delta_{\pi_\theta}$, so it always pushes $\delta_{\pi_\theta}$ upward toward preferring $y_w$. Meanwhile, the RLHF optimal policy satisfies $\delta_{\pi^*} = \delta_{\pi_{\text{ref}}} + \Delta r^*/\beta$, which can be $\leq 0$. When this happens, DPO's optimum ($\delta_{\pi_\theta} \to \infty$) diverges from RLHF's optimum ($\delta_{\pi^*} < 0$), and the two methods optimize fundamentally different objectives (Theorem 3.5).

## D.1. Proof of Necessary Condition for Assumption 3.1

Proof of Proposition 3.2 (Necessary Condition for Assumption 3.1).

*Proof.* Human preference $y_w \succ y_l$ implies $r^*(y_w) - r^*(y_l) > 0$.

By the core relationship (Equation (5)):

$$\delta_{\pi^*} = \delta_{\pi_{\text{ref}}} + \frac{r^*(y_w) - r^*(y_l)}{\beta} \tag{40}$$

For Assumption 3.1 to hold, we require $\delta_{\pi^*} > 0$:

$$\delta_{\pi_{\text{ref}}} + \frac{r^*(y_w) - r^*(y_l)}{\beta} > 0 \tag{41}$$

Rearranging yields the necessary condition. $\qquad\square$

## D.2. Proof of Non-emptiness and Weak Gradient

Proof of Proposition 3.4 (Non-emptiness and Weak gradient).

*Proof.* **Non-emptiness:** Since $\delta_{\pi_{\text{ref}}} < -\Delta r^*/\beta < 0$, any policy with $\delta_{\pi_{\text{ref}}} < \delta_\pi < 0$ belongs to $\mathcal{U}$.

**Weak gradient:** The gradient of $\mathcal{L}_{\text{DPO}}$ with respect to policy parameters is:

$$\nabla_\theta \mathcal{L}_{\text{DPO}} = -\mathbb{E}\left[\sigma(-\beta(\delta_{\pi_\theta} - \delta_{\pi_{\text{ref}}})) \cdot \beta \cdot (\nabla_\theta \log \pi_\theta(y_w|x) - \nabla_\theta \log \pi_\theta(y_l|x))\right] \tag{42}$$

For $\pi \in \mathcal{U}$, we have $\delta_\pi > \delta_{\pi_{\text{ref}}}$, so $\delta_\pi - \delta_{\pi_{\text{ref}}} > 0$, making $-\beta(\delta_\pi - \delta_{\pi_{\text{ref}}}) < 0$ and thus $\sigma(-\beta(\delta_\pi - \delta_{\pi_{\text{ref}}})) < 0.5$. The gradient direction pushes $\delta_{\pi_\theta}$ to increase (i.e., $\nabla_\theta \log \pi_\theta(y_w|x) - \nabla_\theta \log \pi_\theta(y_l|x)$ increases).

However, as $\delta_\pi$ increases toward 0 (the boundary of preference violation), the difference $\delta_\pi - \delta_{\pi_{\text{ref}}}$ becomes larger, causing $-\beta(\delta_\pi - \delta_{\pi_{\text{ref}}})$ to become more negative and $\sigma(-\beta(\delta_\pi - \delta_{\pi_{\text{ref}}})) \to 0$. This causes the gradient magnitude to become progressively weaker, constituting a weak-gradient problem conditioned on the quality of $\pi_{\text{ref}}$. Since $\delta_\pi < 0$ throughout $\mathcal{U}$, the policy remains trapped in the preference-violating region while the loss continues to decrease. $\qquad\square$

## D.3. Proof of Conditional Equivalence of DPO and RLHF

Proof of Theorem 3.5 (Conditional Equivalence of DPO and RLHF)

*Proof.* We prove both directions of the if-and-only-if statement. Throughout, let $\pi^*$ denote the RLHF-optimal policy and $\pi_{\text{DPO}}$ denote the DPO-optimal policy.

($\Rightarrow$) **Sufficiency:** Assume Condition (15) holds for all $(x, y_w, y_l) \in \mathcal{D}$.

*Step 1: RLHF-optimal policy respects human preferences.* By Eq. 3, the RLHF-optimal policy satisfies:

$$\delta_{\pi^*}(x, y_w, y_l) = \delta_{\pi_{\text{ref}}}(x, y_w, y_l) + \frac{r^*(x, y_w) - r^*(x, y_l)}{\beta}. \tag{43}$$

Under Condition (15), we have $\delta_{\pi_{\text{ref}}} > -\Delta r^*/\beta$, where $\Delta r^* := r^*(x, y_w) - r^*(x, y_l) > 0$. Therefore:

$$\delta_{\pi^*} = \delta_{\pi_{\text{ref}}} + \frac{\Delta r^*}{\beta} > 0, \tag{44}$$

implying $\pi^*(y_w|x) > \pi^*(y_l|x)$, which aligns with the human preference $y_w \succ y_l$.

*Step 2: Bradley-Terry model is well-defined at $\pi^*$.* Since $\delta_{\pi^*} > 0$, the Bradley-Terry preference probability at $\pi^*$ is:

$$p^*(y_w \succ y_l|x) = \sigma(r^*(x, y_w) - r^*(x, y_l)) = \sigma(\beta(\delta_{\pi^*} - \delta_{\pi_{\text{ref}}})) > 0.5. \tag{45}$$

This is consistent with the human preference structure, validating the use of the Bradley-Terry model in DPO's derivation.

*Step 3: $\pi^*$ is a stationary point of the DPO objective.* The DPO objective is:

$$\mathcal{L}_{\text{DPO}}(\pi) = -\mathbb{E}_{(x,y_w,y_l)\sim\mathcal{D}} \left[ \log \sigma(\beta(\delta_\pi - \delta_{\pi_{\text{ref}}})) \right]. \tag{46}$$

Taking the gradient with respect to policy parameters $\theta$:

$$\nabla_\theta \mathcal{L}_{\text{DPO}} = -\mathbb{E} \left[ \sigma(-\beta(\delta_{\pi_\theta} - \delta_{\pi_{\text{ref}}})) \cdot \beta \cdot \nabla_\theta \delta_{\pi_\theta} \right], \tag{47}$$

where $\nabla_\theta \delta_{\pi_\theta} = \nabla_\theta \log \pi_\theta(y_w|x) - \nabla_\theta \log \pi_\theta(y_l|x)$.

By the reward reparameterization (Lemma 4), at $\pi = \pi^*$:

$$r^*(x, y) = \beta \log \frac{\pi^*(y|x)}{\pi_{\text{ref}}(y|x)} + \beta \log Z(x), \tag{48}$$

where $Z(x)$ is the partition function. Substituting into the Bradley-Terry model:

$$p^*(y_w \succ y_l|x) = \sigma(\beta(\delta_{\pi^*} - \delta_{\pi_{\text{ref}}})). \tag{49}$$

This shows that $\pi^*$ satisfies the maximum likelihood condition for the Bradley-Terry model parameterized by DPO, making it a stationary point of $\mathcal{L}_{\text{DPO}}$.

*Step 4: $\pi^*$ is the global optimum of the DPO objective.* The DPO loss $\mathcal{L}_{\text{DPO}}(\pi) = -\mathbb{E}[\log \sigma(\beta(\delta_\pi - \delta_{\pi_{\text{ref}}}))]$ is strictly convex in $\delta_\pi$. To see this, note that $f(z) = -\log \sigma(z) = \log(1 + e^{-z})$ has second derivative $f''(z) = \sigma(z)\sigma(-z) > 0$, establishing strict convexity.

Under Condition (15), $\pi^*$ satisfies the first-order optimality condition (Step 3). Since $\mathcal{L}_{\text{DPO}}$ depends only on $\delta_\pi$ and is strictly convex in this quantity, the optimal $\delta_\pi$ values are uniquely determined. Specifically, any policy $\tilde{\pi}$ minimizing $\mathcal{L}_{\text{DPO}}$ must satisfy:

$$\delta_{\tilde{\pi}}(x, y_w, y_l) = \delta_{\pi^*}(x, y_w, y_l) \quad \forall (x, y_w, y_l) \in \mathcal{D}. \tag{50}$$

This equality of log-probability ratios implies that $\tilde{\pi}$ and $\pi^*$ have identical preference structures: $\tilde{\pi}(y_w|x)/\tilde{\pi}(y_l|x) = \pi^*(y_w|x)/\pi^*(y_l|x)$ for all preference pairs. Therefore, $\pi_{\text{DPO}}$ and $\pi^*$ are equivalent in terms of preference ordering, establishing that DPO and RLHF optimize the same objective under the given condition.

($\Leftarrow$) **Necessity:** We prove the contrapositive: if Condition (15) is violated, then DPO and RLHF optimize different objectives.

Suppose there exists $(x_0, y_w^0, y_l^0) \in \mathcal{D}$ such that:

$$\delta_{\pi_{\text{ref}}}(x_0, y_w^0, y_l^0) \leq -\frac{r^*(x_0, y_w^0) - r^*(x_0, y_l^0)}{\beta}. \tag{51}$$

*Step 1: RLHF-optimal policy violates human preference at $(x_0, y_w^0, y_l^0)$.* By Eq. 3:

$$\delta_{\pi^*}(x_0, y_w^0, y_l^0) = \delta_{\pi_{\text{ref}}}(x_0, y_w^0, y_l^0) + \frac{r^*(x_0, y_w^0) - r^*(x_0, y_l^0)}{\beta} \leq 0, \tag{52}$$

implying $\pi^*(y_l^0|x_0) \geq \pi^*(y_w^0|x_0)$. Thus, $\pi^*$ prefers the dispreferred response at this data point, yet RLHF accepts this as optimal due to the KL regularization constraint.

*Step 2: $\pi^*$ is not optimal for the DPO objective.* We construct a policy $\pi_\epsilon$ that achieves lower DPO loss than $\pi^*$. For the prompt $x_0$ where the condition is violated, define:

$$\pi_\epsilon(y|x_0) = \frac{\pi^*(y|x_0)\exp(\epsilon[\mathbb{I}(y = y_w^0) - \mathbb{I}(y = y_l^0)])}{\sum_{y'}\pi^*(y'|x_0)\exp(\epsilon[\mathbb{I}(y' = y_w^0) - \mathbb{I}(y' = y_l^0)])} \tag{53}$$

for some $\epsilon > 0$, and $\pi_\epsilon(y|x) = \pi^*(y|x)$ for all $x \neq x_0$. This construction ensures $\pi_\epsilon$ is a valid probability distribution with:

$$\delta_{\pi_\epsilon}(x_0, y_w^0, y_l^0) = \delta_{\pi^*}(x_0, y_w^0, y_l^0) + 2\epsilon. \tag{54}$$

The DPO loss can be decomposed as:

$$\mathcal{L}_{\text{DPO}}(\pi) = \mathbb{E}_{(x,y_w,y_l)\sim\mathcal{D}}\left[-\log\sigma(\beta(\delta_\pi(x, y_w, y_l) - \delta_{\pi_{\text{ref}}}(x, y_w, y_l)))\right]. \tag{55}$$

Let $p_0$ denote the probability mass of $(x_0, y_w^0, y_l^0)$ in $\mathcal{D}$. Since $\pi_\epsilon = \pi^*$ for all $x \neq x_0$, the difference in losses is:

$$\mathcal{L}_{\text{DPO}}(\pi_\epsilon) - \mathcal{L}_{\text{DPO}}(\pi^*) = p_0\left[-\log\sigma(\beta(\delta_{\pi^*} + 2\epsilon - \delta_{\pi_{\text{ref}}})) + \log\sigma(\beta(\delta_{\pi^*} - \delta_{\pi_{\text{ref}}}))\right] \tag{56}$$

$$= p_0\log\frac{\sigma(\Delta r^*)}{\sigma(\Delta r^* + 2\beta\epsilon)}, \tag{57}$$

where we used $\delta_{\pi^*} - \delta_{\pi_{\text{ref}}} = \Delta r^*/\beta$ from Step 1.

Since $\sigma$ is strictly increasing and $\epsilon > 0$, we have $\sigma(\Delta r^* + 2\beta\epsilon) > \sigma(\Delta r^*)$, which implies:

$$\mathcal{L}_{\text{DPO}}(\pi_\epsilon) < \mathcal{L}_{\text{DPO}}(\pi^*). \tag{58}$$

Therefore, $\pi^*$ is not a global minimum of the DPO objective.

*Step 3: DPO optimizes a different objective.* Since $\pi^*$ is not optimal for DPO, we have $\pi_{\text{DPO}} \neq \pi^*$. RLHF optimizes $\mathbb{E}[r] - \beta\,\text{KL}(\pi\|\pi_{\text{ref}})$ and converges to $\pi^*$, while DPO optimizes $\mathbb{E}[\log\sigma(\beta(\delta_\pi - \delta_{\pi_{\text{ref}}}))]$ and converges to $\pi_{\text{DPO}} \neq \pi^*$. Therefore, the two methods optimize fundamentally different objectives when Condition (15) is violated. □

### D.4. Proof of Optimal Policy for Constrained RLHF

**Remark D.1** (Single Preference Pair per Prompt (Notational Convention)). *For notational clarity, we adopt the convention that each response $y$ appears at most once as a winner and once as a loser for each prompt $x$ in the preference dataset $\mathcal{D}$. This is purely a notational simplification and does not limit the generality of our results: when a response appears in multiple preference pairs, the margin contributions aggregate linearly, and all theoretical guarantees remain valid (see Proposition G.1).*

**Definition D.2** (Augmented Reward). *Under the notation above, for a given preference dataset $\mathcal{D}$, define the augmented reward function:*

$$\tilde{r}(x,y) = \begin{cases} r(x,y) + \gamma & \text{if } \exists(x, y_w, y_l) \in \mathcal{D} \text{ with } y = y_w \\ r(x,y) - \gamma & \text{if } \exists(x, y_w, y_l) \in \mathcal{D} \text{ with } y = y_l \\ r(x,y) & \text{otherwise} \end{cases} \tag{59}$$

*This corresponds to the case where each response appears at most once as a winner and once as a loser for each prompt, with uniform weighting over preference pairs.*

Proof of Theorem 4.8(Optimal Policy for Constrained RLHF).

*Proof.* We first reformulate the margin term in the objective function. Under the notation in Remark D.1, the margin term can be written as:

$$\gamma\mathbb{E}_{(x,y_w,y_l)\sim\mathcal{D}}[\delta_\pi] = \gamma\mathbb{E}_{(x,y_w,y_l)\sim\mathcal{D}}[\log\pi(y_w|x) - \log\pi(y_l|x)] \tag{60}$$

For each prompt $x$, collecting all preference pairs and responses:

$$= \gamma \mathbb{E}_x \left[ \sum_{(y_w, y_l) \in \mathcal{P}(x)} p(y_w, y_l | x)(\log \pi(y_w | x) - \log \pi(y_l | x)) \right] \tag{61}$$

This can be rewritten as an expectation over responses by defining:

$$c(x, y) = \gamma \sum_{(y_w, y_l) \in \mathcal{P}(x)} p(y_w, y_l | x)(\mathbb{I}(y = y_w) - \mathbb{I}(y = y_l)) \tag{62}$$

Then:

$$\gamma \mathbb{E}_{(x, y_w, y_l) \sim \mathcal{D}}[\delta_\pi] = \mathbb{E}_x \left[ \sum_y c(x, y) \log \pi(y | x) \right] \tag{63}$$

For a fixed prompt $x$, the Lagrangian is:

$$\mathcal{L}_x = \sum_y \pi(y | x) r(x, y) - \beta \sum_y \pi(y | x) \log \frac{\pi(y | x)}{\pi_{\text{ref}}(y | x)} + \sum_y c(x, y) \log \pi(y | x) - \lambda(x) \left( \sum_y \pi(y | x) - 1 \right) \tag{64}$$

Taking the derivative with respect to $\pi(y | x)$ and setting to zero:

$$\frac{\partial \mathcal{L}_x}{\partial \pi(y | x)} = r(x, y) - \beta \log \frac{\pi(y | x)}{\pi_{\text{ref}}(y | x)} - \beta + \frac{c(x, y)}{\pi(y | x)} - \lambda(x) = 0 \tag{65}$$

Rearranging gives Equation (18). For a preference pair, taking the difference between the conditions for $y_w$ and $y_l$ yields Equation (19). □

### D.5. Proof of Absolute Advantage Guarantee

Before providing detailed proof, we first introduce the Lemma.

**Lemma D.3** (Absolute Advantage Guarantee: Sample Level). *For a preference pair $(x, y_w, y_l)$, if the hyper-parameter $\gamma$ in CPO satisfies:*

$$\gamma > \beta \cdot \frac{\max \left\{ 0, -\delta_{\pi_{ref}}(x, y_w, y_l) - \frac{\Delta r^*(x, y_w, y_l)}{\beta} \right\}}{\frac{1}{\pi_{ref}(y_w | x)} + \frac{1}{\pi_{ref}(y_l | x)}}, \tag{66}$$

*then CPO guarantees absolute advantage: $\delta_{\pi_{CPO}^*}(x, y_w, y_l) > 0$ for this pair.*

Proof of Lemma D.3 (Absolute Advantage Guarantee)

*Proof.* Let $\pi_{\text{CPO}}^*$ denote the optimal policy for CPO, satisfying:

$$r(x, y_w) - r(x, y_l) = \beta(\delta_{\pi_{\text{CPO}}^*} - \delta_{\pi_{\text{ref}}}) - \tilde{\gamma}_{\text{ref}}(x, y_w, y_l), \tag{67}$$

where $\tilde{\gamma}_{\text{ref}}(x, y_w, y_l) = \gamma \left( \frac{1}{\pi_{\text{ref}}(y_w | x)} + \frac{1}{\pi_{\text{ref}}(y_l | x)} \right)$ is the reference-based adaptive margin. This results in a sample-adaptive larger margin than DPO:

$$\delta_{\pi_{\text{CPO}}^*} = \delta_{\pi_{\text{DPO}}^*} + \frac{\tilde{\gamma}_{\text{ref}}(x, y_w, y_l)}{\beta}. \tag{68}$$

**Part 1 (Margin relationship):** From Eq. 20 (derived from Theorem 4.8 with $c(x, y_w) = \gamma$ and $c(x, y_l) = -\gamma$), we have:

$$r(y_w) - r(y_l) = \beta(\delta_{\pi^*} - \delta_{\pi_{\text{ref}}}) - \gamma \left( \frac{1}{\pi^*(y_w | x)} + \frac{1}{\pi^*(y_l | x)} \right) \tag{69}$$

In CPO, we approximate the optimal policy probabilities in the margin term with the reference policy probabilities (as justified in Eq. (24) and Proposition H.2):

$$\gamma \left( \frac{1}{\pi^*(y_w|x)} + \frac{1}{\pi^*(y_l|x)} \right) \approx \gamma \left( \frac{1}{\pi_{\text{ref}}(y_w|x)} + \frac{1}{\pi_{\text{ref}}(y_l|x)} \right) = \tilde{\gamma}_{\text{ref}}(x, y_w, y_l) \tag{70}$$

Therefore:

$$r(y_w) - r(y_l) = \beta(\delta_{\pi^*_{\text{CPO}}} - \delta_{\pi_{\text{ref}}}) - \tilde{\gamma}_{\text{ref}}(x, y_w, y_l) \tag{71}$$

For standard DPO ($\gamma = 0$):

$$r(y_w) - r(y_l) = \beta(\delta_{\pi^*_{\text{DPO}}} - \delta_{\pi_{\text{ref}}}) \tag{72}$$

Rearranging the CPO equation:

$$\delta_{\pi^*_{\text{CPO}}} = \delta_{\pi_{\text{ref}}} + \frac{r(y_w) - r(y_l)}{\beta} + \frac{\tilde{\gamma}_{\text{ref}}(x, y_w, y_l)}{\beta} \tag{73}$$

Comparing with DPO:

$$\delta_{\pi^*_{\text{DPO}}} = \delta_{\pi_{\text{ref}}} + \frac{r(y_w) - r(y_l)}{\beta} \tag{74}$$

Subtracting yields:

$$\delta_{\pi^*_{\text{CPO}}} = \delta_{\pi^*_{\text{DPO}}} + \frac{\tilde{\gamma}_{\text{ref}}(x, y_w, y_l)}{\beta} \tag{75}$$

**Part 2 (Absolute advantage):** To ensure $\delta_{\pi^*_{\text{CPO}}}(x, y_w, y_l) > 0$, we require:

$$\delta_{\pi_{\text{ref}}}(x, y_w, y_l) + \frac{\Delta r^*(x, y_w, y_l)}{\beta} + \frac{\tilde{\gamma}_{\text{ref}}(x, y_w, y_l)}{\beta} > 0 \tag{76}$$

Rearranging:

$$\frac{\tilde{\gamma}_{\text{ref}}(x, y_w, y_l)}{\beta} > -\delta_{\pi_{\text{ref}}}(x, y_w, y_l) - \frac{\Delta r^*(x, y_w, y_l)}{\beta} \tag{77}$$

$$\tilde{\gamma}_{\text{ref}}(x, y_w, y_l) > -\beta \delta_{\pi_{\text{ref}}}(x, y_w, y_l) - \Delta r^*(x, y_w, y_l) \tag{78}$$

Substituting the definition of $\tilde{\gamma}_{\text{ref}}$:

$$\gamma \left( \frac{1}{\pi_{\text{ref}}(y_w|x)} + \frac{1}{\pi_{\text{ref}}(y_l|x)} \right) > -\beta \delta_{\pi_{\text{ref}}}(x, y_w, y_l) - \Delta r^*(x, y_w, y_l) \tag{79}$$

Solving for $\gamma$:

$$\gamma > \frac{-\beta \delta_{\pi_{\text{ref}}}(x, y_w, y_l) - \Delta r^*(x, y_w, y_l)}{\frac{1}{\pi_{\text{ref}}(y_w|x)} + \frac{1}{\pi_{\text{ref}}(y_l|x)}} \tag{80}$$

When $-\delta_{\pi_{\text{ref}}} - \frac{\Delta r^*}{\beta} \leq 0$ (i.e., standard DPO already satisfies absolute advantage for this pair), any $\gamma \geq 0$ suffices. Otherwise, we need $\gamma$ to compensate for the deficit. This is captured by:

$$\gamma > \beta \cdot \frac{\max \left\{ 0, -\delta_{\pi_{\text{ref}}}(x, y_w, y_l) - \frac{\Delta r^*(x, y_w, y_l)}{\beta} \right\}}{\frac{1}{\pi_{\text{ref}}(y_w|x)} + \frac{1}{\pi_{\text{ref}}(y_l|x)}} \tag{81}$$

$\square$

**Remark D.4** (Relationship to Constant Margin Approximation). *For simplified analysis, one can approximate the adaptive margin with a constant by setting:*

$$\tilde{\gamma}_{ref}(x, y_w, y_l) \approx 2\gamma' \quad where \quad \gamma' = \gamma \cdot \mathbb{E}_{(x,y_w,y_l)\sim\mathcal{D}}\left[\frac{1}{2}\left(\frac{1}{\pi_{ref}(y_w|x)} + \frac{1}{\pi_{ref}(y_l|x)}\right)\right] \tag{82}$$

*This constant approximation simplifies hyperparameter interpretation but comes at the cost of:*

- ***Looser bounds***: *The constant must be chosen to handle the worst-case preference pair, leading to $\gamma^* \approx \beta \max_{(x,y_w,y_l)\in\mathcal{D}} \max\{0, -\delta_{\pi_{ref}} - \frac{\Delta r^*}{\beta}\}$, which can be significantly larger than the adaptive $\gamma^*$ in Eq. (27).*

- ***Suboptimal regularization***: *Over-regularizes high-confidence pairs (where $\pi_{ref}$ assigns large probabilities) and under-regularizes low-confidence pairs.*

*In practice, we recommend using the full adaptive margin as implemented in Algorithm 1, which provides tighter theoretical guarantees and better empirical performance while requiring no additional computational cost (the margin terms are precomputed once before training).*

Proof of Theorem 4.9 (Absolute Advantage Guarantee), given the Lemma above.

*Proof.* For each $(x, y_w, y_l) \in \mathcal{D}$, Theorem D.3 requires:

$$\gamma > \beta \cdot \frac{\max\left\{0, -\delta_{\pi_{\text{ref}}}(x, y_w, y_l) - \frac{\Delta r^*(x,y_w,y_l)}{\beta}\right\}}{\frac{1}{\pi_{\text{ref}}(y_w|x)} + \frac{1}{\pi_{\text{ref}}(y_l|x)}} \tag{83}$$

To satisfy this condition for all preference pairs simultaneously, we take the maximum over the dataset:

$$\gamma^* = \max_{(x,y_w,y_l)\in\mathcal{D}} \beta \cdot \frac{\max\left\{0, -\delta_{\pi_{\text{ref}}}(x, y_w, y_l) - \frac{\Delta r^*(x,y_w,y_l)}{\beta}\right\}}{\frac{1}{\pi_{\text{ref}}(y_w|x)} + \frac{1}{\pi_{\text{ref}}(y_l|x)}} \tag{84}$$

Any $\gamma \geq \gamma^*$ then satisfies the condition for all pairs, guaranteeing absolute advantage across the entire dataset. $\square$

## D.6. Proof of Avoiding Pathological Convergence in CPO

Proof of Theorem 4.10 (CPO Avoids Pathological Convergence).

*Proof.* Recall $\mathcal{U} = \{\pi : \delta_\pi < 0 \text{ and } \delta_\pi > \delta_{\pi_{\text{ref}}}\}$.

By Theorem D.3, when $\gamma \geq \gamma^*$:

$$\delta_{\pi^*_{\text{CPO}}} > 0 \tag{85}$$

Therefore $\pi^*_{\text{CPO}} \notin \mathcal{U}$. Under Assumption I.1 (Lipschitz continuous gradients, bounded parameter domain), the CPO loss $\mathcal{L}_{\text{CPO}}$ is convex in the log-probability space, and gradient descent converges to the global optimum $\pi^*_{\text{CPO}}$. Since $\pi^*_{\text{CPO}} \notin \mathcal{U}$, the optimization avoids $\mathcal{U}$ entirely. $\square$

## D.7. Derivation of CPO gradients

Derivation of CPO gradients, i.e., Eq. 28.

*Proof.* Let $z = \beta(\delta_{\pi_\theta} - \delta_{\pi_{\text{ref}}}) - \tilde{\gamma}_{\text{ref}}$. Then:

$$\mathcal{L}_{\text{CPO}} = -\mathbb{E}[\log \sigma(z)] \tag{86}$$

Taking the gradient with respect to $\theta$:

$$\nabla_\theta \mathcal{L}_{\text{CPO}} = -\mathbb{E}\left[\frac{\sigma'(z)}{\sigma(z)} \cdot \nabla_\theta z\right] \tag{87}$$

Since $\tilde{\gamma}_{\text{ref}}$ is constant with respect to $\theta$:

$$\nabla_\theta z = \beta \cdot \nabla_\theta(\log \pi_\theta(y_w|x) - \log \pi_\theta(y_l|x)) = \beta \cdot (\nabla_\theta \log \pi_\theta(y_w|x) - \nabla_\theta \log \pi_\theta(y_l|x)) \tag{88}$$

Using the identity $\sigma'(z)/\sigma(z) = 1 - \sigma(z) = \sigma(-z)$:

$$\nabla_\theta \mathcal{L}_{\text{CPO}} = -\mathbb{E}\left[\sigma(-z) \cdot \beta \cdot (\nabla_\theta \log \pi_\theta(y_w|x) - \nabla_\theta \log \pi_\theta(y_l|x))\right] \tag{89}$$

Substituting $-z = -\beta(\delta_{\pi_\theta} - \delta_{\pi_{\text{ref}}}) + \tilde{\gamma}_{\text{ref}} = \beta(\delta_{\pi_{\text{ref}}} - \delta_{\pi_\theta}) + \tilde{\gamma}_{\text{ref}}$, we obtain the stated form. $\square$

## E. E-CPOC: Conservative Explicitly Constrained Preference Optimization

This section provides a complete treatment of E-CPOC (Conservative Explicitly Constrained Preference Optimization), our primary method that requires no knowledge of reward differences. We present the theoretical foundation, algorithm, and guarantees.

### E.1. Motivation and Derivation

The general adaptive margin loss derived in Section K.1 requires reward differences $\Delta r = r(y_w) - r(y_l)$, which are typically unknown. To derive a reward-model-free method with provable guarantees, we exploit a **conservative upper bound** based on worst-case analysis.

**Proposition E.1** (Monotonicity of $\Phi$ in $\Delta r$). *The adaptive margin function $\Phi(\delta_{\pi_{ref}}, \Delta r; \gamma) = \max\{0, \gamma - \delta_{\pi_{ref}} - \Delta r/\beta\}$ is monotone non-increasing in $\Delta r$.*

*Proof.* For fixed $\delta_{\pi_{\text{ref}}}$ and $\gamma$, compute the derivative:

$$\frac{\partial \Phi}{\partial(\Delta r)} = \begin{cases} -1/\beta & \text{if } \gamma - \delta_{\pi_{\text{ref}}} - \Delta r/\beta > 0 \\ 0 & \text{if } \gamma - \delta_{\pi_{\text{ref}}} - \Delta r/\beta \leq 0 \end{cases} \tag{90}$$

Therefore, $\Phi$ is monotone non-increasing in $\Delta r$. $\square$

**Corollary E.2** (Conservative Bound). *Since human preference data satisfies $\Delta r^* > 0$ by the Bradley-Terry model, and $\Phi$ is monotone decreasing in $\Delta r$, the maximum value of $\Phi$ over all valid $\Delta r^* > 0$ is achieved in the limit $\Delta r \to 0^+$:*

$$\Phi_{cons}(\delta_{\pi_{ref}}) := \Phi(\delta_{\pi_{ref}}, 0) = \max\{0, \gamma - \delta_{\pi_{ref}}\} \geq \Phi(\delta_{\pi_{ref}}, \Delta r^*) \quad \forall \Delta r^* > 0 \tag{91}$$

This motivates the conservative adaptive margin function:

$$\Phi_{\text{cons}}(\delta_{\pi_{\text{ref}}}; \gamma, \tau) = \frac{1}{\tau} \log\left(1 + \exp\left(\tau(\gamma - \delta_{\pi_{\text{ref}}})\right)\right) \tag{92}$$

## E.2. E-CPOC Algorithm

---

**Algorithm 2** E-CPOC: Conservative Explicitly Constrained Preference Optimization (No Reward Model)

---

**Require:** Preference dataset $\mathcal{D} = \{(x^{(i)}, y_w^{(i)}, y_l^{(i)})\}_{i=1}^N$
**Require:** Reference policy $\pi_{\text{ref}}$
**Require:** Hyperparameters: $\beta > 0$ (temperature), $\gamma > 0$ (target margin), $\tau > 0$ (smoothness)
**Require:** Learning rate $\eta$
1: Initialize policy parameters $\theta$ (e.g., from $\pi_{\text{ref}}$)
2: **Precompute:** For each $(x, y_w, y_l) \in \mathcal{D}$:
3:     $\delta_{\text{ref}}^{(i)} \leftarrow \log \pi_{\text{ref}}(y_w|x) - \log \pi_{\text{ref}}(y_l|x)$
4:     $\Phi^{(i)} \leftarrow \frac{1}{\tau} \log(1 + \exp(\tau(\gamma - \delta_{\text{ref}}^{(i)})))$    ▷ Conservative margin
5:     $\Psi^{(i)} \leftarrow \beta \Phi^{(i)}$    ▷ Adaptive margin
6: **for** each training iteration **do**
7:     Sample batch $\mathcal{B} \subset \mathcal{D}$
8:     **for** each $(x, y_w, y_l) \in \mathcal{B}$ **do**
9:         $\delta_\theta \leftarrow \log \pi_\theta(y_w|x) - \log \pi_\theta(y_l|x)$
10:        logits $\leftarrow \beta(\delta_\theta - \delta_{\text{ref}}^{(i)}) - \Psi^{(i)}$
11:        $\ell \leftarrow -\log \sigma(\text{logits})$
12:     **end for**
13:     Update: $\theta \leftarrow \theta - \eta \nabla_\theta \frac{1}{|\mathcal{B}|} \sum \ell$
14: **end for**
15: **Return**: $\pi_\theta$

---

## E.3. Theoretical Guarantees

**Theorem E.3** (E-CPOC Absolute Advantage Guarantee). *For any preference pair $(x, y_w, y_l)$ with true reward difference $\Delta r^* > 0$, the E-CPOC optimal policy satisfies:*

$$\delta_{\pi_{cons}^*} = \delta_{\pi_{ref}} + \frac{\Delta r^*}{\beta} + \Phi_{cons}(\delta_{\pi_{ref}}; \gamma, \tau) \tag{93}$$

*(1) Upper bound property:*

$$\delta_{\pi_{cons}^*} \geq \delta_{\pi^*}(\Delta r^*) \tag{94}$$

*(2) Absolute advantage: Choosing*

$$\gamma \geq \gamma_{cons}^* := \max_{(x,y_w,y_l)\in\mathcal{D}} \{-\delta_{\pi_{ref}}(x, y_w, y_l)\} \tag{95}$$

*guarantees $\delta_{\pi_{cons}^*} \geq \gamma > 0$ for all preference pairs in $\mathcal{D}$.*

*(3) Sample-adaptive behavior:*

- *Difficult samples ($\delta_{\pi_{ref}} \ll 0$): $\Phi_{cons}(\delta_{\pi_{ref}}) \approx \gamma - \delta_{\pi_{ref}}$ (large correction)*

- *Easy samples ($\delta_{\pi_{ref}} \gg \gamma$): $\Phi_{cons}(\delta_{\pi_{ref}}) \approx 0$ (minimal correction)*

- *Neutral samples ($\delta_{\pi_{ref}} \approx \gamma$): $\Phi_{cons}(\delta_{\pi_{ref}}) \approx \frac{\log 2}{\tau}$ (smooth transition)*

## E.4. Proof of Theorem E.3

*Proof.* From Theorem 4.12, the optimal policy for constrained RLHF satisfies:

$$\delta_{\pi^*} = \delta_{\pi_{\text{ref}}} + \frac{\Delta r}{\beta} + \Phi(\delta_{\pi_{\text{ref}}}, \Delta r; \gamma, \tau) \tag{96}$$

**Step 1: Upper bound property**

By Proposition E.1 and Corollary E.2, using the conservative margin $\Phi_{\text{cons}}(\delta_{\pi_{\text{ref}}}) := \Phi(\delta_{\pi_{\text{ref}}}, 0)$:

$$\delta_{\pi_{\text{cons}}^*} = \delta_{\pi_{\text{ref}}} + \frac{\Delta r^*}{\beta} + \Phi_{\text{cons}}(\delta_{\pi_{\text{ref}}}) \tag{97}$$

$$= \delta_{\pi_{\text{ref}}} + \frac{\Delta r^*}{\beta} + \Phi(\delta_{\pi_{\text{ref}}}, 0) \tag{98}$$

$$\geq \delta_{\pi_{\text{ref}}} + \frac{\Delta r^*}{\beta} + \Phi(\delta_{\pi_{\text{ref}}}, \Delta r^*) \tag{99}$$

$$= \delta_{\pi^*}(\Delta r^*) \tag{100}$$

This proves property (1).

**Step 2: Absolute advantage guarantee**

To ensure $\delta_{\pi_{\text{cons}}^*} > 0$, we need:

$$\delta_{\pi_{\text{ref}}} + \frac{\Delta r^*}{\beta} + \Phi_{\text{cons}}(\delta_{\pi_{\text{ref}}}) > 0 \tag{101}$$

In the worst case where $\Delta r^* \to 0^+$, this reduces to:

$$\delta_{\pi_{\text{ref}}} + \Phi_{\text{cons}}(\delta_{\pi_{\text{ref}}}) > 0 \tag{102}$$

**Case 1**: If $\gamma - \delta_{\pi_{\text{ref}}} > 0$ (constraint is active), then:

$$\Phi_{\text{cons}}(\delta_{\pi_{\text{ref}}}) = \gamma - \delta_{\pi_{\text{ref}}} \tag{103}$$

Therefore:

$$\delta_{\pi_{\text{cons}}^*} \approx \delta_{\pi_{\text{ref}}} + (\gamma - \delta_{\pi_{\text{ref}}}) = \gamma \tag{104}$$

**Case 2**: If $\gamma - \delta_{\pi_{\text{ref}}} \leq 0$ (constraint is not active), then:

$$\Phi_{\text{cons}}(\delta_{\pi_{\text{ref}}}) = 0 \tag{105}$$

and

$$\delta_{\pi_{\text{cons}}^*} = \delta_{\pi_{\text{ref}}} + \frac{\Delta r^*}{\beta} \tag{106}$$

For this case, we need $\delta_{\pi_{\text{ref}}} \geq 0$ for absolute advantage. This is guaranteed when $\gamma \geq -\delta_{\pi_{\text{ref}}}$ for all pairs.

Combining both cases, choosing $\gamma \geq \gamma_{\text{cons}}^* := \max_{(x, y_w, y_l) \in \mathcal{D}} \{ -\delta_{\pi_{\text{ref}}}(x, y_w, y_l) \}$ ensures:

$$\delta_{\pi_{\text{cons}}^*} \geq \min\{\gamma, \delta_{\pi_{\text{ref}}} + \Delta r^*/\beta\} \geq \min\{\gamma, \gamma\} = \gamma > 0 \tag{107}$$

This proves property (2).

**Step 3: Sample-adaptive behavior**

From the softplus formulation $\Phi_{\text{cons}}(\delta_{\pi_{\text{ref}}}) = \frac{1}{\tau} \log(1 + \exp(\tau(\gamma - \delta_{\pi_{\text{ref}}})))$:

- When $\delta_{\pi_{\text{ref}}} \ll 0$ (difficult samples): $\gamma - \delta_{\pi_{\text{ref}}} \gg 0$, so:

$$\Phi_{\text{cons}}(\delta_{\pi_{\text{ref}}}) \approx \frac{1}{\tau} \cdot \tau(\gamma - \delta_{\pi_{\text{ref}}}) = \gamma - \delta_{\pi_{\text{ref}}} \tag{108}$$

- When $\delta_{\pi_{\text{ref}}} \gg \gamma$ (easy samples): $\gamma - \delta_{\pi_{\text{ref}}} \ll 0$, so:

$$\Phi_{\text{cons}}(\delta_{\pi_{\text{ref}}}) \approx \frac{1}{\tau} \log(1 + \exp(\tau(\gamma - \delta_{\pi_{\text{ref}}}))) \approx \frac{1}{\tau} \exp(\tau(\gamma - \delta_{\pi_{\text{ref}}})) \to 0 \tag{109}$$

- When $\delta_{\pi_{\mathrm{ref}}} \approx \gamma$ (neutral samples): $\gamma - \delta_{\pi_{\mathrm{ref}}} \approx 0$, so:

$$\Phi_{\mathrm{cons}}(\delta_{\pi_{\mathrm{ref}}}) \approx \frac{1}{\tau} \log(1 + 1) = \frac{\log 2}{\tau} \tag{110}$$

This proves property (3). $\square$

## E.5. Properties of E-CPOC

**Proposition E.4** (Properties of $\Phi_{\mathrm{cons}}$). *The conservative adaptive margin function $\Phi_{cons}(\delta_{\pi_{ref}}; \gamma, \tau)$ satisfies:*

1. **Non-negativity**: $\Phi_{cons}(\delta_{\pi_{ref}}) \geq 0$ *for all* $\delta_{\pi_{ref}}$

2. **Monotonicity in** $\delta_{\pi_{ref}}$: $\frac{\partial \Phi_{cons}}{\partial \delta_{\pi_{ref}}} = -\sigma\left(\tau(\gamma - \delta_{\pi_{ref}})\right) < 0$

3. **Boundary behavior**:

$$\lim_{\delta_{\pi_{ref}} \to -\infty} \Phi_{cons} = \gamma - \delta_{\pi_{ref}} \quad \text{(strong compensation)} \tag{111}$$

$$\lim_{\delta_{\pi_{ref}} \to +\infty} \Phi_{cons} = 0 \quad \text{(no compensation needed)} \tag{112}$$

4. **Interpretability**: $\Phi_{cons}$ *measures the* degree of constraint violation, *providing exactly the margin needed to satisfy the constraint in the worst-case scenario.*

*Proof.* Properties (1) and (3) follow from the properties of softplus. For (2), compute:

$$\frac{\partial \Phi_{\mathrm{cons}}}{\partial \delta_{\pi_{\mathrm{ref}}}} = -\frac{1}{\tau} \cdot \frac{\tau \exp(\tau(\gamma - \delta_{\pi_{\mathrm{ref}}}))}{1 + \exp(\tau(\gamma - \delta_{\pi_{\mathrm{ref}}}))} = -\sigma(\tau(\gamma - \delta_{\pi_{\mathrm{ref}}})) \tag{113}$$

Since $\sigma(z) \in (0, 1)$ for all $z$, the derivative is strictly negative. $\square$

## E.6. Gradient Analysis and Sample Weighting

**Proposition E.5** (E-CPOC Gradient). *The gradient of the E-CPOC loss is:*

$$\nabla_\theta \mathcal{L}_{\text{E-CPOC}} = -\mathbb{E}\left[\beta \cdot w(\delta_{\pi_\theta}, \delta_{\pi_{ref}}) \cdot (\nabla_\theta \log \pi_\theta(y_w|x) - \nabla_\theta \log \pi_\theta(y_l|x))\right] \tag{114}$$

*where the weight function is:*

$$w(\delta_{\pi_\theta}, \delta_{\pi_{ref}}) = \sigma\left(\beta(\delta_{\pi_{ref}} - \delta_{\pi_\theta}) + \Psi_{cons}(\delta_{\pi_{ref}})\right) \tag{115}$$

*Proof.* Define $z = \beta(\delta_{\pi_\theta} - \delta_{\pi_{\mathrm{ref}}}) - \Psi_{\mathrm{cons}}(\delta_{\pi_{\mathrm{ref}}})$. Then:

$$\nabla_\theta \mathcal{L}_{\text{E-CPOC}} = -\mathbb{E}\left[\frac{\sigma'(z)}{\sigma(z)} \cdot \beta \cdot (\nabla_\theta \log \pi_\theta(y_w|x) - \nabla_\theta \log \pi_\theta(y_l|x))\right] \tag{116}$$

Using $\sigma'(z)/\sigma(z) = 1 - \sigma(z) = \sigma(-z)$:

$$= -\mathbb{E}\left[\beta \cdot \sigma(-z) \cdot (\nabla_\theta \log \pi_\theta(y_w|x) - \nabla_\theta \log \pi_\theta(y_l|x))\right] \tag{117}$$

Since $-z = -\beta(\delta_{\pi_\theta} - \delta_{\pi_{\mathrm{ref}}}) + \Psi_{\mathrm{cons}}(\delta_{\pi_{\mathrm{ref}}}) = \beta(\delta_{\pi_{\mathrm{ref}}} - \delta_{\pi_\theta}) + \Psi_{\mathrm{cons}}(\delta_{\pi_{\mathrm{ref}}})$, we obtain the stated form. $\square$

**Proposition E.6** (Adaptive Gradient Weighting via $\Phi_{\mathrm{cons}}$). *The E-CPOC weight function $w(\delta_{\pi_\theta}, \delta_{\pi_{ref}})$ implements automatic sample difficulty weighting through the adaptive margin function $\Phi_{cons}$:*

1. **Difficult samples** ($\delta_{\pi_{ref}} \ll \gamma$):

$$\Phi_{cons}(\delta_{\pi_{ref}}) \approx \gamma - \delta_{\pi_{ref}} \gg 0 \tag{118}$$

*The large positive $\Phi_{cons}$ significantly increases the weight, providing stronger gradient signal to push $\delta_{\pi_\theta}$ upward.*

2. **Easy samples** $(\delta_{\pi_{ref}} \gg \gamma)$:

$$\Phi_{cons}(\delta_{\pi_{ref}}) \approx 0 \tag{119}$$

The weight reduces to standard DPO, avoiding unnecessary emphasis on already well-aligned samples.

3. **Neutral samples** $(\delta_{\pi_{ref}} \approx \gamma)$:

$$\Phi_{cons}(\delta_{\pi_{ref}}) \approx \frac{\log 2}{\tau} \tag{120}$$

Provides smooth transition between difficult and easy regimes.

*Proof.* The weight function is:

$$w(\delta_{\pi_\theta}, \delta_{\pi_{ref}}) = \sigma\left(\beta(\delta_{\pi_{ref}} - \delta_{\pi_\theta}) + \beta\Phi_{cons}(\delta_{\pi_{ref}})\right) \tag{121}$$

The term $\Psi_{cons}(\delta_{\pi_{ref}}) = \beta\Phi_{cons}(\delta_{\pi_{ref}})$ acts as an additive boost to the logit. When $\delta_{\pi_{ref}} \ll \gamma$, $\Phi_{cons}$ is large, increasing the sigmoid input and thus the weight. When $\delta_{\pi_{ref}} \gg \gamma$, $\Phi_{cons} \approx 0$, and the weight behaves like standard DPO. The smooth transition follows from the softplus formulation of $\Phi_{cons}$. $\square$

**Remark E.7** (Interpretability of Weighting). *The E-CPOC weighting scheme has a clear interpretation: $\Phi_{cons}(\delta_{\pi_{ref}})$ measures how much the constraint would be violated in the worst-case scenario (when $\Delta r \to 0^+$). When $\delta_{\pi_{ref}}$ is far below $\gamma$ (difficult sample), $\Phi_{cons}$ is large, and the term $+\Psi_{cons}$ in the gradient weight $w = \sigma(\beta(\delta_{\pi_{ref}} - \delta_{\pi_\theta}) + \Psi_{cons})$ automatically increases gradient strength, especially when $\delta_{\pi_\theta}$ is also small. When $\delta_{\pi_{ref}}$ exceeds $\gamma$ (easy sample), $\Phi_{cons} \approx 0$, and E-CPOC behaves like standard DPO. This automatic adaptation emerges naturally from the constrained optimization framework with conservative bounds, rather than being heuristically designed.*

### E.7. Comparison with CPO

**Remark E.8** (E-CPOC vs CPO). *E-CPOC and CPO (Section 4.2) are related but distinct:*

- *Constraint type:*
  - *CPO: Soft constraint (encouraged via margin term)*
  - *E-CPOC: Hard constraint (enforced via Lagrange multipliers)*

- *Margin function:*
  - *CPO: $\tilde{\gamma}_{ref} = \gamma\left(\frac{1}{\pi_{ref}(y_w|x)} + \frac{1}{\pi_{ref}(y_l|x)}\right)$ (constant $\gamma$)*
  - *E-CPOC: $\Psi_{cons} = \beta\Phi_{cons}(\delta_{\pi_{ref}})$ (adaptive $\Phi_{cons}$)*

- *Adaptivity:*
  - *CPO: Constant margin $\gamma$ for all samples*
  - *E-CPOC: Sample-adaptive margin $\Phi_{cons}(\delta_{\pi_{ref}})$ that increases for difficult samples*

- *Theoretical derivation:*
  - *CPO: Augmented reward formulation*
  - *E-CPOC: Constrained optimization + KKT conditions + conservative bound*

- *Guarantee strength:*
  - *CPO: Absolute advantage when $\gamma \geq \gamma^*$*
  - *E-CPOC: Absolute advantage when $\gamma \geq \gamma^*_{cons}$ (same form, but with adaptive correction)*

**Key difference:** *E-CPOC uses an adaptive margin $\Phi_{cons}(\delta_{\pi_{ref}})$ that automatically scales based on sample difficulty, while CPO uses a constant margin $\gamma$. This makes E-CPOC more sample-efficient, focusing optimization effort on difficult pairs while avoiding over-regularization on easy pairs.*

## F. RLHF Behavior Under Assumption Violation

It is instructive to contrast DPO's behavior with RLHF in the same scenario.

When $\delta_{\pi_{\text{ref}}} < -\Delta r^*/\beta$, RLHF optimization (Definition 2.1) converges to $\pi^*$ satisfying:

$$\delta_{\pi^*} = \delta_{\pi_{\text{ref}}} + \frac{\Delta r^*}{\beta} < 0 \tag{122}$$

Thus RLHF also produces a policy preferring $y_l$ over $y_w$. However, this is the *optimal* trade-off between reward maximization and KL regularization RLHF correctly optimizes its stated objective.

The fundamental difference between RLHF and DPO under assumption violation:

- **RLHF**: Transparently optimizes $\mathbb{E}[r] - \beta\,\text{KL}$. If this leads to preference violations, it is because the KL penalty is too strong or the reward model is inaccurate problems that can be diagnosed and addressed by adjusting $\beta$ or improving the reward model.

- **DPO**: Claims to optimize the same objective as RLHF but actually optimizes relative advantage $\delta_{\pi_\theta} - \delta_{\pi_{\text{ref}}}$. When Assumption 3.1 fails, DPO silently optimizes a different objective, with no indication in the loss curve that anything is wrong. The pathology is invisible to standard monitoring.

**The reference policy anchor.** A natural concern is that KL regularization to $\pi_{\text{ref}}$ limits the learned policy's ability to discover novel solutions beyond the reference policy's support—a property sometimes referred to as the "poisonous anchor" effect. We note that this is a fundamental property of the KL-regularized RLHF framework itself (Eq. 2), not specific to CPO. DPO, RDPO, IPO, and all methods derived from the RLHF objective share this constraint by design. CPO's scope is to fix DPO's *conditional equivalence* issue within this framework, not to redesign the RLHF paradigm. Addressing the reference-bound ceiling would require moving beyond KL-regularized RLHF entirely.

That said, CPO's constraint term actually provides *more* flexibility than standard DPO to deviate from $\pi_{\text{ref}}$ on critical pairs. The adaptive margin $\tilde{\gamma}_{\text{ref}}$ (Eq. 25 pushes $\delta_{\pi_\theta}$ beyond what the KL penalty alone would allow, especially for hard pairs where $\pi_{\text{ref}}$ is misaligned. In this sense, CPO partially relaxes the anchor effect precisely where it matters most.

## G. Generalization to Multiple Appearances

**Proposition G.1** (Generalization to Multiple Appearances). *For a response $y$ appearing in multiple preference pairs for prompt $x$, define the aggregated margin coefficient:*

$$c(x,y) = \gamma \sum_{(y_w, y_l) \in \mathcal{P}(x)} p(y_w, y_l|x)(\mathbb{I}(y = y_w) - \mathbb{I}(y = y_l)) \tag{123}$$

*where $\mathcal{P}(x)$ denotes all preference pairs for prompt $x$.*

*Then the optimality condition (e.g., Theorems 4.8) remain valid with $c(x,y)$ replacing the simplified coefficients $\pm\gamma$. Furthermore, all approximation error bounds (Proposition K.2) and convergence guarantees (Proposition I.2) remain unchanged, as they depend only on the stationarity of the objective and the KL regularization strength $\beta$.*

*Proof.* The margin term in the Constrained RLHF objective is linear in preference pairs:

$$\gamma \mathbb{E}_{(x,y_w,y_l) \sim \mathcal{D}}[\delta_\pi] = \mathbb{E}_x\left[\sum_y c(x,y) \log \pi(y|x)\right] \tag{124}$$

The first-order optimality condition remains:

$$\beta \log \frac{\pi^*(y|x)}{\pi_{\text{ref}}(y|x)} = r(x,y) + \frac{c(x,y)}{\pi^*(y|x)} - \beta - \lambda(x) \tag{125}$$

For CPO, E-CPOC, and A-CPO, the key insight is that the margin terms $\tilde{\gamma}_{\text{ref}}$, $\Psi_{\text{cons}}$, and $\tilde{w}_{\text{ref}}$ are precomputed based on $\pi_{\text{ref}}$ and remain stationary during optimization. When a response appears in multiple pairs, the algorithm naturally aggregates the gradients through mini-batch computation:

$$\nabla_\theta \mathcal{L} = \sum_{(x,y_w,y_l)\in\mathcal{D}} \nabla_\theta \ell(x, y_w, y_l) \tag{126}$$

The approximation error bounds depend on $\|\pi^* - \pi_{\text{ref}}\|$, which is controlled by $\beta$ regardless of the preference structure. Specifically, for any response $y$:

$$\left|\frac{1}{\pi^*(y|x)} - \frac{1}{\pi_{\text{ref}}(y|x)}\right| \leq \frac{1}{\min\{\pi^*(y|x), \pi_{\text{ref}}(y|x)\}^2} \cdot |\pi^*(y|x) - \pi_{\text{ref}}(y|x)| \tag{127}$$

This bound is independent of how many times $y$ appears in preference pairs. Therefore, all theoretical guarantees (approximation accuracy, convergence, absolute advantage) transfer to the general case with multiple appearances. □

# H. Approximation Accuracy

## H.1. Approximation Accuracy

The CPO loss (Equation (25)) is derived by approximating the optimal policy probabilities in the margin term with the reference policy probabilities (Equation (24)). We now provide an explicit error bound for this approximation under the following mild regularity assumption.

**Assumption H.1** (CPO Approximation Regularity). *For the CPO approximation (Equation (24)), let $q_0 := p_{\min}\, e^{-2R_{\max}/\beta}$:*

 (i) *(**Bounded reference support.**) There exists $p_{\min} > 0$ such that $\pi_{ref}(y|x) \geq p_{\min}$ for all $(x, y_w, y_l) \in \mathcal{D}$ and $y \in \{y_w, y_l\}$.*

 (ii) *(**Bounded rewards.**) The reward function satisfies $\sup_{(x,y)} |r(x,y)| \leq R_{\max}$ for some $R_{\max} > 0$.*

 (iii) *(**Moderate constraint strength.**) When $\gamma > 0$, the constraint strength satisfies $\gamma \leq \beta\, q_0/(2e) = \beta\, p_{\min}\, e^{-2R_{\max}/\beta}/(2e)$. This ensures the preference constraint does not dominate the KL regularization, so that the constrained optimal policy remains close to the unconstrained one.*

*Condition (i) ensures the reference policy has well-defined support on all preference data. This is naturally satisfied for autoregressive language models, where $\pi_{ref}(y|x) > 0$ for all sequences $y$ in the vocabulary, and $p_{\min}$ can be taken as the minimum over the finite dataset $\mathcal{D}$. Condition (ii) is standard in RLHF theory (Ziegler et al., 2019; Gao et al., 2023); in practice, reward models output bounded scores and clipping is commonly applied. Condition (iii) is mild: for large $\beta$ (strong KL regularization), $q_0 \to p_{\min}$ and condition (iii) reduces to $\gamma \leq \beta\, p_{\min}/(2e)$; in practice, $\gamma$ is a small tuning parameter (typically $\gamma \ll \beta$) that controls the strength of the preference alignment constraint relative to the KL penalty. The quantity $q_0 = p_{\min}\, e^{-2R_{\max}/\beta}$ is the policy probability lower bound in the unconstrained ($\gamma = 0$) case.*

**Proposition H.2** (CPO Approximation Error Bound). *Under Assumption H.1, let $\pi^*$ denote the optimal policy of the Constrained RLHF objective (Definition 4.7), let $q_0 = p_{\min}\, e^{-2R_{\max}/\beta}$, and define the effective reward bound*

$$\tilde{R}_{\max} := R_{\max} + \frac{\gamma}{q_0} = R_{\max} + \frac{\gamma\, e^{2R_{\max}/\beta}}{p_{\min}}. \tag{128}$$

*Then the CPO margin approximation error (Equation (24)) satisfies, for each preference pair $(x, y_w, y_l) \in \mathcal{D}$:*

$$|\tilde{\gamma}^*(x, y_w, y_l) - \tilde{\gamma}_{\text{ref}}(x, y_w, y_l)| = O\left(\frac{\gamma\sqrt{\tilde{R}_{\max}/\beta}}{q_0^2}\right). \tag{129}$$

*When $\gamma = 0$ (unconstrained RLHF/DPO baseline), this reduces to $\tilde{R}_{\max} = R_{\max}$, $q_0$ coincides with the optimal policy lower bound, and only conditions (i)–(ii) are needed. When $\gamma > 0$, condition (iii) is additionally required. In both cases, the approximation error vanishes as $\beta \to \infty$ (since $\tilde{R}_{\max}/\beta \to 0$ for fixed $\gamma$), confirming that strong KL regularization ensures accurate approximation.*

*Proof.* The approximation error decomposes as:

$$|\tilde{\gamma}^* - \tilde{\gamma}_{\text{ref}}| = \gamma \left| \left( \frac{1}{\pi^*(y_w|x)} - \frac{1}{\pi_{\text{ref}}(y_w|x)} \right) + \left( \frac{1}{\pi^*(y_l|x)} - \frac{1}{\pi_{\text{ref}}(y_l|x)} \right) \right| \leq \gamma \sum_{y \in \{y_w, y_l\}} \left| \frac{1}{\pi^*(y|x)} - \frac{1}{\pi_{\text{ref}}(y|x)} \right|. \quad (130)$$

We bound each term in three steps. Throughout, $\pi^*$ denotes the optimal policy of the *Constrained* RLHF objective (Definition 4.7), which satisfies the first-order condition (Equation (18)) rather than the closed-form solution of unconstrained RLHF. Let $q_0 = p_{\min} e^{-2R_{\max}/\beta}$ denote the unconstrained ($\gamma = 0$) policy lower bound.

**Step 1: Lower bound on $\pi^*(y|x)$.**

*Case $\gamma = 0$ (unconstrained RLHF).* The optimal policy has the closed form $\pi^*(y|x) = \pi_{\text{ref}}(y|x) \exp(r(x,y)/\beta)/Z(x)$ (Equation (3)), where $Z(x) \leq \exp(R_{\max}/\beta)$. Combined with Assumption H.1(i), $\pi^*(y|x) \geq q_0$.

*Case $\gamma > 0$ (constrained RLHF).* We establish the lower bound via a self-consistency (fixed-point) argument. From the FOC (Equation (18)), the normalization constraint $\sum_y \pi^*(y|x) = 1$ implies that $\pi^*$ is a fixed point of the mapping

$$T(\pi)(y) = \frac{\pi_{\text{ref}}(y|x) \exp\big((r(x,y) + c(x,y)/\pi(y|x))/\beta\big)}{\sum_{y'} \pi_{\text{ref}}(y'|x) \exp\big((r(x,y') + c(x,y')/\pi(y'|x))/\beta\big)}, \quad (131)$$

where $|c(x,y)| \leq \gamma$ (under the single-pair convention, Remark D.1). We show that the set $S = \{\pi : \pi(y|x) \geq q_0/e \ \forall y, \ \sum_y \pi(y|x) = 1\}$ is mapped into itself by $T$, i.e., $T(S) \subseteq S$.

Take any $\pi \in S$, so $\pi(y|x) \geq q_0/e$ for all $y$. For any response $y$:

- The numerator of $T(\pi)(y)$ satisfies $\pi_{\text{ref}}(y|x) \exp\big((r(y) + c(y)/\pi(y))/\beta\big) \geq p_{\min} \cdot \exp\big((-R_{\max} - \gamma e/q_0)/\beta\big)$, since $r(y) \geq -R_{\max}$ and $c(y)/\pi(y) \geq -\gamma/\pi(y) \geq -\gamma e/q_0$.

- The denominator satisfies $\sum_{y'} \pi_{\text{ref}}(y'|x) \exp\big((r(y') + c(y')/\pi(y'))/\beta\big) \leq \exp\big((R_{\max} + \gamma e/q_0)/\beta\big)$, since $r(y') \leq R_{\max}$ and $c(y')/\pi(y') \leq \gamma/\pi(y') \leq \gamma e/q_0$.

Therefore:

$$T(\pi)(y) \geq p_{\min} \cdot \exp\left( \frac{-2(R_{\max} + \gamma e/q_0)}{\beta} \right) = p_{\min} \cdot \exp(-2\tilde{R}'_{\max}/\beta), \quad (132)$$

where $\tilde{R}'_{\max} = R_{\max} + \gamma e/q_0$. Under Assumption H.1(iii), $\gamma \leq \beta q_0/(2e)$, so $\gamma e/q_0 \leq \beta/2$ and:

$$T(\pi)(y) \geq p_{\min} \cdot e^{-2R_{\max}/\beta} \cdot e^{-2\gamma e/(\beta q_0)} \geq q_0 \cdot e^{-1} = q_0/e. \quad (133)$$

Thus $T(\pi) \in S$. Since $\pi^* = T(\pi^*)$ and the Constrained RLHF objective is strictly concave (guaranteeing a unique optimum), the optimal policy must lie in $S$:

$$\pi^*(y|x) \geq q_0/e = p_{\min} e^{-(2R_{\max}/\beta+1)} =: q_{\min}. \quad (134)$$

**Step 2: KL divergence bound.**

*Case $\gamma = 0$.* Since $\pi^*$ maximizes the RLHF objective, comparing with $\pi_{\text{ref}}$ (which has KL $= 0$) yields $\beta \, \text{KL}(\pi^*(\cdot|x) \| \pi_{\text{ref}}(\cdot|x)) \leq 2R_{\max}$.

*Case $\gamma > 0$.* Since $\pi^*$ maximizes the Constrained RLHF objective (Definition 4.7), evaluating at $\pi_{\text{ref}}$ (for which KL $= 0$):

$$\mathbb{E}_{\pi^*}[r] - \beta \, \text{KL}(\pi^* \| \pi_{\text{ref}}) + \gamma \mathbb{E}[\delta_{\pi^*}] \geq \mathbb{E}_{\pi_{\text{ref}}}[r] + \gamma \mathbb{E}[\delta_{\pi_{\text{ref}}}]. \quad (135)$$

Rearranging: $\beta \, \text{KL}(\pi^* \| \pi_{\text{ref}}) \leq (\mathbb{E}_{\pi^*}[r] - \mathbb{E}_{\pi_{\text{ref}}}[r]) + \gamma(\mathbb{E}[\delta_{\pi^*}] - \mathbb{E}[\delta_{\pi_{\text{ref}}}])$. The reward difference is at most $2R_{\max}$. For the $\delta$ term, using $\pi^*(y|x) \geq q_{\min} = q_0/e$ from Step 1 and $\pi_{\text{ref}}(y|x) \geq p_{\min}$:

$$|\delta_{\pi^*}| \leq 2\log(1/q_{\min}), \quad |\delta_{\pi_{\text{ref}}}| \leq 2\log(1/p_{\min}). \quad (136)$$

Since $\log(1/q_{\min}) = 2R_{\max}/\beta + 1 + \log(1/p_{\min})$, and using $\log(1/p) \leq 1/p$ for all $p > 0$:

$$\gamma|\delta_{\pi^*} - \delta_{\pi_{\text{ref}}}| \leq \gamma(|\delta_{\pi^*}| + |\delta_{\pi_{\text{ref}}}|) \leq \gamma\left(\frac{4R_{\max}}{\beta} + 2 + \frac{4}{p_{\min}}\right). \tag{137}$$

Under Assumption H.1(iii), $\gamma \leq \beta q_0/(2e) = \beta p_{\min} e^{-2R_{\max}/\beta}/(2e)$. Each summand is $O(\gamma/p_{\min})$: specifically, $\gamma \cdot 4R_{\max}/\beta \leq 4R_{\max}q_0/(2e) \leq 2R_{\max}p_{\min}$; $2\gamma \leq \beta p_{\min}/(e)$; and $4\gamma/p_{\min} \leq 4\beta q_0/(2e \cdot p_{\min}) = 2\beta e^{-2R_{\max}/\beta}/e \leq 2\beta/e$. Thus:

$$\beta\,\mathrm{KL}(\pi^*(\cdot|x)\|\pi_{\text{ref}}(\cdot|x)) \leq 2R_{\max} + O(\gamma/p_{\min}) = O(\tilde{R}_{\max}). \tag{138}$$

In both cases, by Pinsker's inequality:

$$|\pi^*(y|x) - \pi_{\text{ref}}(y|x)| \leq \|\pi^*(\cdot|x) - \pi_{\text{ref}}(\cdot|x)\|_1 \leq \sqrt{2\,\mathrm{KL}(\pi^*\|\pi_{\text{ref}})} = O\left(\sqrt{\tilde{R}_{\max}/\beta}\right). \tag{139}$$

**Step 3: Combining via mean value theorem.** Applying the mean value theorem to $f(p) = 1/p$:

$$\left|\frac{1}{\pi^*(y|x)} - \frac{1}{\pi_{\text{ref}}(y|x)}\right| \leq \frac{|\pi^*(y|x) - \pi_{\text{ref}}(y|x)|}{\min\{\pi^*(y|x), \pi_{\text{ref}}(y|x)\}^2} = O\left(\frac{\sqrt{\tilde{R}_{\max}/\beta}}{q_{\min}^2}\right) = O\left(\frac{\sqrt{\tilde{R}_{\max}/\beta}}{q_0^2}\right), \tag{140}$$

where the last step uses $q_{\min} = q_0/e$, so $q_{\min}^2 = q_0^2/e^2 = \Theta(q_0^2)$. Summing over $y \in \{y_w, y_l\}$ and multiplying by $\gamma$ yields the stated bound. When $\gamma = 0$, we recover $\tilde{R}_{\max} = R_{\max}$ and $q_{\min} = q_0$. For fixed $\gamma$, $\tilde{R}_{\max}/\beta \to 0$ as $\beta \to \infty$, confirming the error vanishes under strong KL regularization. □

**Remark H.3** (Breakdown Regime and Self-Correcting Margins). *The $1/\pi$ terms in Eq. 23 amplify approximation error at low probabilities, precisely in the failure mode regime where $\pi_{ref}(y_w|x)$ is small. However, this affects the tightness of CPO's equivalence to constrained RLHF (Proposition H.2), not CPO's own guarantees. CPO's loss (Eq. 25) uses $\tilde{\gamma}_{ref}$ computed entirely from $\pi_{ref}$, forming a well-defined stationary objective. The absolute advantage guarantee (Theorem 3.5) and avoidance of $\mathcal{U}$ (Theorem 4.10) are properties of CPO's own optimal policy—they do not depend on the approximation quality. Moreover, the adaptive margin is self-correcting in this regime. When $\pi_{ref}(y_w|x)$ is small, $\tilde{\gamma}_{ref} = \gamma(1/\pi_{ref}(y_w|x) + 1/\pi_{ref}(y_l|x))$ becomes large, applying stronger correction precisely for these hard pairs. The approximation to constrained RLHF is looser, but CPO compensates by pushing harder where it matters most—trading* equivalence tightness *for practical robustness.*

The bound in Equation (129) formalizes the intuition that the approximation error is small when: (1) the KL regularization strength $\beta$ is large (making $\sqrt{\tilde{R}_{\max}/\beta}$ small), (2) the reference policy does not assign extremely low probabilities to responses in $\mathcal{D}$ (making $1/q_0^2$ bounded), (3) the reward function is bounded (making $R_{\max}$ finite), and (4) the constraint strength $\gamma$ is moderate relative to $\beta q_0$ (ensuring $\tilde{R}_{\max}$ remains controlled). We note that E-CPOC (Section 4.5) does not require this approximation and hence does not need Assumption H.1.

**Pairwise ranking vs. probability magnitude.** We clarify that the assumption violation (Assumption 3.1) and the margin approximation (Eq. 24) concern different mathematical quantities. Assumption 3.1 requires $\delta_{\pi^*} = \delta_{\pi_{\text{ref}}} + \Delta r^*/\beta > 0$, a pairwise ranking condition on log-ratios: violation occurs when $\delta_{\pi_{\text{ref}}} < -\Delta r^*/\beta$, flipping the ordering of $y_w$ and $y_l$. KL regularization controls aggregate distributional distance but does not prevent sign flips in pairwise log-ratios. In contrast, the approximation $1/\pi^* \approx 1/\pi_{\text{ref}}$ in Eq. 24 concerns probability magnitudes, which KL does control (Proposition H.2). The two claims are therefore fully compatible: magnitude closeness does not preclude sign flips in log-ratios.

## H.2. Why Not Use $\pi_\theta$ in the Margin?

One might ask: why not use the current policy $\pi_\theta$ in the margin, i.e., $\tilde{\gamma}_\theta = \gamma(1/\pi_\theta(y_w|x) + 1/\pi_\theta(y_l|x))$?

The answer is that this creates a *non-stationary optimization problem*. The gradient would be:

$$\nabla_\theta \mathcal{L} = \nabla_\theta\left[-\log \sigma\left(\beta\delta_{\pi_\theta} - \tilde{\gamma}_\theta\right)\right] \tag{141}$$

This requires computing $\nabla_\theta \tilde{\gamma}_\theta$, which involves second-order derivatives of the policy (since $\tilde{\gamma}_\theta$ contains $1/\pi_\theta$). More importantly, the loss function itself changes as $\theta$ changes during training, violating the standard assumptions for gradient descent convergence.

Using $\pi_{\text{ref}}$ instead makes the margin term *constant with respect to* $\theta$, yielding a stationary objective where standard optimization theory applies. This is not merely a practical convenience but a theoretical necessity for rigorous convergence guarantees.

### H.3. Simplified Constant Margin

If $\pi_{\text{ref}}$ assigns roughly equal probabilities to winners and losers across the dataset, or if we desire the simplest implementation, we can use a constant margin approximation:

$$\tilde{\gamma}_{\text{ref}}(x, y_w, y_l) \approx 2\gamma' \tag{142}$$

where $\gamma'$ can be estimated from the dataset:

$$\gamma' = \frac{\gamma}{2|\mathcal{D}|} \sum_{(x,y_w,y_l)\in\mathcal{D}} \left( \frac{1}{\pi_{\text{ref}}(y_w|x)} + \frac{1}{\pi_{\text{ref}}(y_l|x)} \right) \tag{143}$$

This reduces CPO to its simplest form:

$$\mathcal{L}_{\text{CPO}}(\pi_\theta) = -\mathbb{E}_{(x,y_w,y_l)\sim\mathcal{D}} \left[ \log \sigma \left( \beta(\delta_{\pi_\theta} - \delta_{\pi_{\text{ref}}}) - 2\gamma' \right) \right] \tag{144}$$

This is the most practical form of CPO, requiring only a single scalar hyperparameter $\gamma'$.

The CPO solution elegantly addresses DPO's theoretical flaw through an adaptive margin mechanism. The margin $\tilde{\gamma}_{\text{ref}}(x, y_w, y_l)$ automatically adjusts based on the reference policy's confidence: hard pairs (low $\pi_{\text{ref}}$ probabilities) receive larger margins, while easy pairs (high $\pi_{\text{ref}}$ probabilities) receive smaller margins. This adaptive behavior emerges naturally from the first-order optimality conditions of Constrained RLHF, while the use of $\pi_{\text{ref}}$ ensures a stationary optimization objective with rigorous convergence guarantees.

## I. Convergence Analysis of CPO

**Assumption I.1** (Smoothness and Boundedness). *The policy parameterization $\theta \mapsto \pi_\theta$ satisfies:*

- *(i) (**Lipschitz continuous gradients.**) The gradient of the loss function is Lipschitz continuous: there exists a constant $L > 0$ such that $\|\nabla_\theta \mathcal{L}(\theta_1) - \nabla_\theta \mathcal{L}(\theta_2)\| \le L\|\theta_1 - \theta_2\|$ for all $\theta_1, \theta_2 \in \Theta$.*

- *(ii) (**Bounded parameter domain.**) The parameter space $\Theta \subset \mathbb{R}^d$ is bounded: $\sup_{\theta\in\Theta} \|\theta\| \le B$ for some constant $B > 0$.*

*These are standard regularity conditions in non-convex optimization (Nocedal, 2006). Condition (i) is satisfied when the policy network uses smooth activation functions (e.g., GELU, SiLU) and the loss is composed of smooth operations ($\log$, $\sigma$). Condition (ii) can be enforced via weight clipping or projection, which is common practice in LLM fine-tuning.*

**Proposition I.2** (Convergence of Stationary CPO). *The CPO loss with reference-based margin (Equation (25)) defines a stationary optimization problem. Under Assumption I.1, gradient descent on $\mathcal{L}_{CPO}(\pi_\theta)$ converges to a stationary point.*

*Proof.* Since $\tilde{\gamma}_{\text{ref}}$ does not depend on $\theta$, the loss $\mathcal{L}_{\text{CPO}}(\pi_\theta)$ is a standard differentiable function of $\theta$. Under Assumption I.1, standard results in optimization theory guarantee that gradient descent converges to a stationary point (Nocedal, 2006).

For the convex case: the negative log-likelihood $-\log\sigma(z)$ is convex in $z$. When $\delta_{\pi_\theta}$ is linear in $\theta$ (as in tabular or linear function approximation), the composition yields a convex loss. In this case, any stationary point is a global optimum, and gradient descent achieves a convergence rate of $O(1/T)$ for smooth convex functions. $\square$

Furthermore, if the loss is convex in the policy parameters (as in the linear case or with appropriate regularization), then any stationary point is a global optimum, and gradient descent converges to the global optimum.

The convergence guarantee in Proposition I.2 relies critically on the stationarity of the objective. If we had used $\tilde{\gamma}_\theta$ (depending on $\pi_\theta$) instead, the standard convergence theory would not apply, and we would need to analyze the dynamics of a time-varying optimization problem, which is significantly more complex and may not guarantee convergence to a meaningful solution.

This highlights the importance of our choice to use $\pi_{\text{ref}}$ in the margin term: it is not merely a practical approximation, but a theoretical necessity for obtaining a well-behaved optimization problem with provable convergence guarantees.

**Computational overhead of CPO.** CPO's precomputation overhead is essentially identical to DPO's. Standard DPO already requires a forward pass over the entire dataset to compute and cache $\delta_{\text{ref}}^{(i)} = \log \pi_{\text{ref}}(y_w|x) - \log \pi_{\text{ref}}(y_l|x)$ before training. CPO simply reuses the same forward pass to additionally compute $\tilde{\gamma}_{\text{ref}}^{(i)} = \gamma(1/\pi_{\text{ref}}(y_w|x) + 1/\pi_{\text{ref}}(y_l|x))$—two scalar divisions and one addition per sample, with negligible cost. The storage overhead is one extra scalar per sample. During training, CPO's per-iteration cost is identical to DPO: the only difference is subtracting $\tilde{\gamma}_{\text{ref}}^{(i)}$ from the logits (Algorithm 1, line 11). There is no additional forward/backward pass, no reward model inference, and no architectural change. CPO scales to millions of pairs as easily as DPO.

## J. KKT conditions for Explicitly Constrained RLHF

**Theorem J.1** (Optimal Policy for Explicitly Constrained RLHF). *The optimal solution to the constrained RLHF problem satisfies the first-order optimality condition:*

$$\beta \log \frac{\pi^*(y|x)}{\pi_{ref}(y|x)} = r(x,y) + \frac{\tilde{\mu}^*(x,y)}{\pi^*(y|x)} - \beta - \lambda(x) \tag{145}$$

*where $\tilde{\mu}^*$ are the optimal Lagrange multipliers and $\lambda(x)$ is the Lagrange multiplier for the normalization constraint.*

*For a preference pair $(y_w, y_l)$, this implies:*

$$\beta(\delta_{\pi^*} - \delta_{\pi_{ref}}) = r(y_w) - r(y_l) + \frac{\tilde{\mu}^*(y_w)}{\pi^*(y_w|x)} - \frac{\tilde{\mu}^*(y_l)}{\pi^*(y_l|x)} \tag{146}$$

*Proof.* For a fixed prompt $x$, the Lagrangian can be written as:

$$\mathcal{L}_x = \sum_y \pi(y|x)r(x,y) - \beta \sum_y \pi(y|x) \log \frac{\pi(y|x)}{\pi_{\text{ref}}(y|x)} + \sum_y \tilde{\mu}(x,y) \log \pi(y|x) - \lambda(x) \left( \sum_y \pi(y|x) - 1 \right) \tag{147}$$

where $\tilde{\mu}(x,y)$ is defined in Lemma J.3.

Taking the derivative with respect to $\pi(y|x)$ and setting to zero:

$$\frac{\partial \mathcal{L}_x}{\partial \pi(y|x)} = r(x,y) - \beta \log \frac{\pi(y|x)}{\pi_{\text{ref}}(y|x)} - \beta + \frac{\tilde{\mu}(x,y)}{\pi(y|x)} - \lambda(x) = 0 \tag{148}$$

Rearranging gives Eq. (145). For a preference pair, taking the difference between the conditions for $y_w$ and $y_l$ yields Eq. (146). $\square$

### J.1. Lagrange multipliers

We solve the constrained optimization problem using the method of Lagrange multipliers.

**Theorem J.2** (Lagrangian Formulation). *The Lagrangian for the constrained RLHF problem (Definition 4.11) is:*

$$\mathcal{L}(\pi, \{\mu\}) = \mathbb{E}_{x,y\sim\pi}[r(x,y)] - \beta \, \text{KL}(\pi\|\pi_{ref}) + \sum_{(x,y_w,y_l)\in\mathcal{D}} \mu_{x,y_w,y_l} \left( \delta_\pi(x,y_w,y_l) - \gamma(x,y_w,y_l) \right) \tag{149}$$

*where $\mu_{x,y_w,y_l} \geq 0$ are the Lagrange multipliers (dual variables) for each constraint.*

**Lemma J.3** (Constraint Term Aggregation). *The constraint term in Equation* (149) *can be expressed as an aggregated sum over individual responses. Specifically, the constraint contribution to the Lagrangian is:*

$$\sum_{(x,y_w,y_l)\in\mathcal{D}} \mu_{x,y_w,y_l} \delta_\pi(x,y_w,y_l) = \sum_x \sum_y \tilde{\mu}(x,y) \log \pi(y|x), \tag{150}$$

*where the aggregated multiplier function is:*

$$\tilde{\mu}(x,y) = \sum_{\substack{(y_w,y_l)\in\mathcal{P}(x) \\ y_w=y}} \mu_{x,y_w,y_l} - \sum_{\substack{(y_w,y_l)\in\mathcal{P}(x) \\ y_l=y}} \mu_{x,y_w,y_l} \tag{151}$$

*Moreover, differentiating the Lagrangian with respect to $\pi(y|x)$ yields the term $\tilde{\mu}(x,y)/\pi(y|x)$ in the first-order optimality condition, which acts as an* effective reward augmentation *at the level of the first-order conditions (but not at the level of the objective function itself).*

*Proof.* Expanding $\delta_\pi = \log \pi(y_w|x) - \log \pi(y_l|x)$:

$$\sum_{(x,y_w,y_l)} \mu_{x,y_w,y_l} \delta_\pi = \sum_{(x,y_w,y_l)} \mu_{x,y_w,y_l} (\log \pi(y_w|x) - \log \pi(y_l|x)) \tag{152}$$

$$= \sum_x \sum_{(y_w,y_l)\in\mathcal{P}(x)} \mu_{x,y_w,y_l} \log \pi(y_w|x) - \sum_x \sum_{(y_w,y_l)\in\mathcal{P}(x)} \mu_{x,y_w,y_l} \log \pi(y_l|x) \tag{153}$$

For each $(x,y)$, collecting all terms where $y$ appears:

$$= \sum_x \sum_y \log \pi(y|x) \cdot \tilde{\mu}(x,y) \tag{154}$$

This establishes the aggregation identity. The effective reward augmentation property follows from differentiation: taking $\frac{\partial}{\partial \pi(y|x)}$ of the constraint term $\sum_y \tilde{\mu}(x,y) \log \pi(y|x)$ yields $\tilde{\mu}(x,y)/\pi(y|x)$. Combined with the reward and KL derivatives, this produces the first-order condition (Theorem J.1):

$$r(x,y) + \frac{\tilde{\mu}(x,y)}{\pi(y|x)} - \beta \log \frac{\pi(y|x)}{\pi_{\text{ref}}(y|x)} - \beta - \lambda(x) = 0,$$

in which $\tilde{\mu}(x,y)/\pi(y|x)$ plays the role of an effective reward augmentation. Note that this augmentation interpretation holds at the level of the first-order conditions, not at the objective level, since the constraint term $\sum_y \tilde{\mu}(x,y) \log \pi(y|x)$ differs from $\sum_y \pi(y|x) \cdot \tilde{\mu}(x,y)/\pi(y|x) = \sum_y \tilde{\mu}(x,y)$. $\square$

**Remark J.4** (First-Order Condition Augmentation). *The critical observation is that when we take the derivative of the Lagrangian with respect to $\pi(y|x)$, the term $\tilde{\mu}(x,y)/\pi(y|x)$ appears naturally in the first-order condition. While this term cannot be interpreted as an augmented reward at the objective level (since $\sum_y \tilde{\mu}(x,y) \log \pi(y|x) \neq \mathbb{E}_{y\sim\pi}[\tilde{\mu}(x,y)/\pi(y|x)]$), it provides an exact characterization of the optimal policy structure through the first-order conditions. For a single preference pair, the effective margin contribution $\tilde{\mu}(x,y_w)/\pi^*(y_w|x) - \tilde{\mu}(x,y_l)/\pi^*(y_l|x)$ in the first-order condition characterizes the log-probability ratio of the optimal policy.*

### J.2. KKT Conditions and Adaptive Margin Function

The Karush-Kuhn-Tucker (KKT) conditions characterize the optimal Lagrange multipliers.

**Theorem J.5** (KKT Conditions). *At optimality, the Lagrange multipliers $\{\mu^*\}$ and policy $\pi^*$ satisfy:*

$$\mu^*_{x,y_w,y_l} \geq 0 \quad \textit{(dual feasibility)} \tag{155}$$

$$\delta_{\pi^*}(x,y_w,y_l) \geq \gamma(x,y_w,y_l) \quad \textit{(primal feasibility)} \tag{156}$$

$$\mu^*_{x,y_w,y_l} \cdot (\delta_{\pi^*}(x,y_w,y_l) - \gamma(x,y_w,y_l)) = 0 \quad \textit{(complementary slackness)} \tag{157}$$

The complementary slackness condition Eq. (157) reveals that:

- If constraint is *inactive* ($\delta_{\pi^*} > \gamma$): then $\mu^* = 0$

- If constraint is *active* ($\delta_{\pi^*} = \gamma$): then $\mu^* > 0$ compensates for the constraint

Here, we also leverage the single preference pair for closed-form solution, which can be extended to general case as discussed in deriving CPO.

**Proposition J.6** (Effective Margin Contribution). *Under the notational convention of Remark D.1, for a preference pair* $(x, y_w, y_l)$, *define the* effective margin contribution:

$$M^*_{x,y_w,y_l} := \frac{\tilde{\mu}^*(y_w)}{\pi^*(y_w|x)} - \frac{\tilde{\mu}^*(y_l)}{\pi^*(y_l|x)} = \mu^*_{x,y_w,y_l}\left(\frac{1}{\pi^*(y_w|x)} + \frac{1}{\pi^*(y_l|x)}\right). \tag{158}$$

*Then $M^*$ admits the closed-form expression:*

$$M^*_{x,y_w,y_l} = \beta \max\left\{0, \gamma - \delta_{\pi_{ref}} - \frac{r(y_w) - r(y_l)}{\beta}\right\}. \tag{159}$$

*Proof.* From Theorem J.1 (Eq. (146)), for a preference pair:

$$\beta(\delta_{\pi^*} - \delta_{\pi_{\text{ref}}}) = r(y_w) - r(y_l) + \frac{\tilde{\mu}^*(y_w)}{\pi^*(y_w|x)} - \frac{\tilde{\mu}^*(y_l)}{\pi^*(y_l|x)} = \Delta r + M^*. \tag{160}$$

Therefore:

$$\delta_{\pi^*} = \delta_{\pi_{\text{ref}}} + \frac{\Delta r}{\beta} + \frac{M^*}{\beta}. \tag{161}$$

**Case 1**: If $\delta_{\pi_{\text{ref}}} + \frac{\Delta r}{\beta} \geq \gamma$, the constraint $\delta_{\pi^*} \geq \gamma$ is satisfied without enforcement, so $\mu^* = 0$ by complementary slackness, implying $M^* = 0$.

**Case 2**: If $\delta_{\pi_{\text{ref}}} + \frac{\Delta r}{\beta} < \gamma$, the constraint becomes active: $\delta_{\pi^*} = \gamma$. Solving:

$$\gamma = \delta_{\pi_{\text{ref}}} + \frac{\Delta r}{\beta} + \frac{M^*}{\beta} \implies M^* = \beta\left(\gamma - \delta_{\pi_{\text{ref}}} - \frac{\Delta r}{\beta}\right). \tag{162}$$

Combining both cases yields the stated formula. Note that while the individual Lagrange multiplier $\mu^*$ depends implicitly on $\pi^*$ (through the relation $M^* = \mu^*(1/\pi^*(y_w|x) + 1/\pi^*(y_l|x))$), the effective margin contribution $M^*$ admits a closed form that depends only on the reference policy $\pi_{\text{ref}}$, the reward difference $\Delta r$, and the constraint parameter $\gamma$. $\square$

### J.3. The Unified Adaptive Margin Function

We now introduce the key theoretical contribution: a unified function that elegantly encapsulates all dependence on the reference policy.

**Definition J.7** (Adaptive Margin Function). *The adaptive margin function $\Phi : \mathbb{R} \times \mathbb{R} \to \mathbb{R}$ is defined as the smooth approximation:*

$$\Phi(\delta_{\pi_{ref}}, \Delta r; \gamma, \tau) = \frac{1}{\tau}\log\left(1 + \exp\left(\tau\left(\gamma - \delta_{\pi_{ref}} - \frac{\Delta r}{\beta}\right)\right)\right), \tag{163}$$

*where $\tau > 0$ controls smoothness. This is the softplus function applied to the constraint violation, providing a differentiable relaxation of the hard constraint $\max\{0, \gamma - \delta_{\pi_{ref}} - \Delta r/\beta\}$. As $\tau \to \infty$, $\Phi$ converges to the hard constraint.*

**Remark J.8** (Conservative Version Without Reward Model). *The above formulation requires estimating reward differences $\Delta r = r(y_w) - r(y_l)$ via a reward model. To avoid this requirement while maintaining theoretical guarantees, we can use a* **conservative upper bound** *by setting $\Delta r = 0$ (worst-case assumption):*

$$\Phi_{cons}(\delta_{\pi_{ref}}; \gamma, \tau) = \frac{1}{\tau}\log\left(1 + \exp\left(\tau(\gamma - \delta_{\pi_{ref}})\right)\right) \tag{164}$$

*Since $\Phi$ is monotone non-increasing in $\Delta r$, we have $\Phi_{cons}(\delta_{\pi_{ref}}) \geq \Phi(\delta_{\pi_{ref}}, \Delta r)$ for any $\Delta r > 0$. This conservative version requires no reward model while providing provable alignment guarantees (see Appendix E for details).*

**Proposition J.9** (Properties of $\Phi$). *The adaptive margin function $\Phi$ satisfies:*

1. **Non-negativity**: $\Phi(\delta_{\pi_{ref}}, \Delta r) \geq 0$ *for all $\delta_{\pi_{ref}}, \Delta r$*

2. **Monotonicity in $\delta_{\pi_{ref}}$**: $\frac{\partial \Phi}{\partial \delta_{\pi_{ref}}} = -\sigma\left(\tau\left(\gamma - \delta_{\pi_{ref}} - \frac{\Delta r}{\beta}\right)\right) < 0$

3. **Asymptotic behavior**: *As $\delta_{\pi_{ref}} \to -\infty$, $\Phi(\delta_{\pi_{ref}}, \Delta r) = \gamma - \delta_{\pi_{ref}} - \Delta r/\beta + o(1)$ (strong compensation). As $\delta_{\pi_{ref}} \to +\infty$, $\Phi(\delta_{\pi_{ref}}, \Delta r) \to 0$ (no compensation needed). That is, $\Phi$ asymptotically recovers the hard constraint $\max\{0, \gamma - \delta_{\pi_{ref}} - \Delta r/\beta\}$.*

4. **Interpretability**: $\Phi$ *measures the* degree of constraint violation*, providing exactly the margin needed to satisfy the constraint.*

*Proof.* Properties (1) and (3) follow from the properties of softplus. For (2), compute:

$$\frac{\partial \Phi}{\partial \delta_{\pi_{ref}}} = -\frac{1}{\tau} \cdot \frac{\tau \exp(\tau(\gamma - \delta_{\pi_{ref}} - \Delta r/\beta))}{1 + \exp(\tau(\gamma - \delta_{\pi_{ref}} - \Delta r/\beta))} = -\sigma(\tau(\gamma - \delta_{\pi_{ref}} - \Delta r/\beta)) \tag{165}$$

Since $\sigma(z) \in (0, 1)$ for all $z$, the derivative is strictly negative. $\square$

### J.4. Log-probability Ratio of Optimal Policy

Proof of Theorem 4.12 (Log-probability Ratio of Optimal Policy)

*Proof.* From Proposition J.6, the effective margin contribution under the hard constraint satisfies $M^*_{\text{hard}} = \beta \max\{0, \gamma - \delta_{\pi_{ref}} - \Delta r/\beta\}$. For finite smoothness parameter $\tau > 0$, we replace the hard constraint with its smooth relaxation (Definition J.7):

$$M^*_\tau := \beta\Phi(\delta_{\pi_{ref}}, \Delta r; \gamma, \tau) = \frac{\beta}{\tau} \log\left(1 + \exp\left(\tau\left(\gamma - \delta_{\pi_{ref}} - \frac{\Delta r}{\beta}\right)\right)\right).$$

This smooth relaxation satisfies $M^*_\tau \geq M^*_{\text{hard}}$ for all $\tau > 0$ (since softplus dominates the ReLU function), and $M^*_\tau \to M^*_{\text{hard}}$ as $\tau \to \infty$.

We define the *smoothed Explicitly Constrained RLHF* as the constrained optimization problem in which the hard constraint $\delta_\pi \geq \gamma$ is replaced by the smooth barrier penalty corresponding to $\Phi$. The optimal policy of this smoothed problem satisfies:

$$\delta_{\pi^*} = \delta_{\pi_{ref}} + \frac{\Delta r}{\beta} + \Phi(\delta_{\pi_{ref}}, \Delta r; \gamma, \tau).$$

As $\tau \to \infty$, this recovers the hard-constraint solution $\delta_{\pi^*} = \delta_{\pi_{ref}} + \Delta r/\beta + \max\{0, \gamma - \delta_{\pi_{ref}} - \Delta r/\beta\}$. Throughout this work, when we refer to the "Explicitly Constrained RLHF optimal policy" in the context of the smoothed formulation (parameterized by $\tau$), we mean this smoothed solution. $\square$

## K. Explicitly Constrained Preference Optimization

### K.1. E-CPOC Loss Derivation

**Definition K.1** (General Adaptive Margin Loss). *From the first-order optimality condition (Theorem J.1), the optimal policy satisfies:*

$$\beta(\delta_{\pi^*} - \delta_{\pi_{ref}}) = r(y_w) - r(y_l) + \frac{\tilde{\mu}^*(y_w)}{\pi^*(y_w|x)} - \frac{\tilde{\mu}^*(y_l)}{\pi^*(y_l|x)} \tag{166}$$

*For a single preference pair, we have $\tilde{\mu}^*(y_w) = \mu^*$ and $\tilde{\mu}^*(y_l) = -\mu^*$, so the effective margin contribution (Proposition J.6) is:*

$$\frac{\tilde{\mu}^*(y_w)}{\pi^*(y_w|x)} - \frac{\tilde{\mu}^*(y_l)}{\pi^*(y_l|x)} = \mu^*\left(\frac{1}{\pi^*(y_w|x)} + \frac{1}{\pi^*(y_l|x)}\right) = M^* = \beta\Phi(\delta_{\pi_{ref}}, \Delta r). \tag{167}$$

*A crucial observation is that the effective margin contribution $M^* = \beta\Phi$ admits a closed form that does not depend on $\pi^*$. This is because the individual Lagrange multiplier $\mu^*$ and the inverse probabilities $1/\pi^*$ are coupled through the KKT conditions in such a way that their product $M^*$ depends only on the constraint parameters, reference policy, and reward difference (Proposition J.6). Therefore, from Eq. (166):*

$$\Delta r = \beta(\delta_{\pi^*} - \delta_{\pi_{ref}}) - \beta\Phi(\delta_{\pi_{ref}}, \Delta r). \tag{168}$$

*Applying the Bradley-Terry model and approximating $\pi^*$ with $\pi_\theta$, the general adaptive margin loss takes the form:*

$$\mathcal{L}(\pi_\theta) = -\mathbb{E}_{(x,y_w,y_l)\sim\mathcal{D}}\left[\log\sigma\left(\beta\left(\delta_{\pi_\theta} - \delta_{\pi_{ref}}\right) - \Psi(\delta_{\pi_{ref}}, \Delta r)\right)\right] \tag{169}$$

*where the adaptive margin function is $\Psi(\delta_{\pi_{ref}}, \Delta r) = \beta\Phi(\delta_{\pi_{ref}}, \Delta r)$, with*

$$\Phi(\delta_{\pi_{ref}}, \Delta r; \gamma, \tau) = \frac{1}{\tau}\log\left(1 + \exp\left(\tau\left(\gamma - \delta_{\pi_{ref}} - \frac{\Delta r}{\beta}\right)\right)\right). \tag{170}$$

*Since the true reward difference $\Delta r$ is generally unknown, we derive the **E-CPOC loss** by exploiting the monotonicity of $\Phi$ in $\Delta r$ (Proposition E.1). Since preference data satisfies $\Delta r > 0$ by the Bradley-Terry model, the conservative upper bound $\Phi_{cons}(\delta_{\pi_{ref}}) := \Phi(\delta_{\pi_{ref}}, 0) \geq \Phi(\delta_{\pi_{ref}}, \Delta r^*)$ yields:*

$$\mathcal{L}_{E\text{-}CPOC}(\pi_\theta) = -\mathbb{E}_{(x,y_w,y_l)\sim\mathcal{D}}\left[\log\sigma\left(\beta\left(\delta_{\pi_\theta} - \delta_{\pi_{ref}}\right) - \Psi_{cons}(\delta_{\pi_{ref}})\right)\right] \tag{171}$$

*where $\Psi_{cons}(\delta_{\pi_{ref}}) = \beta\Phi_{cons}(\delta_{\pi_{ref}})$ with $\Phi_{cons}$ from Eq. (164).*

This E-CPOC formulation:

- Requires no knowledge of reward differences, maintaining DPO's key advantage of not needing a reward model

- Provides a provable upper bound: $\delta_{\pi^*_{cons}} \geq \delta_{\pi^*_{EC\text{-}RLHF}}$ (stronger alignment)

- Offers sample-adaptive margins through $\Phi_{cons}(\delta_{\pi_{ref}})$

- Has provable alignment guarantees (see Appendix E)

**Proposition K.2** (Approximation Error Bound). *The approximation error in Eq. (24) can be bounded. For any response $y$, using the mean value theorem on $f(p) = 1/p$:*

$$\left|\frac{1}{\pi^*(y|x)} - \frac{1}{\pi_{ref}(y|x)}\right| \leq \frac{1}{\min\{\pi^*(y|x), \pi_{ref}(y|x)\}^2} \cdot |\pi^*(y|x) - \pi_{ref}(y|x)| \tag{172}$$

*Under the KL constraint in the RLHF objective, when $\beta$ is sufficiently large, $\pi^*$ remains close to $\pi_{ref}$. Specifically, using Pinsker's inequality:*

$$\|\pi^*(\cdot|x) - \pi_{ref}(\cdot|x)\|_1 \leq \sqrt{2\,\mathrm{KL}(\pi^*(\cdot|x)\|\pi_{ref}(\cdot|x))} \tag{173}$$

*Under Assumption H.1 (bounded reference support $p_{\min}$, bounded rewards $R_{\max}$, and moderate constraint strength $\gamma$), the explicit bound from Proposition H.2 applies, giving an approximation error of $O(\gamma\sqrt{\tilde{R}_{\max}/\beta}/q_0^2)$ for each preference pair, where $q_0 = p_{\min}e^{-2R_{\max}/\beta}$ and $\tilde{R}_{\max} = R_{\max} + \gamma/q_0$. This bound is independent of how many times a response appears in preference pairs, so the approximation accuracy guarantees transfer to the general case with multiple appearances.*

*Proof.* The first bound follows from the mean value theorem applied to $f(p) = 1/p$. The second statement follows from Pinsker's inequality, which relates the total variation distance to the KL divergence. The explicit bound follows from Proposition H.2: Steps 1–3 of its proof bound each term $|1/\pi^*(y|x) - 1/\pi_{ref}(y|x)|$ by $O(\sqrt{\tilde{R}_{\max}/\beta}/q_{\min}^2)$ where $q_{\min} = q_0/e$ and $q_0 = p_{\min}e^{-2R_{\max}/\beta}$, independently of the preference pair structure. Since this per-response bound depends only on the individual policy probabilities $\pi^*(y|x)$ and $\pi_{ref}(y|x)$, it holds regardless of how many preference pairs the response participates in. $\square$

## K.2. Algorithm

The E-CPOC training procedure is presented in Algorithm 2 (Appendix E). The key differences from standard DPO are: (1) precomputation of sample-adaptive margins $\Phi_{\text{cons}}^{(i)}$ based on reference policy quality (lines 2-5), and (2) subtracting the scaled adaptive margin $\beta\Phi_{\text{cons}}^{(i)}$ from the logits during training. The precomputation step can be done once before training, making the per-iteration cost identical to CPO and DPO. No reward model is required.

**Remark K.3** (Extension with Reward Model). *When estimated reward differences* $\{\Delta\hat{r}^{(i)}\}_{i=1}^{N}$ *are available from a trained reward model, the conservative margin* $\Phi_{cons}(\delta_{\pi_{ref}}) = \Phi(\delta_{\pi_{ref}}, 0)$ *can be replaced by the tighter margin* $\Phi(\delta_{\pi_{ref}}, \Delta\hat{r}/\beta)$ *in the precomputation step. This reduces the conservatism of the margin while introducing a reward model error of at most* $\epsilon_{\text{RM}}/\beta$ *(Remark L.12).*

# L. Equivalence Between E-CPOC and Explicitly Constrained RLHF

In this section, we establish the theoretical equivalence between E-CPOC and Explicitly Constrained RLHF. The main result is the *conservative upper bound* equivalence (Theorem L.17), which requires only four standard assumptions (Assumptions 4.1–4.4; stated in Sec. 4.1), together with a structural condition for full policy equivalence. The $\ell^2$-$\delta$-proximity assumption admits a verifiable sufficient condition via a bridge from loss suboptimality under a mild non-degeneracy condition (Corollary L.18), with an $N$-*independent* bound.

**Notation.** Throughout this section, we write $\Delta r^* := r^*(x, y_w) - r^*(x, y_l)$ to denote the true latent reward difference, corresponding to $\Delta r$ in the main text. Here $r^*$ is the latent reward function whose existence is implied by the Bradley-Terry model (Assumption 4.1).

## L.1. Detailed Discussion of Assumptions

All assumptions for the E-CPOC equivalence result (Theorem L.17) are stated in Section 4.1. Here we provide extended discussion, the Loss-to-Delta bridge, and the dependency analysis.

The *core assumptions*—Assumptions 4.1–4.3 (Bradley-Terry model, approximate realizability, finite-sample data) and Assumption 4.4 ($\ell^2$-$\delta$-proximity)—are standard or mild conditions inherited from the preference learning and statistical estimation literature. The $\ell^2$-$\delta$-proximity condition (Assumption 4.4) uses the natural mean-square norm that the loss function directly controls, and is strictly weaker than the pointwise ($\ell^\infty$) condition $\max_i |\delta_{\hat{\pi},i} - \delta_i^*| \le \epsilon_{\text{opt}}$. While stated in terms of the unobservable class-optimal policy $\pi_{\text{MLE}}^*$, it admits a *verifiable sufficient condition*: under a mild non-degeneracy condition on preference probabilities (Assumption 4.5), small training loss gap implies $\ell^2$-$\delta$-proximity with an $N$-*independent* bound via the strict convexity of the E-CPOC loss (Proposition L.3). The pointwise bound needed by the error propagation lemma is then recovered through a standard $\ell^2$-to-$\ell^\infty$ norm conversion (Lemma L.1). This bridge reduces the core optimization requirement to a directly monitorable training criterion. Corollary L.18 combines this bridge with the equivalence theorem to provide a self-contained verifiable equivalence guarantee. The *structural condition* (Condition 4.6) is required only for extending pairwise $\delta$-equivalence to full policy equivalence—it is *not* needed for the core pairwise results. Table 1 provides a precise dependency map.

**Extended discussion of Assumption 4.2 (Approximate Realizability).** A sufficient condition for small $\epsilon_{\text{approx}}$ is that the minimax approximation error $\inf_\theta \max_i |\delta_{\pi_\theta,i} - \delta_i^*|$ is small, since the properness of the cross-entropy scoring rule ensures the MLE is at least as good as any fixed $\theta$ in aggregate loss.

**Extended discussion of Assumption 4.4 ($\ell^2$-$\delta$-Proximity).** This condition is *strictly weaker* than the pointwise ($\ell^\infty$) condition $\max_i |\delta_{\hat{\pi},i} - \delta_{\pi_{\text{MLE}}^*,i}| \le \epsilon_{\text{opt}}$: any policy satisfying the latter also satisfies the former with $\epsilon_{\text{opt},2} \le \epsilon_{\text{opt}}$, but the converse does not hold, since $\ell^2$-$\delta$-proximity permits larger deviations on a small number of difficult data points as long as the average error remains controlled. The pointwise bound required by Lemma L.6 is recovered via Lemma L.1.

Under the mild non-degeneracy condition (Assumption 4.5), a small training loss gap $\epsilon_{\text{loss}} := \mathcal{L}(\hat{\pi}) - \inf_\theta \mathcal{L}(\pi_\theta)$ implies $\ell^2$-$\delta$-proximity with an $N$-*independent* bound $\epsilon_{\text{opt},2} = \sqrt{2\epsilon_{\text{loss}}/(\beta^2\kappa_0)}$ (Proposition L.3). This bridge provides a *verifiable sufficient condition*: one monitors $\epsilon_{\text{loss}}$ during training, and the $\ell^2$-$\delta$-proximity bound follows automatically. The derived bound is exact when $\delta_{\pi_\theta}$ is affine in $\theta$ (e.g., tabular or linear function approximation). When $\epsilon_{\text{opt},2} = 0$, it reduces to exact global optimality in $\delta$-space.

**Lemma L.1** (From $\ell^2$ to Pointwise $\delta$-Proximity). *Under $\ell^2$-$\delta$-Proximity (Assumption 4.4) with parameter $\epsilon_{\text{opt},2}$, the*

*pointwise bound holds for all $1 \leq i \leq N$:*

$$|\delta_{\hat{\pi},i} - \delta_{\pi^*_{\mathrm{MLE}},i}| \leq \sqrt{N} \cdot \epsilon_{\mathrm{opt},2} =: \epsilon_{\mathrm{opt}}.$$

*Proof.* By the $\ell^\infty$–$\ell^2$ norm inequality: $\max_{1 \leq i \leq N} |a_i| \leq \sqrt{\sum_{i=1}^N a_i^2} = \sqrt{N \cdot \frac{1}{N} \sum_{i=1}^N a_i^2} \leq \sqrt{N} \cdot \epsilon_{\mathrm{opt},2}$. $\qquad \square$

**Remark L.2** (Stationary-Point Convergence). *A weaker optimization condition—that the returned policy $\hat{\pi}$ is an $\epsilon_{\mathrm{grad}}$-approximate stationary point, i.e., $\|\nabla_\theta \mathcal{L}(\hat{\theta})\| \leq \epsilon_{\mathrm{grad}}$—suffices for the convergence guarantees of CPO and E-CPOC (Proposition I.2). This condition is* provably achievable *by standard gradient descent under Lipschitz-continuous gradients in $O(1/\epsilon_{\mathrm{grad}}^2)$ iterations. While stationary-point convergence does not directly imply $\ell^2$-$\delta$-proximity (Assumption 4.4), it provides the foundation upon which stronger optimization guarantees are built in practice.*

**Extended discussion of Assumption 4.5 (Non-degenerate Preferences).** Recall that $\kappa_0$ (Eq. (16)) is defined with the E-CPOC margin function $g_i(\delta_i) = \beta(\delta_i - \delta_{\mathrm{ref},i}) - \Psi_{\mathrm{cons},i}$. The condition $\kappa_0 > 0$ is automatically satisfied whenever the optimal log-odds $g_i(\delta_i^*)$ are finite, which holds for any smooth parameterization with bounded parameters (Assumption I.1). The quantity $\kappa_0$ measures the curvature of the logistic loss at the optimum and serves as the *bridge constant* that converts a verifiable loss suboptimality condition into the core $\ell^2$-$\delta$-proximity requirement (Assumption 4.4) via Proposition L.3. It is closely related to the inverse sensitivity constant $L_\sigma^{-1}$ appearing in Lemma L.6(iii); the condition $\kappa_0 > 0$ is consistent with the requirement that preference probabilities be bounded away from 0 and 1.

**Proposition L.3** (Loss Suboptimality Implies $\ell^2$-$\delta$-Proximity). *Consider the E-CPOC loss $\mathcal{L}(\pi_\theta) = -\frac{1}{N} \sum_{i=1}^N \log \sigma(g_i(\delta_{\pi_\theta,i}))$, where $g_i(\delta_i) = \beta(\delta_i - \delta_{\mathrm{ref},i}) - \Psi_{\mathrm{cons},i}$ is affine in $\delta_i$. Suppose Assumption 4.5 holds ($\kappa_0 > 0$, cf. Eq. (16)) and the returned policy $\hat{\pi}$ achieves loss gap $\epsilon_{\mathrm{loss}} := \mathcal{L}(\hat{\pi}) - \inf_\theta \mathcal{L}(\pi_\theta)$. Then:*

(a) *(**Convex case.**) When $\delta_{\pi_\theta}$ is affine in $\theta$, for $\epsilon_{\mathrm{loss}}$ below a data-dependent threshold (ensuring the curvature lower bound holds in a neighborhood of the optimum; see proof for the explicit self-consistency condition), Assumption 4.4 ($\ell^2$-$\delta$-proximity) holds with:*

$$\epsilon_{\mathrm{opt},2} = \sqrt{\frac{2\epsilon_{\mathrm{loss}}}{\beta^2 \kappa_0}}. \tag{174}$$

*Note that this bound is* independent of *the dataset size $N$. The corresponding pointwise bound via Lemma L.1 is $\epsilon_{\mathrm{opt}} = \sqrt{N}\,\epsilon_{\mathrm{opt},2} = \sqrt{2N\epsilon_{\mathrm{loss}}/(\beta^2\kappa_0)}$.*

(b) *(**General case.**) For general smooth parameterizations $\theta \mapsto \delta_{\pi_\theta}$, the bound (174) holds to leading order for sufficiently small $\epsilon_{\mathrm{loss}}$, with the correction being $O(\epsilon_{\mathrm{loss}})$.*

*This proposition establishes that loss suboptimality, together with Assumption 4.5 (non-degenerate preferences), implies $\ell^2$-$\delta$-proximity (Assumption 4.4) with an $N$-independent bound. The $\sqrt{N}$ factor that appears in the pointwise bound arises solely from the standard $\ell^2$-to-$\ell^\infty$ norm conversion (Lemma L.1), not from the bridge itself. In practice, one verifies the loss gap $\epsilon_{\mathrm{loss}}$ during training and uses this bridge to obtain the $\ell^2$-$\delta$-proximity guarantee required by the equivalence theorem (Theorem L.17).*

*Proof.* **Part (a): Convex case.** When $\delta_{\pi_\theta} = A\theta + b$ is affine in $\theta$, the chain rule gives $\nabla_\theta \mathcal{L} = A^T \nabla_\delta \mathcal{L}$. Since $\theta^*$ minimizes $\mathcal{L}$ over $\Theta$, the variational first-order optimality condition yields, for every feasible $\hat{\theta} \in \Theta$:

$$\langle \nabla_\delta \mathcal{L}(\delta^*), \hat{\delta} - \delta^* \rangle = \langle A^T \nabla_\delta \mathcal{L}(\delta^*), \hat{\theta} - \theta^* \rangle = \langle \nabla_\theta \mathcal{L}(\theta^*), \hat{\theta} - \theta^* \rangle \geq 0, \tag{175}$$

with equality when $\theta^*$ is an interior point of $\Theta$ (e.g., unconstrained tabular or linear parameterizations with $\Theta = \mathbb{R}^d$).

By Taylor's theorem with integral remainder, writing $f_i(\delta_i) = -\log \sigma(g_i(\delta_i))$:

$$\mathcal{L}(\hat{\delta}) - \mathcal{L}(\delta^*) = \langle \nabla_\delta \mathcal{L}(\delta^*), \hat{\delta} - \delta^* \rangle + \frac{1}{N} \sum_{i=1}^N \int_0^1 (1-t) f_i''(\delta_i^* + t(\hat{\delta}_i - \delta_i^*))\, dt \cdot (\hat{\delta}_i - \delta_i^*)^2$$

$$\geq \frac{1}{N} \sum_{i=1}^N \int_0^1 (1-t) f_i''(\delta_i^* + t(\hat{\delta}_i - \delta_i^*))\, dt \cdot (\hat{\delta}_i - \delta_i^*)^2, \tag{176}$$

where we dropped the non-negative first-order term (175) and used the separability of $\mathcal{L}$ in $\delta$-space. Since $f_i''(\delta_i) = \beta^2 \sigma(g_i(\delta_i))(1 - \sigma(g_i(\delta_i))) > 0$ for all $\delta_i$, by the weighted mean value theorem each integral satisfies:

$$\int_0^1 (1-t) \, f_i''(\delta_i^* + t(\hat{\delta}_i - \delta_i^*)) \, dt = \frac{1}{2} f_i''(\tilde{\delta}_i)$$

for some $\tilde{\delta}_i$ on the line segment between $\delta_i^*$ and $\hat{\delta}_i$. Since $\kappa_0 > 0$ and $\sigma(z)(1 - \sigma(z))$ is continuous, there exists $r_0 > 0$ (depending on $\beta$, $\kappa_0$, and the data) such that $f_i''(\delta_i) \geq \beta^2 \kappa_0$ for all $|\delta_i - \delta_i^*| \leq r_0$ and all $1 \leq i \leq N$. The bound is self-consistent whenever $\epsilon_{\text{loss}} \leq \beta^2 \kappa_0 r_0^2 / (2N)$, since this ensures $\max_i |\hat{\delta}_i - \delta_i^*| \leq r_0$, keeping $\tilde{\delta}_i$ within the curvature neighborhood. For such $\epsilon_{\text{loss}}$:

$$\epsilon_{\text{loss}} \geq \mathcal{L}(\hat{\delta}) - \mathcal{L}(\delta^*) \geq \frac{\beta^2 \kappa_0}{2N} \sum_{i=1}^N (\hat{\delta}_i - \delta_i^*)^2.$$

Therefore:

$$\frac{1}{N} \sum_{i=1}^N (\hat{\delta}_i - \delta_i^*)^2 \leq \frac{2\epsilon_{\text{loss}}}{\beta^2 \kappa_0},$$

yielding $\epsilon_{\text{opt},2} = \sqrt{2\epsilon_{\text{loss}}/(\beta^2 \kappa_0)}$, which is independent of $N$. The pointwise bound follows via Lemma L.1: $\max_i |\hat{\delta}_i - \delta_i^*| \leq \sqrt{N} \cdot \epsilon_{\text{opt},2} = \sqrt{2N\epsilon_{\text{loss}}/(\beta^2 \kappa_0)}$.

**Part (b): General case.** For a general smooth parameterization $\theta \mapsto \delta_{\pi_\theta}$, let $J = \partial\delta/\partial\theta|_{\theta^*}$ denote the Jacobian at the optimum. The displacement decomposes as $\hat{\delta} - \delta^* = J(\hat{\theta} - \theta^*) + O(\|\hat{\theta} - \theta^*\|^2)$. By the variational optimality condition, $\langle \nabla_\theta \mathcal{L}(\theta^*), \hat{\theta} - \theta^* \rangle \geq 0$, which gives $\langle \nabla_\delta \mathcal{L}(\delta^*), J(\hat{\theta} - \theta^*) \rangle \geq 0$. The first-order gradient term then satisfies $\langle \nabla_\delta \mathcal{L}(\delta^*), \hat{\delta} - \delta^* \rangle \geq -C\|\hat{\theta} - \theta^*\|^2$ for a constant $C$ depending on $\|\nabla_\delta \mathcal{L}(\delta^*)\|$ and the Lipschitz constant of $\theta \mapsto \delta_{\pi_\theta}$. For sufficiently small $\epsilon_{\text{loss}}$, this higher-order correction is dominated by the quadratic curvature term in (176), and the bound (174) holds to leading order. $\qquad\square$

**Remark L.4** (Interpretation of Proposition L.3). *Three aspects of the bridge merit discussion:*

(a) **Role of** $\kappa_0$. *The logistic curvature* $\kappa_0 = \min_i \sigma(g_i(\delta_i^*))(1 - \sigma(g_i(\delta_i^*)))$ *measures the* least decisive *preference pair at the class-optimal policy. When all preferences are well-separated ($\kappa_0$ bounded away from 0), the loss landscape is strongly curved and small loss gaps imply tight $\ell^2$-$\delta$-proximity. Conversely, when some preferences approach determinism ($\kappa_0 \to 0$), the loss surface flattens and the bound degrades—this is the correct behavior, as the loss becomes insensitive to $\delta$ in that regime. The condition $\kappa_0 > 0$ (Assumption 4.5) is a mild regularity condition that ensures the bridge from loss suboptimality to $\ell^2$-$\delta$-proximity is well-defined.*

(b) $N$-**independence of the** $\ell^2$ **bridge and the role of** $\sqrt{N}$ **in the pointwise bound.** *The $\ell^2$-$\delta$-proximity bridge $\epsilon_{\text{opt},2} = \sqrt{2\epsilon_{\text{loss}}/(\beta^2 \kappa_0)}$ is independent of $N$, reflecting the fact that the loss function is an average over data points and naturally controls the mean-square $\delta$-error. The $\sqrt{N}$ factor appears only when converting from $\ell^2$ to the pointwise ($\ell^\infty$) bound via Lemma L.1: $\epsilon_{\text{opt}} = \sqrt{N} \, \epsilon_{\text{opt},2}$. This conversion is tight in the worst case (all error concentrated on a single preference pair) but conservative in typical settings where errors are spread across data points. The formulation as $\ell^2$-$\delta$-proximity (Assumption 4.4) cleanly separates the $N$-independent bridge from the standard norm conversion, making the source of $\sqrt{N}$ transparent.*

(c) **Verifiability and the bridge role.** *The $\ell^2$-$\delta$-proximity $\epsilon_{\text{opt},2}$ (Assumption 4.4) is stated in terms of the unobservable class-optimal policy $\pi_{\text{MLE}}^*$, whereas the loss gap $\epsilon_{\text{loss}}$ can be estimated during training by monitoring the training loss. This makes loss suboptimality the* natural practical criterion*: one monitors $\epsilon_{\text{loss}}$ during training, and Proposition L.3 bridges to the core $\ell^2$-$\delta$-proximity requirement (Assumption 4.4) under the mild non-degeneracy condition (Assumption 4.5). Corollary L.18 provides the combined verifiable equivalence guarantee.*

**Structural Condition for Full Policy Equivalence.** Condition 4.6 (Connected Comparison Graph, stated in Section 4.1) is required *only* for extending pairwise $\delta$-equivalence to full policy equivalence. It is not needed for the core pairwise equivalence bound (Theorem L.17). The diameter $d$ controls the error accumulation along graph paths when extending pairwise equivalence to individual log-probabilities. In practice, preference datasets with reasonable coverage over the response space naturally satisfy this condition with moderate diameter.

**Remark L.5** (Interpretation of Assumption Dependencies). *Several observations follow from Table 1:*

*(i) The **pairwise $\delta$-equivalence** result for E-CPOC—the core theoretical contribution—requires only four core assumptions: Assumptions 4.1–4.3 (Bradley-Terry model, approximate realizability, finite-sample data) and Assumption 4.4 ($\ell^2$-$\delta$-proximity). These are standard or mild conditions in the statistical learning literature. Notably, the $\ell^2$-$\delta$-proximity condition is the most direct optimization requirement for the equivalence result: it quantifies the quality of the returned policy in the $\delta$-space that governs preference probabilities using the natural mean-square norm, without imposing any global loss landscape condition or pointwise (max-norm) constraints.*

*(ii) The **bridge pathway**: loss suboptimality and Assumption 4.5 (non-degenerate preferences) together provide a verifiable sufficient condition for the core $\ell^2$-$\delta$-proximity requirement via Proposition L.3, with an $N$-independent bound $\epsilon_{\text{opt},2} = \sqrt{2\epsilon_{\text{loss}}/(\beta^2 \kappa_0)}$. Corollary L.18 packages this bridge with the equivalence theorem into a self-contained verifiable guarantee. This is the recommended practical pathway: one monitors the training loss gap $\epsilon_{\text{loss}}$ during training (directly verifiable), and under the mild non-degeneracy condition, the $\ell^2$-$\delta$-proximity bound follows automatically. The derived bound is exact in convex settings (tabular or linear function approximation). For deep neural network policies, the bridge provides a principled justification: if the training loss converges and preferences are non-degenerate, the $\delta$-equivalence bound follows.*

*(iii) Condition 4.6 (connected comparison graph) is needed solely for extending from pairwise log-probability ratios to individual policy probabilities. In applications where the pairwise preference structure is the primary object of interest (e.g., ranking quality, preference alignment verification), this condition can be dropped entirely.*

*(iv) Stationary-point convergence (Remark L.2) is provably achievable and suffices for the convergence guarantees (Proposition I.2). The overall structure cleanly separates concerns: stationary-point convergence for optimization convergence, $\ell^2$-$\delta$-proximity (Assumption 4.4) for equivalence, and the loss suboptimality bridge (Proposition L.3) for practical verifiability.*

## L.2. Error Propagation under Weakened Assumptions

The following lemma characterizes how the approximation errors from the weakened assumptions propagate through the MLE-based equivalence argument for E-CPOC.

**Lemma L.6** (Unified Error Propagation). *Consider the E-CPOC preference model, which takes the form $p_\pi(y_w \succ y_l|x) = \sigma(g(\delta_\pi, x, y_w, y_l))$ where $g$ is a known function that is affine in $\delta_\pi$ with slope $\beta$. Let $\delta_{\text{target}}$ denote the unique log-probability ratio achieving $p_\pi = p^*$ under the Bradley-Terry model (Assumption 4.1). Under Assumptions 4.2–4.3 and Assumption 4.4 ($\ell^2$-$\delta$-proximity):*

*(i) (**Approximate realizability error.**) The MLE optimal policy $\pi^*_{\text{MLE}}$ over the policy class satisfies $|\delta_{\pi^*_{\text{MLE}}}(x_i) - \delta_{\text{target}}(x_i)| \leq \epsilon_{\text{approx}}$ for all $i$.*

*(ii) (**Approximate optimality error.**) Under Assumption 4.4 ($\ell^2$-$\delta$-proximity) and Lemma L.1, the returned policy $\hat{\pi}$ satisfies $|\delta_{\hat{\pi}}(x_i) - \delta_{\pi^*_{\text{MLE}}}(x_i)| \leq \epsilon_{\text{opt}} := \sqrt{N}\,\epsilon_{\text{opt},2}$ for all $i$. In practice, $\epsilon_{\text{opt}}$ is typically obtained via the bridge $\epsilon_{\text{opt}} = \sqrt{N}\,\epsilon_{\text{opt},2} = \sqrt{2N\epsilon_{\text{loss}}/(\beta^2 \kappa_0)}$ from loss suboptimality and Assumption 4.5 (Proposition L.3 combined with Lemma L.1; see also Corollary L.18).*

*(iii) (**Statistical estimation error.**) The population-level MLE target $\delta_{\text{target}}$ is replaced by the finite-sample MLE target $\hat{\delta}_{\text{target}}$, satisfying $|\hat{\delta}_{\text{target}}(x_i) - \delta_{\text{target}}(x_i)| \leq L_\sigma^{-1}\epsilon_{\text{stat}}$, where $L_\sigma^{-1} := 1/(\beta \cdot \inf_i \sigma(\Delta r_i^*)(1 - \sigma(\Delta r_i^*)))$ is the inverse sensitivity constant incorporating both the sigmoid curvature and the model slope $\beta$.*

*Combining by triangle inequality:*

$$|\delta_{\hat{\pi}}(x_i) - \delta_{\text{target}}(x_i)| \leq \epsilon_{\text{approx}} + \epsilon_{\text{opt}} + L_\sigma^{-1}\epsilon_{\text{stat}}, \quad \forall i.$$

*When $\epsilon_{\text{approx}} = \epsilon_{\text{opt}} = \epsilon_{\text{stat}} = 0$, the MLE achieves $\delta_{\hat{\pi}} = \delta_{\text{target}}$ exactly.*

*Proof.* **Part (i).** This is the direct content of Assumption 4.2: the population MLE satisfies $|\delta_{\pi^*_{\text{MLE}}}(x_i) - \delta_{\text{target}}(x_i)| \leq \epsilon_{\text{approx}}$ for all $i$.

**Part (ii).** By Assumption 4.4 ($\ell^2$-$\delta$-proximity) and Lemma L.1, the returned policy $\hat{\pi}$ satisfies $|\delta_{\hat{\pi}}(x_i) - \delta_{\pi^*_{\mathrm{MLE}}}(x_i)| \leq \epsilon_{\mathrm{opt}} := \sqrt{N}\,\epsilon_{\mathrm{opt},2}$ for all $i$. When $\epsilon_{\mathrm{opt},2}$ is obtained via the bridge (Proposition L.3), one uses loss suboptimality with gap $\epsilon_{\mathrm{loss}}$ and Assumption 4.5 ($\kappa_0 > 0$) to obtain $\epsilon_{\mathrm{opt},2} = \sqrt{2\epsilon_{\mathrm{loss}}/(\beta^2\kappa_0)}$ and hence $\epsilon_{\mathrm{opt}} = \sqrt{2N\epsilon_{\mathrm{loss}}/(\beta^2\kappa_0)}$.

**Part (iii).** Under finite data (Assumption 4.3), the empirical preference frequency $\hat{p}_N$ satisfies $|\hat{p}_N(y_w \succ y_l|x) - p^*(y_w \succ y_l|x)| \leq \epsilon_{\mathrm{stat}}$. The finite-sample MLE target satisfies $g(\hat{\delta}_{\mathrm{target},i}) = \mathrm{logit}(\hat{p}_{N,i})$. By the mean value theorem applied to $\mathrm{logit}(p) = \log(p/(1-p))$, we have $|\mathrm{logit}(\hat{p}_{N,i}) - \mathrm{logit}(p^*_i)| \leq \epsilon_{\mathrm{stat}}/(\tilde{p}_i(1-\tilde{p}_i))$ for some $\tilde{p}_i$ between $\hat{p}_{N,i}$ and $p^*_i$. Since $g$ is affine in $\delta$ with slope $\beta$, $|\hat{\delta}_{\mathrm{target},i} - \delta_{\mathrm{target},i}| = \frac{1}{\beta}|\mathrm{logit}(\hat{p}_{N,i}) - \mathrm{logit}(p^*_i)| \leq \frac{\epsilon_{\mathrm{stat}}}{\beta\cdot\tilde{p}_i(1-\tilde{p}_i)}$. Bounding $\tilde{p}_i(1-\tilde{p}_i) \geq \inf_i \sigma(\Delta r^*_i)(1-\sigma(\Delta r^*_i))$ for $N$ sufficiently large, we obtain $|\hat{\delta}_{\mathrm{target},i} - \delta_{\mathrm{target},i}| \leq L_\sigma^{-1}\epsilon_{\mathrm{stat}}$ with $L_\sigma^{-1} := 1/(\beta\cdot\inf_i \sigma(\Delta r^*_i)(1-\sigma(\Delta r^*_i)))$.

The final bound follows by the triangle inequality: $|\delta_{\hat{\pi},i} - \delta_{\mathrm{target},i}| \leq |\delta_{\hat{\pi},i} - \delta_{\pi^*_{\mathrm{MLE}},i}| + |\delta_{\pi^*_{\mathrm{MLE}},i} - \hat{\delta}_{\mathrm{target},i}| + |\hat{\delta}_{\mathrm{target},i} - \delta_{\mathrm{target},i}| \leq \epsilon_{\mathrm{opt}} + \epsilon_{\mathrm{approx}} + L_\sigma^{-1}\epsilon_{\mathrm{stat}}$. □

## L.3. E-CPOC and Explicitly Constrained RLHF Equivalence

We establish the equivalence between E-CPOC and Explicitly Constrained RLHF. E-CPOC uses hard constraints enforced via KKT conditions and does not require a reward model.

**Lemma L.7** (KKT Conditions for EC-RLHF). *At optimality, the Lagrange multipliers $\{\mu^*\}$ and policy $\pi^*$ for Explicitly Constrained RLHF satisfy:*

1. ***Dual feasibility**: $\mu^*_{x,y_w,y_l} \geq 0$*

2. ***Primal feasibility**: $\delta_{\pi^*}(x, y_w, y_l) \geq \gamma$*

3. ***Complementary slackness**: $\mu^*_{x,y_w,y_l} \cdot (\delta_{\pi^*}(x, y_w, y_l) - \gamma) = 0$*

*The complementary slackness condition reveals:*

- *If constraint is inactive ($\delta_{\pi^*} > \gamma$): then $\mu^* = 0$*

- *If constraint is active ($\delta_{\pi^*} = \gamma$): then $\mu^* > 0$ compensates for the constraint*

*Proof.* Standard KKT conditions for inequality-constrained optimization. □

**Proposition L.8** (Effective Margin Contribution for EC-RLHF). *For a preference pair $(x, y_w, y_l)$, define the effective margin contribution $M^* := \mu^* \left( \frac{1}{\pi^*(y_w|x)} + \frac{1}{\pi^*(y_l|x)} \right)$, where $\mu^*$ is the optimal Lagrange multiplier. Then:*

$$M^*_{x,y_w,y_l} = \beta \max\left\{ 0, \gamma - \delta_{\pi_{ref}} - \frac{\Delta r^*}{\beta} \right\}.$$

*Proof.* From Theorem J.1, the first-order condition for a single preference pair gives:

$$\beta(\delta_{\pi^*} - \delta_{\pi_{\mathrm{ref}}}) = \Delta r^* + M^*.$$

**Case 1**: If $\delta_{\pi_{\mathrm{ref}}} + \frac{\Delta r^*}{\beta} \geq \gamma$, the constraint $\delta_{\pi^*} \geq \gamma$ is satisfied without enforcement, so $\mu^* = 0$ by complementary slackness, implying $M^* = 0$.

**Case 2**: If $\delta_{\pi_{\mathrm{ref}}} + \frac{\Delta r^*}{\beta} < \gamma$, the constraint becomes active: $\delta_{\pi^*} = \gamma$. Solving:

$$\beta(\gamma - \delta_{\pi_{\mathrm{ref}}}) = \Delta r^* + M^* \implies M^* = \beta\left( \gamma - \delta_{\pi_{\mathrm{ref}}} - \frac{\Delta r^*}{\beta} \right).$$

Combining both cases yields the result. Note that while $\mu^*$ individually depends on $\pi^*$, the combined quantity $M^*$ admits a closed form independent of $\pi^*$ (cf. Proposition J.6). □

**Definition L.9** (Adaptive Margin Function). *Define the smooth approximation to the hard constraint:*

$$\Phi(\delta_{\pi_{ref}}, \Delta r^*; \gamma, \tau) = \frac{1}{\tau} \log\left(1 + \exp\left(\tau\left(\gamma - \delta_{\pi_{ref}} - \frac{\Delta r^*}{\beta}\right)\right)\right)$$

*This is the* softplus function *applied to the constraint violation, providing a differentiable relaxation of:*

$$\max\left\{0, \gamma - \delta_{\pi_{ref}} - \frac{\Delta r^*}{\beta}\right\}$$

*As $\tau \to \infty$, $\Phi$ converges to the hard constraint.*

**Proposition L.10** (Properties of $\Phi$). *The adaptive margin function $\Phi$ satisfies the following properties (cf. Proposition J.9 for the corresponding results in the main paper notation):*

1. **Non-negativity**: $\Phi(\delta_{\pi_{ref}}, \Delta r^*) \geq 0$ *for all inputs*

2. **Monotonicity in $\delta_{\pi_{ref}}$**:
$$\frac{\partial \Phi}{\partial \delta_{\pi_{ref}}} = -\sigma\left(\tau\left(\gamma - \delta_{\pi_{ref}} - \frac{\Delta r^*}{\beta}\right)\right) < 0$$

3. **Monotonicity in $\Delta r^*$**:
$$\frac{\partial \Phi}{\partial(\Delta r^*)} = -\frac{1}{\beta}\sigma\left(\tau\left(\gamma - \delta_{\pi_{ref}} - \frac{\Delta r^*}{\beta}\right)\right) < 0$$

4. **Asymptotic behavior**:

$$\Phi(\delta_{\pi_{ref}}, \Delta r^*) = \gamma - \delta_{\pi_{ref}} - \frac{\Delta r^*}{\beta} + O(e^{-\tau(\gamma - \delta_{\pi_{ref}} - \Delta r^*/\beta)}) \quad \text{as } \delta_{\pi_{ref}} \to -\infty \tag{177}$$

$$\Phi(\delta_{\pi_{ref}}, \Delta r^*) = O(e^{\tau(\gamma - \delta_{\pi_{ref}} - \Delta r^*/\beta)}) \to 0 \quad \text{as } \delta_{\pi_{ref}} \to +\infty \tag{178}$$

*That is, $\Phi$ asymptotically approaches the hard constraint $\max\{0, \gamma - \delta_{\pi_{ref}} - \Delta r^*/\beta\}$ in both limits.*

5. **Interpretability**: $\Phi$ *measures the* degree of constraint violation

*Proof.* Properties (1) and (4) follow from the properties of softplus. For (2) and (3), compute the derivatives using the chain rule and the identity $\frac{d}{dz}\text{softplus}(z) = \sigma(z)$. $\qquad\square$

**Lemma L.11** (Relationship Between $\Phi$ and Effective Margin). *The adaptive margin function $\Phi$ is related to the effective margin contribution by:*
$$M^*_{x,y_w,y_l} = \beta\Phi(\delta_{\pi_{ref}}, \Delta r^*; \gamma, \tau),$$

*where $\Phi$ serves as the smooth relaxation of $M^*/\beta$. In the hard constraint limit ($\tau \to \infty$), this reduces to $M^* = \beta\max\{0, \gamma - \delta_{\pi_{ref}} - \Delta r^*/\beta\}$.*

*Proof.* Direct comparison of Proposition L.8 and Definition L.9: the effective margin under the hard constraint satisfies $M^*_{\text{hard}} = \beta\max\{0, \gamma - \delta_{\pi_{ref}} - \Delta r^*/\beta\}$, which equals $\beta$ times the ReLU function of the constraint violation. The smooth relaxation $M^*_\tau = \beta\Phi(\delta_{\pi_{ref}}, \Delta r^*; \gamma, \tau)$ provides a differentiable approximation satisfying $M^*_\tau \geq M^*_{\text{hard}}$ for all $\tau > 0$, with $M^*_\tau \to M^*_{\text{hard}}$ as $\tau \to \infty$. $\qquad\square$

**Remark L.12** (Extension with Reward Model). *When an approximate reward model $\hat{r}$ is available, one can use the tighter margin function $\Phi(\delta_{\pi_{ref}}, \Delta\hat{r})$ instead of the conservative $\Phi_{cons}(\delta_{\pi_{ref}}) = \Phi(\delta_{\pi_{ref}}, 0)$. By the Lipschitz continuity of $\Phi$ in $\Delta r^*$ (Proposition L.10(3)), the resulting error in the pairwise log-probability ratio due to reward model inaccuracy is bounded by $\epsilon_{\text{RM}}/\beta$, where $\epsilon_{\text{RM}} := \max_{(x,y_w,y_l)} |\Delta\hat{r} - \Delta r^*|$. The $1/\beta$ attenuation implies that stronger KL regularization mitigates reward model inaccuracy. When $\epsilon_{\text{RM}} = 0$, the equivalence with EC-RLHF becomes exact. However, since the conservative version (E-CPOC) already provides provable alignment guarantees without any reward model, we focus on E-CPOC as the primary method.*

**L.4. E-CPOC: Conservative Bound Without Reward Model**

When reward differences are unavailable, we derive a conservative bound based on worst-case analysis.

**Proposition L.13** (Monotonicity of $\Phi$ in $\Delta r^*$). *The adaptive margin function $\Phi(\delta_{\pi_{ref}}, \Delta r^*)$ is monotone non-increasing in $\Delta r^*$:*

$$\frac{\partial \Phi}{\partial(\Delta r^*)} = -\frac{1}{\beta}\sigma\left(\tau\left(\gamma - \delta_{\pi_{ref}} - \frac{\Delta r^*}{\beta}\right)\right) \leq 0$$

*Proof.* Direct computation of the derivative using the chain rule and the fact that $\frac{d}{dz}\text{softplus}(z) = \sigma(z)$. $\qquad\square$

**Corollary L.14** (Conservative Bound). *Since preference data satisfies $\Delta r^* > 0$ by the Bradley-Terry model, and $\Phi$ is monotone decreasing in $\Delta r^*$ (Proposition L.13), the maximum value of $\Phi$ over all valid $\Delta r^* > 0$ is achieved in the limit $\Delta r^* \to 0^+$:*

$$\Phi_{cons}(\delta_{\pi_{ref}}) := \Phi(\delta_{\pi_{ref}}, 0) = \frac{1}{\tau}\log\left(1 + \exp\left(\tau(\gamma - \delta_{\pi_{ref}})\right)\right)$$

*satisfies:*

$$\Phi_{cons}(\delta_{\pi_{ref}}) \geq \Phi(\delta_{\pi_{ref}}, \Delta r^*) \quad \forall \Delta r^* > 0$$

*Proof.* By monotonicity (Proposition L.13), $\Phi$ is maximized when $\Delta r^*$ is minimized. Since $\Delta r^* > 0$ for all preference data, the supremum is achieved at $\Delta r^* \to 0^+$. $\qquad\square$

**Definition L.15** (E-CPOC Loss). *The E-CPOC (Conservative Explicitly Constrained Preference Optimization) loss is:*

$$\mathcal{L}_{E\text{-}CPOC}(\pi; \pi_{ref}) = -\mathbb{E}_{(x,y_w,y_l)\sim\mathcal{D}}\left[\log\sigma\left(\beta(\delta_\pi - \delta_{\pi_{ref}}) - \Psi_{cons}(\delta_{\pi_{ref}})\right)\right]$$

*where:*

$$\Psi_{cons}(\delta_{\pi_{ref}}) = \beta\Phi_{cons}(\delta_{\pi_{ref}}),$$

*with:*

$$\Phi_{cons}(\delta_{\pi_{ref}}; \gamma, \tau) = \frac{1}{\tau}\log\left(1 + \exp\left(\tau(\gamma - \delta_{\pi_{ref}})\right)\right).$$

**Proposition L.16** (Properties of $\Phi_{\text{cons}}$). *The conservative adaptive margin function $\Phi_{cons}(\delta_{\pi_{ref}}; \gamma, \tau)$ satisfies:*

1. **Non-negativity**: $\Phi_{cons}(\delta_{\pi_{ref}}) \geq 0$ *for all* $\delta_{\pi_{ref}}$

2. **Monotonicity in** $\delta_{\pi_{ref}}$: $\frac{\partial\Phi_{cons}}{\partial\delta_{\pi_{ref}}} = -\sigma\left(\tau(\gamma - \delta_{\pi_{ref}})\right) < 0$

3. **Asymptotic behavior**:

$$\Phi_{cons}(\delta_{\pi_{ref}}) = \gamma - \delta_{\pi_{ref}} + O(e^{-\tau(\gamma-\delta_{ref})}) \quad \text{as } \delta_{\pi_{ref}} \to -\infty \text{ (strong compensation)} \tag{179}$$

$$\Phi_{cons}(\delta_{\pi_{ref}}) = O(e^{\tau(\gamma-\delta_{ref})}) \to 0 \quad \text{as } \delta_{\pi_{ref}} \to +\infty \text{ (no compensation)} \tag{180}$$

*That is, $\Phi_{cons}$ asymptotically recovers $\max\{0, \gamma - \delta_{\pi_{ref}}\}$ in both limits.*

4. **Interpretability**: $\Phi_{cons}$ *measures the* degree of constraint violation *in the worst-case scenario*

*Proof.* Properties (1) and (3) follow from the properties of softplus. For (2), compute:

$$\frac{\partial\Phi_{\text{cons}}}{\partial\delta_{\pi_{\text{ref}}}} = -\frac{1}{\tau}\cdot\frac{\tau\exp(\tau(\gamma - \delta_{\pi_{\text{ref}}}))}{1 + \exp(\tau(\gamma - \delta_{\pi_{\text{ref}}}))} = -\sigma(\tau(\gamma - \delta_{\pi_{\text{ref}}}))$$

Since $\sigma(z) \in (0,1)$ for all $z$, the derivative is strictly negative. $\qquad\square$

**Theorem L.17** (E-CPOC Upper Bound Equivalence). *Under Assumptions 4.1–4.3 and Assumption 4.4 ($\ell^2$-$\delta$-proximity with parameter $\epsilon_{opt,2}$), the E-CPOC policy $\hat{\pi}_{cons}$ satisfies (up to the unified error $\epsilon_{approx} + \epsilon_{opt} + L_\sigma^{-1}\epsilon_{stat}$ from Lemma L.6, where $\epsilon_{opt} = \sqrt{N}\,\epsilon_{opt,2}$ via Lemma L.1), without requiring a reward model. In practice, $\epsilon_{opt}$ is obtained from the verifiable loss gap via the bridge $\epsilon_{opt} = \sqrt{N}\,\epsilon_{opt,2} = \sqrt{2N\epsilon_{loss}/(\beta^2\kappa_0)}$ (Proposition L.3 and Lemma L.1, requiring additionally loss suboptimality and Assumption 4.5; see Corollary L.18):*

*(1) Upper Bound Property:*

$$\delta_{\pi_{E\text{-}CPOC}^*} \geq \delta_{\pi_{EC\text{-}RLHF}^*}(\Delta r^*)$$

*for any true reward difference $\Delta r^* > 0$.*

*(2) Worst-Case Margin Structure: The E-CPOC optimal policy decomposes as:*

$$\delta_{\pi_{E\text{-}CPOC}^*} = \delta_{\pi_{EC\text{-}RLHF}^*}(\Delta r^* = 0) + \frac{\Delta r^*}{\beta}$$

*That is, E-CPOC adopts the margin correction of the worst-case EC-RLHF solution (with $\Delta r^* = 0$), supplemented by the true reward difference $\Delta r^*/\beta$. Since $\Delta r^* > 0$, this implies $\delta_{\pi_{E\text{-}CPOC}^*} \geq \delta_{\pi_{EC\text{-}RLHF}^*}(\Delta r^* = 0)$, making E-CPOC strictly more conservative than the worst-case EC-RLHF.*

*(3) Absolute Advantage Guarantee: For any $\gamma > 0$, the exact (non-asymptotic) bound holds:*

$$\delta_{\pi_{E\text{-}CPOC}^*} > \gamma + \frac{\Delta r^*}{\beta} > \gamma > 0$$

*for all preference pairs in $\mathcal{D}$. Note that this guarantee holds for any positive $\gamma$; no minimum threshold is theoretically required. In practice, however, we recommend choosing $\gamma \geq \gamma_{cons}^* := \max_{(x,y_w,y_l)\in\mathcal{D}}\{-\delta_{\pi_{ref}}(x,y_w,y_l)\}$ to ensure the constraint correction is meaningful for the hardest samples (i.e., those where the reference policy most strongly disfavors the preferred response), resulting in the constraint being approximately binding at these samples.*

*(4) Sample-Adaptive Behavior:*

- *Difficult samples ($\delta_{\pi_{ref}} \ll 0$): $\Phi_{cons}(\delta_{\pi_{ref}}) \approx \gamma - \delta_{\pi_{ref}} \gg 0$ (large correction)*

- *Easy samples ($\delta_{\pi_{ref}} \gg \gamma$): $\Phi_{cons}(\delta_{\pi_{ref}}) \approx 0$ (minimal correction)*

- *Neutral samples ($\delta_{\pi_{ref}} \approx \gamma$): $\Phi_{cons}(\delta_{\pi_{ref}}) \approx \frac{\log 2}{\tau}$ (smooth transition)*

Furthermore, for full policy equivalence, under the additional connected comparison graph condition (Condition 4.6), the pairwise $\delta$-equivalence extends to individual policy probabilities: $\hat{\pi}_{cons}(y|x) \approx \pi_{EC\text{-}RLHF,cons}^*(y|x)$ for all responses $y$ appearing in preference pairs for $x$, with the individual log-probability error bounded by $O(d \cdot (\epsilon_{approx} + \epsilon_{opt} + L_\sigma^{-1}\epsilon_{stat}))$, where $d$ is the maximum comparison graph diameter.

*Proof.* **Step 1: Characterization of E-CPOC Optimal Policy**

The E-CPOC loss (Definition L.15) defines a preference model $p_\pi(y_w \succ y_l|x) = \sigma(\beta(\delta_\pi - \delta_{\pi_{ref}}) - \Psi_{cons}(\delta_{\pi_{ref}}))$. By the Bradley-Terry model (Assumption 4.1), the true preference probability is $p^*(y_w \succ y_l|x) = \sigma(\Delta r^*)$. The cross-entropy loss $-\mathbb{E}_{p^*}[\log p_\pi]$ is minimized when $p_\pi = p^*$, which by the strict monotonicity of $\sigma$ uniquely determines a target $\delta_{target}$ for each preference pair. Under Assumptions 4.1–4.3, Assumption 4.4 ($\ell^2$-$\delta$-proximity), Lemma L.1, and Lemma L.6, MLE yields the returned policy satisfying (up to the error $\epsilon_{approx} + \epsilon_{opt} + L_\sigma^{-1}\epsilon_{stat}$):

$$\beta(\delta_{\pi_{cons}^*} - \delta_{\pi_{ref}}) - \Psi_{cons}(\delta_{\pi_{ref}}) = \Delta r^*, \tag{181}$$

which rearranges to:

$$\delta_{\pi_{cons}^*} = \delta_{\pi_{ref}} + \frac{\Delta r^*}{\beta} + \Phi_{cons}(\delta_{\pi_{ref}}), \tag{182}$$

where we used $\Psi_{cons} = \beta\Phi_{cons}$ (Definition L.15).

**Step 2: Upper Bound Property**

By Corollary L.14:

$$\Phi_{\text{cons}}(\delta_{\pi_{\text{ref}}}) \geq \Phi(\delta_{\pi_{\text{ref}}}, \Delta r^*) \tag{183}$$

Therefore:

$$\delta_{\pi_{\text{cons}}^*} = \delta_{\pi_{\text{ref}}} + \frac{\Delta r^*}{\beta} + \Phi_{\text{cons}}(\delta_{\pi_{\text{ref}}}) \tag{184}$$

$$\geq \delta_{\pi_{\text{ref}}} + \frac{\Delta r^*}{\beta} + \Phi(\delta_{\pi_{\text{ref}}}, \Delta r^*) \tag{185}$$

$$= \delta_{\pi_{\text{EC-RLHF}}^*}(\Delta r^*) \tag{186}$$

This proves property (1).

**Step 3: Worst-Case Margin Structure**

By Definition L.15, $\Phi_{\text{cons}}(\delta_{\pi_{\text{ref}}}) = \Phi(\delta_{\pi_{\text{ref}}}, 0)$. The E-CPOC margin function is structurally identical to the general adaptive margin function (Definition L.9) evaluated at $\Delta r^* = 0$.

From Eq. (182), the E-CPOC optimal policy satisfies:

$$\delta_{\pi_{\text{cons}}^*} = \delta_{\pi_{\text{ref}}} + \frac{\Delta r^*}{\beta} + \Phi_{\text{cons}}(\delta_{\pi_{\text{ref}}}). \tag{187}$$

Meanwhile, by Theorem 4.12, the EC-RLHF optimal policy evaluated at $\Delta r^* = 0$ satisfies:

$$\delta_{\pi_{\text{EC-RLHF}}^*}(\Delta r^* = 0) = \delta_{\pi_{\text{ref}}} + \frac{0}{\beta} + \Phi(\delta_{\pi_{\text{ref}}}, 0) = \delta_{\pi_{\text{ref}}} + \Phi_{\text{cons}}(\delta_{\pi_{\text{ref}}}). \tag{188}$$

Combining these two equalities yields:

$$\delta_{\pi_{\text{cons}}^*} = \delta_{\pi_{\text{EC-RLHF}}^*}(\Delta r^* = 0) + \frac{\Delta r^*}{\beta}. \tag{189}$$

Since $\Delta r^* > 0$, we have $\delta_{\pi_{\text{cons}}^*} > \delta_{\pi_{\text{EC-RLHF}}^*}(\Delta r^* = 0)$.

**Interpretation:** The E-CPOC optimal policy inherits the worst-case margin correction $\Phi_{\text{cons}}(\delta_{\pi_{\text{ref}}})$ from EC-RLHF at $\Delta r^* = 0$, while additionally benefiting from the true positive reward difference $\Delta r^*/\beta$. This means E-CPOC is *strictly more conservative* than the worst-case EC-RLHF policy, which is precisely the desired property.

This proves property (2).

**Step 4: Absolute Advantage Guarantee**

We wish to ensure $\delta_{\pi_{\text{cons}}^*} > 0$ for all preference pairs. From Eq. (182):

$$\delta_{\pi_{\text{cons}}^*} = \delta_{\pi_{\text{ref}}} + \frac{\Delta r^*}{\beta} + \Phi_{\text{cons}}(\delta_{\pi_{\text{ref}}}). \tag{190}$$

We use an *exact* lower bound on $\delta_{\pi_{\text{ref}}} + \Phi_{\text{cons}}(\delta_{\pi_{\text{ref}}})$. By the softplus definition $\Phi_{\text{cons}}(\delta_{\pi_{\text{ref}}}) = \frac{1}{\tau} \log(1 + \exp(\tau(\gamma - \delta_{\pi_{\text{ref}}})))$, we have the *strict* inequality:

$$\Phi_{\text{cons}}(\delta_{\pi_{\text{ref}}}) > \gamma - \delta_{\pi_{\text{ref}}} \tag{191}$$

for all $\delta_{\pi_{\text{ref}}} \in \mathbb{R}$, since $\frac{1}{\tau} \log(1 + e^{\tau z}) > z$ for all $z \in \mathbb{R}$ and all $\tau > 0$ (the softplus function *strictly* dominates the identity, as $\log(1 + e^{\tau z}) > \tau z$ follows from $1 + e^{\tau z} > e^{\tau z}$). Therefore:

$$\delta_{\pi_{\text{ref}}} + \Phi_{\text{cons}}(\delta_{\pi_{\text{ref}}}) > \delta_{\pi_{\text{ref}}} + (\gamma - \delta_{\pi_{\text{ref}}}) = \gamma, \tag{192}$$

and consequently:

$$\delta_{\pi_{\text{cons}}^*} = \delta_{\pi_{\text{ref}}} + \frac{\Delta r^*}{\beta} + \Phi_{\text{cons}}(\delta_{\pi_{\text{ref}}}) > \gamma + \frac{\Delta r^*}{\beta} > \gamma > 0, \tag{193}$$

where the first strict inequality uses the softplus bound (191) and the second uses $\Delta r^* > 0$ from the Bradley-Terry model. This bound is exact (non-asymptotic) and holds for all $\tau > 0$.

We choose $\gamma > 0$ directly as a hyperparameter. To ensure the constraint is meaningful (i.e., binding for the hardest samples), we set:

$$\gamma \geq \gamma^*_{\text{cons}} := \max_{(x, y_w, y_l) \in \mathcal{D}} \{-\delta_{\pi_{\text{ref}}}(x, y_w, y_l)\}, \tag{194}$$

which guarantees $\delta_{\pi^*_{\text{cons}}} > \gamma + \Delta r^*/\beta > \gamma > 0$ for all preference pairs, proving property (3).

**Step 5: Sample-Adaptive Behavior**

From the softplus formulation $\Phi_{\text{cons}}(\delta_{\pi_{\text{ref}}}) = \frac{1}{\tau} \log(1 + \exp(\tau(\gamma - \delta_{\pi_{\text{ref}}})))$:

- When $\delta_{\pi_{\text{ref}}} \ll 0$ (difficult samples): $\gamma - \delta_{\pi_{\text{ref}}} \gg 0$, so:

$$\Phi_{\text{cons}}(\delta_{\pi_{\text{ref}}}) \approx \frac{1}{\tau} \cdot \tau(\gamma - \delta_{\pi_{\text{ref}}}) = \gamma - \delta_{\pi_{\text{ref}}} \tag{195}$$

- When $\delta_{\pi_{\text{ref}}} \gg \gamma$ (easy samples): $\gamma - \delta_{\pi_{\text{ref}}} \ll 0$, so:

$$\Phi_{\text{cons}}(\delta_{\pi_{\text{ref}}}) \approx \frac{1}{\tau} \log(1 + \exp(\tau(\gamma - \delta_{\pi_{\text{ref}}}))) \approx \frac{1}{\tau} \exp(\tau(\gamma - \delta_{\pi_{\text{ref}}})) \to 0 \tag{196}$$

- When $\delta_{\pi_{\text{ref}}} \approx \gamma$ (neutral samples): $\gamma - \delta_{\pi_{\text{ref}}} \approx 0$, so:

$$\Phi_{\text{cons}}(\delta_{\pi_{\text{ref}}}) \approx \frac{1}{\tau} \log(1 + 1) = \frac{\log 2}{\tau} \tag{197}$$

This proves property (4). $\qquad\qquad\square$

**Corollary L.18** (Verifiable Equivalence Guarantee). *Under Assumptions 4.1–4.3 (Bradley-Terry model, approximate realizability, finite-sample data) and Assumption 4.5 (non-degenerate preferences with curvature $\kappa_0 > 0$), if the returned policy $\hat{\pi}$ achieves a sufficiently small loss gap $\epsilon_{\text{loss}} := \mathcal{L}(\hat{\pi}) - \inf_\theta \mathcal{L}(\pi_\theta)$ (see Proposition L.3 for the precise threshold), then Assumption 4.4 ($\ell^2$-$\delta$-proximity) holds with the $N$-independent bound*

$$\epsilon_{\text{opt},2} = \sqrt{\frac{2\epsilon_{\text{loss}}}{\beta^2 \kappa_0}},$$

*and by Lemma L.1, the pointwise bound $\epsilon_{\text{opt}} = \sqrt{N}\,\epsilon_{\text{opt},2} = \sqrt{2N\epsilon_{\text{loss}}/(\beta^2\kappa_0)}$ holds. All guarantees of Theorem L.17 hold with this explicit $\epsilon_{\text{opt}}$. In particular, the E-CPOC policy satisfies:*

$$|\delta_{\hat{\pi}}(x_i) - \delta_{\text{target}}(x_i)| \leq \epsilon_{\text{approx}} + \sqrt{\frac{2N\epsilon_{\text{loss}}}{\beta^2 \kappa_0}} + L_\sigma^{-1} \epsilon_{\text{stat}}, \quad \forall i.$$

*The bound is exact in the convex case ($\delta_{\pi_\theta}$ affine in $\theta$) and holds to leading order in general (Proposition L.3). This corollary provides a* verifiable *equivalence guarantee: the loss gap $\epsilon_{\text{loss}}$ is directly observable during training, $\epsilon_{\text{stat}}$ is estimable from data, and $\epsilon_{\text{approx}}$ and $\kappa_0$ are properties of the model class and the optimal solution. It follows from Proposition L.3 (loss suboptimality plus non-degeneracy implies $\ell^2$-$\delta$-proximity) combined with Lemma L.1 and Theorem L.17.*

**Remark L.19** (Expanded Error Bound). *The unified error bound of Lemma L.6 and Theorem L.17 takes the general form:*

$$|\delta_{\hat{\pi}}(x_i) - \delta_{\text{target}}(x_i)| \leq \epsilon_{\text{approx}} + \epsilon_{\text{opt}} + L_\sigma^{-1} \epsilon_{\text{stat}}, \quad \forall i,$$

*where $\epsilon_{\text{opt}} = \sqrt{N}\,\epsilon_{\text{opt},2}$ is the pointwise bound derived from the $\ell^2$-$\delta$-proximity parameter $\epsilon_{\text{opt},2}$ (Assumption 4.4) via Lemma L.1. When $\epsilon_{\text{opt},2}$ is obtained via the bridge (Proposition L.3, requiring loss suboptimality and Assumption 4.5; cf. Corollary L.18), this specializes to:*

$$|\delta_{\hat{\pi}}(x_i) - \delta_{\text{target}}(x_i)| \leq \epsilon_{\text{approx}} + \sqrt{\frac{2N\epsilon_{\text{loss}}}{\beta^2 \kappa_0}} + L_\sigma^{-1} \epsilon_{\text{stat}}, \quad \forall i.$$

*This expanded formulation has three practical advantages: (a) the loss gap $\epsilon_{\text{loss}}$ is directly observable during training, (b) the bound makes explicit the dependence on the dataset size $N$, regularization strength $\beta$, and preference decisiveness $\kappa_0$ (Assumption 4.5), and (c) the $\sqrt{N}$ factor is transparently attributable to the $\ell^2$-to-$\ell^\infty$ norm conversion (Lemma L.1), while the underlying $\ell^2$-$\delta$-proximity bridge from loss suboptimality is $N$-independent.*

**Remark L.20** (Properties of the E-CPOC Equivalence)**.** *The E-CPOC equivalence result (Theorem L.17) has the following notable properties:*

| Property | E-CPOC |
|---|---|
| Constraint type | Hard (KKT, worst-case) |
| Reward model | Not needed |
| Margin function | $\beta\Phi_{\text{cons}}(\delta_{\pi_{\text{ref}}})$ |
| Margin adaptivity | Conservative $\Phi_{\text{cons}}$ |
| Equivalence type | Upper bound |
| Error bound | $\Phi_{\text{cons}} \geq \Phi$ (conservative) |
| Assumptions | 4 core ($\ell^2$-$\delta$-proximity) + 1 structural (all mild/standard); bridge via Cor. L.18 |
| Guarantee | Conservative absolute advantage |

*Key Insights:*

- *E-CPOC enforces preference alignment with a worst-case bound via hard constraint $\delta_\pi \geq \gamma$. The upper bound equivalence ensures $\delta_{\pi^*_{E\text{-}CPOC}} \geq \delta_{\pi^*_{EC\text{-}RLHF}}(\Delta r^*)$ for any $\Delta r^* > 0$, without requiring a reward model. The conservative adaptive margin $\beta\Phi_{cons}(\delta_{\pi_{ref}})$ depends only on the reference policy.*

- *The structural decomposition $\delta_{\pi^*} = \delta_{\pi^*_{EC\text{-}RLHF}}(\Delta r^* = 0) + \Delta r^*/\beta$ (worst-case margin plus true reward gap) reveals that E-CPOC is strictly more conservative than the worst-case EC-RLHF. When a reward model is available, the tighter margin $\Phi(\delta_{\pi_{ref}}, \Delta\hat{r})$ can be used instead (Remark L.12).*

- *The core E-CPOC equivalence requires only four mild assumptions (Assumptions 4.1–4.4; see Sec. 4.1), with the connected comparison graph condition (Condition 4.6) needed solely for the extension to full policy equivalence. The $\ell^2$-$\delta$-proximity condition (Assumption 4.4)—formulated in the natural mean-square norm that the loss function directly controls—can be derived from the verifiable loss suboptimality condition and the mild non-degeneracy condition via Proposition L.3 with an $N$-independent bound, rather than assumed directly. The pointwise bound required by the error propagation is then recovered via the standard $\ell^2$-to-$\ell^\infty$ conversion (Lemma L.1). Corollary L.18 provides the combined verifiable guarantee, cleanly separating the core theoretical requirements from the practical verification pathway.*

*Practical Implications:*

- *Use **CPO** when simplicity is preferred (single hyperparameter $\gamma$) and soft constraint encouragement is sufficient.*

- *Use **E-CPOC** when provable alignment guarantees are needed, sample-adaptive margins that automatically adjust for difficult preference pairs are desired, or when reference policy quality varies across the dataset.*

*Both methods address DPO's implicit assumption violation (Assumption 3.1) without requiring a reward model. E-CPOC provides stronger theoretical guarantees through its hard constraint formulation and provable equivalence to explicitly constrained RLHF.*

# M. Preference Learning as Reranking

## M.1. DPO as Soft Margin Ranking

Proof of Proposition 5.1 (DPO as Soft Margin Ranking).

*Proof.* Define $z = \delta_{\pi_\theta} - \delta_{\pi_{\text{ref}}}$. The DPO loss can be written as:

$$\mathcal{L}_{\text{DPO}} = \log(1 + \exp(-\beta z)) = \text{softplus}(-\beta z) \tag{198}$$

We analyze the asymptotic behavior for three cases:

**Case 1 ($z > 0$, margin satisfied):** When $\delta_{\pi_\theta} > \delta_{\pi_{\text{ref}}}$, as $\beta \to \infty$ with $z > 0$ fixed, we have $\exp(-\beta z) \to 0$. Using the Taylor expansion $\log(1 + \epsilon) \approx \epsilon$ for small $\epsilon$:

$$\mathcal{L}_{\text{DPO}} \approx \exp(-\beta z) \to 0 \tag{199}$$

Therefore:

$$\lim_{\beta \to \infty} \frac{1}{\beta} \mathcal{L}_{\text{DPO}} = \lim_{\beta \to \infty} \frac{\exp(-\beta z)}{\beta} = 0 = \max(0, -z) \tag{200}$$

**Case 2 ($z < 0$, margin violated):** When $\delta_{\pi_\theta} < \delta_{\pi_{\text{ref}}}$, we rewrite:

$$\mathcal{L}_{\text{DPO}} = \log(1 + \exp(-\beta z)) \tag{201}$$
$$= \log(\exp(-\beta z)(1 + \exp(\beta z))) \tag{202}$$
$$= -\beta z + \log(1 + \exp(\beta z)) \tag{203}$$

As $\beta \to \infty$ with $z < 0$ fixed, we have $\exp(\beta z) \to 0$ (since $\beta z < 0$). Thus:

$$\log(1 + \exp(\beta z)) \approx \exp(\beta z) \to 0 \tag{204}$$

Therefore:

$$\mathcal{L}_{\text{DPO}} \approx -\beta z = \beta(\delta_{\pi_{\text{ref}}} - \delta_{\pi_\theta}) \tag{205}$$

and:

$$\lim_{\beta \to \infty} \frac{1}{\beta} \mathcal{L}_{\text{DPO}} = -z = \delta_{\pi_{\text{ref}}} - \delta_{\pi_\theta} = \max(0, -z) \tag{206}$$

**Case 3 ($z = 0$, boundary):** When $\delta_{\pi_\theta} = \delta_{\pi_{\text{ref}}}$:

$$\mathcal{L}_{\text{DPO}} = \log(1 + e^0) = \log 2 \tag{207}$$

Therefore:

$$\lim_{\beta \to \infty} \frac{1}{\beta} \mathcal{L}_{\text{DPO}} = \lim_{\beta \to \infty} \frac{\log 2}{\beta} = 0 = \max(0, 0) \tag{208}$$

Combining all cases:

$$\lim_{\beta \to \infty} \frac{1}{\beta} \mathcal{L}_{\text{DPO}} = \begin{cases} 0 & \text{if } z > 0 \\ -z & \text{if } z < 0 \\ 0 & \text{if } z = 0 \end{cases} = \max(0, -z) = \max(0, \delta_{\pi_{\text{ref}}} - \delta_{\pi_\theta}) \tag{209}$$

$\square$

## M.2. Proof of CPO as Corrected Soft Margin Ranking

Proof of Theorem 5.2 (CPO as Corrected Soft Margin Ranking).

*Proof.* Define $z = \delta_{\pi_\theta} - \delta_{\pi_{\text{ref}}} - 2\gamma/\beta$. The CPO loss is:

$$\mathcal{L}_{\text{CPO}} = \log(1 + \exp(-\beta z)) \tag{210}$$

Following the same analysis as Proposition 5.1:

**Case 1 ($z > 0$):** As $\beta \to \infty$, $\exp(-\beta z) \to 0$, so:

$$\lim_{\beta \to \infty} \frac{1}{\beta} \mathcal{L}_{\text{CPO}} = 0 \tag{211}$$

**Case 2** ($z < 0$): As $\beta \to \infty$:

$$\mathcal{L}_{\text{CPO}} \approx -\beta z = \beta \left( \delta_{\pi_{\text{ref}}} + \frac{2\gamma}{\beta} - \delta_{\pi_\theta} \right) \tag{212}$$

Therefore:

$$\lim_{\beta \to \infty} \frac{1}{\beta} \mathcal{L}_{\text{CPO}} = \delta_{\pi_{\text{ref}}} + \frac{2\gamma}{\beta} - \delta_{\pi_\theta} \tag{213}$$

**Case 3** ($z = 0$): $\lim_{\beta \to \infty} \frac{1}{\beta} \mathcal{L}_{\text{CPO}} = 0$.

Combining: $\lim_{\beta \to \infty} \frac{1}{\beta} \mathcal{L}_{\text{CPO}} = \max(0, \delta_{\pi_{\text{ref}}} + 2\gamma/\beta - \delta_{\pi_\theta})$.

By Corollary 4.9, when $\gamma \geq \gamma^*$:

$$\gamma^* = \frac{\beta}{2} \max_{(x, y_w, y_l) \in \mathcal{D}} \max \left\{ 0, -\delta_{\pi_{\text{ref}}}(x, y_w, y_l) - \frac{r^*(y_w) - r^*(y_l)}{\beta} \right\} \tag{214}$$

This ensures that for all preference pairs:

$$m^*_{\text{eff}} = \delta_{\pi_{\text{ref}}} + \frac{2\gamma^*}{\beta} \geq \max\{0, \delta_{\pi_{\text{ref}}} + \Delta r^*/\beta\} > 0 \tag{215}$$

Therefore, the optimization only stops when $\delta_{\pi_\theta} > m^*_{\text{eff}} > 0$, guaranteeing absolute preference alignment. $\qquad\square$

## M.3. Proof of E-CPOC as Adaptive Margin Ranking

Proof of Theorem 5.3 (E-CPOC as Adaptive Margin Ranking).

*Proof.* Define $z = \delta_{\pi_\theta} - \delta_{\pi_{\text{ref}}} - \Phi_{\text{cons}}(\delta_{\pi_{\text{ref}}})$, where $\Phi_{\text{cons}}(\delta_{\pi_{\text{ref}}}) = \Phi(\delta_{\pi_{\text{ref}}}, 0)$ is the conservative margin function. Following the same analysis as Proposition 5.1, we obtain:

$$\lim_{\beta \to \infty} \frac{1}{\beta} \mathcal{L}_{\text{E-CPOC}} = \max(0, -z) = \max(0, \delta_{\pi_{\text{ref}}} + \Phi_{\text{cons}}(\delta_{\pi_{\text{ref}}}) - \delta_{\pi_\theta}) \tag{216}$$

The effective target margin is:

$$m^*(\delta_{\pi_{\text{ref}}}) = \delta_{\pi_{\text{ref}}} + \Phi_{\text{cons}}(\delta_{\pi_{\text{ref}}}) \tag{217}$$

By Proposition J.9, $\Phi_{\text{cons}}(\delta_{\pi_{\text{ref}}}) \geq 0$ for all $\delta_{\pi_{\text{ref}}}$, and:

- When $\delta_{\pi_{\text{ref}}} \ll 0$ (difficult): $\Phi_{\text{cons}}(\delta_{\pi_{\text{ref}}}) \approx \gamma - \delta_{\pi_{\text{ref}}}$ by the softplus approximation, so $m^* \approx \gamma > 0$

- When $\delta_{\pi_{\text{ref}}} \gg 0$ (easy): $\Phi_{\text{cons}}(\delta_{\pi_{\text{ref}}}) \approx 0$, so $m^* \approx \delta_{\pi_{\text{ref}}} > 0$

Therefore, $m^*(\delta_{\pi_{\text{ref}}}) > 0$ for all $\delta_{\pi_{\text{ref}}}$ when $\gamma > 0$, ensuring absolute preference alignment. $\qquad\square$

