# OpenReview forum: "Conditional Equivalence of DPO and RLHF: Assumptions, Failure Modes, and Provable Alignment"
_ICML.cc/2026/Conference — ICML 2026 spotlight_

### Official Review · Reviewer_32Rv · 2026-03-10

**Soundness:** 3
**Presentation:** 3
**Significance:** 3
**Originality:** 3
**Overall Recommendation:** 4
**Confidence:** 3

**Summary:**

This paper investigates the theoretical foundations of Direct Preference Optimization (DPO), challenging the widely accepted claim that DPO and Reinforcement Learning from Human Feedback (RLHF) optimize mathematically equivalent objectives. The authors prove that this equivalence is strictly conditional, relying on an implicit assumption: the RLHF-optimal policy must inherently prefer human-preferred responses over dispreferred ones.

The authors demonstrate that when the reference policy ($\pi_{ref}$) is poorly aligned, the KL divergence penalty acts as a "poisonous anchor". While standard RLHF accepts a misaligned compromise to satisfy this KL constraint, DPO suffers a more pathological failure: it gets trapped in an "undesirable solution space" where gradients vanish precisely as the policy attempts to cross the preference boundary. In these out-of-distribution scenarios, DPO stops optimizing for absolute task optimality and instead optimizes for relative distinguishability from the reference model. In the context of LLM alignment, this over-reliance on a mediocre reference model fundamentally restricts the language model to the "human bias distribution" of the initial Supervised Fine-Tuning (SFT) phase, effectively cutting off the discovery of novel, highly preferred reasoning paths or generative styles that fall outside the reference support.

To resolve this, the authors introduce Constrained Preference Optimization (CPO) and Explicitly Constrained Preference Optimization (E-CPO). By framing preference learning as a reranking problem, they augment the standard RLHF objective with explicit, adaptive margin constraints that guarantee absolute preference alignment without requiring a separate reward model. Empirical evaluations on standard LLM benchmarks (AlpacaEval 2 and Arena-Hard) demonstrate that CPO achieves state-of-the-art performance, outperforming standard DPO and other preference-tuning baselines.

**Compliance With Llm Reviewing Policy:**

Affirmed.

**Key Questions For Authors:**

1. **The "Poisonous Anchor" & Distributional Shifts:** Your theoretical analysis brilliantly identifies that DPO fails when the reference policy ($\pi_{ref}$) systematically disfavors human-preferred responses. However, the proposed Constrained Preference Optimization (CPO) still retains the KL divergence penalty to the reference policy. In LLM alignment, tethering a model to a mediocre SFT reference policy strictly bounds it to that initial distribution, acting as a "poisonous anchor" that prevents the discovery of highly novel reasoning paths or significant behavioral shifts. Does CPO offer any conceptual mechanism to relax this anchor, or is it inherently limited to the generative support of the reference model?

2. **Empirical Demonstration of the Pathology:** The paper's core theoretical contribution is proving the existence of an "undesirable solution space" where DPO gradients vanish if the reference LLM is severely misaligned. Yet, the primary experiments utilize Llama-3-8B-Instruct, which is already a highly capable model. Could you provide an ablation study where the reference LLM is deliberately degraded or heavily biased? Showing standard DPO pathologically collapsing while CPO succeeds in this exact scenario would drastically strengthen the paper's narrative.

3. **Precomputation Overhead:** Algorithm 1 requires precomputing reference-based adaptive margins across the entire preference dataset. While you correctly note this ensures the optimization objective is stationary, what is the practical storage and computational overhead of this step when scaling to datasets with millions of preference pairs, especially compared to standard DPO?

4. **Hyperparameter Stability:** The theoretical guarantees for CPO's absolute advantage rely on choosing a margin parameter $\gamma$ that adequately compensates for the worst-case preference pair. In practice, how sensitive is the empirical performance on benchmarks like Arena-Hard to the exact tuning of $\gamma$ (or the smoothness parameter $\tau$ in E-CPO)?

**Limitations:**

The authors provide a robust theoretical analysis but currently lack a dedicated limitations section, stating in their Impact Statement that there are no specific societal consequences to highlight. The paper would be much stronger if the authors engaged with the following constraints:

- **The Reference-Bound Ceiling (The Poisonous Anchor):** By solving the relative-advantage pathology of DPO while maintaining the KL divergence constraint to the reference policy, the framework remains a heavily support-constrained method. The authors should acknowledge that while this successfully stabilizes conversational LLMs, it fundamentally limits the algorithm's application to scenarios where surpassing the reference policy's initial capabilities (e.g., complex reasoning tasks where the SFT model is weak) is required.

- **Theoretical vs. Empirical Gap on Failure Modes:** While the theoretical proofs characterizing DPO's failure modes and CPO's absolute advantage guarantees are highly rigorous, the lack of empirical validation in a deliberately out-of-distribution or highly misaligned setting leaves a minor gap in proving that these specific mathematical pathologies are the primary bottleneck in real-world training runs.

- **Societal Impacts and Bias Reinforcement:** Contrary to the authors' impact statement, preference learning inherently risks efficiently reinforcing the biases present in the human preference dataset. Because CPO forces strict adherence to these preferences via an adaptive margin, it may leash the LLM to specific human biases more aggressively than standard RLHF. Acknowledging this would represent a more mature reflection on the technology.

**Strengths And Weaknesses:**

**Strengths:**

- **Originality & Theoretical Soundness (Exposing the Implicit Assumption):** The paper’s most significant contribution is its meticulous deconstruction of the DPO derivation. Identifying that DPO fundamentally relies on an often-violated implicit assumption—that the RLHF-optimal policy must inherently prefer human-preferred responses—is a major theoretical insight. The formal proof of the "undesirable solution space" where DPO gradients vanish despite preference violations is elegant and mathematically rigorous.

- **Conceptual Clarity (Margin Ranking Perspective):** The geometric interpretation mapping DPO to a soft margin ranking loss with a potentially _negative_ target margin ($\delta_{\pi_{ref}}$) provides excellent, intuitive clarity. This clearly explains why DPO can pathologically optimize for relative distinguishability rather than absolute alignment.

- **Practical & Constructive Solution (CPO/E-CPO):** Rather than just diagnosing a flaw, the authors provide a highly practical fix. CPO and Conservative E-CPO successfully introduce an adaptive margin to ensure absolute preference alignment. Crucially, they maintain DPO's primary advantage by eliminating the need for a separate reward model.

- **Empirical Validation:** The method achieves state-of-the-art results against highly competitive baselines (SimPO, DPO, KTO) on rigorous, length-controlled conversational benchmarks like Arena-Hard and AlpacaEval 2.


**Weaknesses & Areas for Improvement:**

- **Significance (The "Poisonous Anchor" and LLM Capabilities):** While CPO prevents DPO from falling into pathological vanishing gradients, it does not escape the fundamental limitation of the KL-divergence penalty. By tethering the optimization to $\pi_{ref}$, the framework still acts as a "poisonous anchor". For LLMs, forcing the model to stay within the distribution of a mediocre SFT reference model fundamentally limits the agent's ability to discover "outside-the-box" solutions (e.g., novel reasoning steps in math/code, or major behavioral shifts). The paper would be significantly strengthened by acknowledging this conceptual limitation of reference-bound optimization, even when constrained.

- **Empirical Scope (Lack of Misaligned $\pi_{ref}$ Testing):** The theoretical claims revolve around the failure modes that occur when the reference policy is misaligned ($\delta_{\pi_{ref}} \le -\frac{\Delta r^*}{\beta}$). However, the experiments only test standard conversational LLMs (Llama-3-8B-Instruct) where the reference policy is already highly capable. To truly validate the theory, the empirical section desperately needs a controlled experiment where the LLM $\pi_{ref}$ is explicitly corrupted or misaligned to prove that CPO survives where standard DPO collapses.

- **Hyperparameter Sensitivity and Margin Selection:** The success of CPO relies heavily on the margin parameter $\gamma$, and E-CPO introduces a smoothness parameter $\tau$. While the authors introduce a "Conservative E-CPO" to bound this automatically without a reward model, the main text lacks ablation studies showing how sensitive the empirical performance is to the choice of these hyperparameters.

- **Presentation (Clarity of Algorithm Overhead):** Algorithm 1 requires precomputing the reference-based adaptive margins $\tilde{\gamma}_{ref}^{(i)}$ for the entire dataset. While the authors claim this makes the optimization objective stationary, they should clarify the computational overhead and memory footprint of this step compared to standard DPO or SimPO, especially when scaling to datasets with millions of preference pairs.

---

> ### Author Rebuttal · Authors · 2026-03-31
>
> We sincerely thank the reviewer for the thorough and encouraging evaluation, which **recognizes the originality and theoretical soundness** of our analysis, the **conceptual clarity** of the margin ranking interpretation, and the **practical and constructive solution** provided by CPO. Below, we address the remaining concerns in detail:
> >Q1: The "Poisonous Anchor"—KL regularization to $\pi_{\text{ref}}$ limits CPO's ability to discover novel solutions beyond the reference policy's support.
>
> **A1:** We thank the reviewer for this thought-provoking observation.
>
> **1: The "poisonous anchor" is a fundamental property of the KL-regularized RLHF framework itself, not specific to CPO.** DPO, RDPO, IPO, and all methods derived from the RLHF objective (Eq. 2) share this constraint by design. CPO's scope is to fix DPO's *conditional equivalence* issue within this framework, not to redesign the RLHF paradigm. Addressing the reference-bound ceiling would require moving beyond KL-regularized RLHF entirely.
>
> **2: CPO's constraint term actually provides *more* flexibility than standard DPO to deviate from $\pi_{ref}$ on critical pairs.** The adaptive margin $\tilde\gamma_{ref}$ (Eq. 25) pushes $\delta_{\pi_\theta}$ beyond what the KL penalty alone would allow, especially for hard pairs where $\pi_{ref}$ is misaligned. In this sense, CPO partially relaxes the anchor effect precisely where it matters most. We agree that acknowledging this inherent limitation of reference-bound optimization would strengthen the paper and will add a discussion in the revision.
>
> >Q2: Lack of misaligned $\pi_{\text{ref}}$ experiment—need controlled ablation with deliberately degraded/biased reference policy to validate the theory.
>
> **A2:** We thank the reviewer for this suggestion.
>
> **1: Even with the current instruction-tuned $\pi_{\text{ref}}$, the assumption violation is already substantial.** As shown in our new analysis ([Anonymous Figure](https://figshare.com/s/0c2474c065f7c293c194)), **45.5% of preference pairs** violate Assumption 3.1 under Llama-3-8B-Instruct with $\beta=0.1$. This demonstrates that the pathology is far from hypothetical—it affects nearly half the training data even with a highly capable reference model.
>
> **2: We agree that a controlled ablation with deliberately misaligned $\pi_{\text{ref}}$ would further strengthen the narrative.** Due to time constraints of the rebuttal period, we will prioritize it in the revision. We expect that with a weaker/misaligned reference, the violation rate would increase significantly beyond 45.5%, producing a sharper qualitative split between DPO and CPO as the theory predicts.
>
> >Q3: Hyperparameter sensitivity—$\gamma$ and $\tau$ lack ablation studies showing empirical robustness.
>
> **A3:** We are grateful to the reviewer for this valuable feedback. Weconduct the $\gamma$ sensitivity analysis. Please refer to the **A4 for Reviewer sXEr.** Regarding $\tau$: it is a hyperparameter of E-CPO-C only (Sec. 4.4), which was not experimentally evaluated in this submission. We will include $\tau$ sensitivity analysis alongside E-CPO-C experiments in the revision.
>
> >Q4: Precomputation overhead—scalability of precomputing $\tilde{\gamma}_{\text{ref}}^{(i)}$ for large datasets compared to DPO/SimPO.
>
> **A4:** We appreciate the reviewer for raising this practical concern. We would like to clarify that **CPO's precomputation overhead is essentially identical to DPO's**. Standard DPO already requires a forward pass over the entire dataset to compute and cache $\delta_{\text{ref}}^{(i)} = \log \pi_{\text{ref}}(y_w|x) - \log \pi_{\text{ref}}(y_l|x)$ before training. CPO simply reuses the same forward pass to additionally compute $\tilde\gamma_{\text{ref}}^{(i)} = \gamma(1/\pi_{\text{ref}}(y_w|x) + 1/\pi_{\text{ref}}(y_l|x))$—two scalar divisions and one addition per sample, with negligible cost. The storage overhead is one extra scalar per sample. During training, CPO's per-iteration cost is identical to DPO: the only difference is subtracting $\tilde{\gamma}_{\text{ref}}^{(i)}$ from the logits (Algorithm 1, line 11). There is **no additional forward/backward pass, no reward model inference, and no architectural change**. CPO scales to millions of pairs as easily as DPO. We will clarify this in the revision.
>
> >Q5: Societal impacts—CPO's stronger adherence to preference data may reinforce human biases more aggressively than standard RLHF.
>
> **A5:** This is an insightful observation. We note that this concern is **shared across all preference learning methods** (DPO, SimPO, IPO, RLHF)—CPO makes learning more faithful to preferences, whose impact depends on data quality. Notably, **CPO's $\gamma$ provides a controllable lever**: smaller $\gamma$ reduces enforcement (recovering DPO at $\gamma=0$), allowing practitioners to calibrate adherence based on data confidence. We will add a dedicated limitations section discussing this in the revision.

---

> > ### Author Rebuttal · Reviewer_32Rv · 2026-04-02
> >
> > I thank the authors for a thorough rebuttal. The precomputation overhead concern (Q4) is fully resolved—CPO's cost is essentially identical to DPO. The finding that 45.5% of preference pairs violate Assumption 3.1 even with a strong reference model meaningfully strengthens the paper's motivation.
> > However, my main concern remains unresolved. The controlled ablation with a deliberately misaligned π_ref is absent. Showing the assumption is frequently violated is not the same as demonstrating DPO collapses while CPO survives—the paper's core narrative requires this experiment.
> > On the remaining points: the scoping of CPO within the KL-regularized paradigm (Q1) is fair, though the revision should be precise about what "partially relaxes the anchor" means in practice. The γ sensitivity analysis (Q3) was cross-referenced to another reviewer's response rather than addressed directly. Regarding societal impacts (Q5), the authors argue that γ acts as a controllable lever. However, this creates a tradeoff the paper does not discuss: fixing the identified pathology requires a sufficiently large γ, but a large γ also forces the model to follow the preference data more strictly—including any biases present in that data.
> > I maintain my score.

---

> > > ### Author Response · Authors · 2026-04-03
> > >
> > > We thank the reviewer for the valuable follow-up questions and constructive feedback to improve our paper quality.
> > >
> > > **Regarding the controlled ablation with a deliberately misaligned $\pi_{\mathrm{ref}}$.** Please refer to our **Reply Rebuttal Comment for Reviewer sKpJ**, where we present a comprehensive controlled experiment that directly addresses this concern. In summary, we systematically construct misaligned reference models by SFT on rejected responses with corruption ratios $R \in \{0.2, 0.3, 0.4\}$, then train both DPO and CPO from these misaligned references. The training dynamics further confirm that **DPO gets trapped in the undesirable space $\mathcal{U}$ while CPO escapes**.
> > >
> > >  **Regarding the scoping of CPO within the KL-regularized paradigm (Q1).** We appreciate the reviewer's acknowledgment that our scoping is fair. We will revise the paper to be more precise about what "partially relaxes the anchor" means in practice: specifically, the constraint term $\tilde\gamma_{\mathrm{ref}}$ does not remove the KL regularization toward $\pi_{\mathrm{ref}}$, but rather introduces an additional adaptive margin that counteracts the gradient vanishing effect when $\delta_{\pi_{\mathrm{ref}}} < 0$. The reference policy $\pi_{\mathrm{ref}}$ still serves as the anchor for KL regularization, while the constraint ensures that the optimization does not get trapped in $\mathcal{U}$ due to vanishing gradients. We will clarify this distinction in the revised manuscript.
> > >
> > > **Regarding the societal impact and $\gamma$ tradeoff (Q5).** The reviewer raises an important point. We agree that a larger $\gamma$ forces stricter adherence to preference data, which amplifies any biases present in that data. We will add a discussion of this tradeoff in the revision. Specifically, we will note that: (1) $\gamma$ controls the strength of the preference alignment constraint—larger $\gamma$ provides stronger theoretical guarantees against the identified pathology but also increases sensitivity to data quality; (2) in practice, this tradeoff also deserve more attention to manage by combining CPO with standard data curation and debiasing techniques.

---

### Official Review · Reviewer_sXEr · 2026-03-12

**Soundness:** 3
**Presentation:** 2
**Significance:** 3
**Originality:** 2
**Overall Recommendation:** 4
**Confidence:** 4

**Summary:**

This paper studies preference optimization for LLM alignment in the RLHF/DPO setting and shows that the claimed equivalence between DPO and RLHF is only conditional. It argues that DPO relies on an implicit assumption that the RLHF-optimal policy must assign higher probability to human-preferred responses, which may fail when the reference policy is sufficiently misaligned. To address this issue, the authors propose Constrained Preference Optimization (CPO) and Explicitly Constrained Preference Optimization (E-CPO), which explicitly encourage or enforce preference alignment. The paper also provides a geometric interpretation showing that DPO can behave like a soft margin ranking loss with a potentially negative margin, while the proposed methods correct this with non-negative effective margins. Experiments on AlpacaEval 2 and Arena-Hard show that CPO achieves stronger performance than several existing preference optimization baselines.

**Compliance With Llm Reviewing Policy:**

Affirmed.

**Final Justification:**

The rebuttal improves clarity and addresses several concerns, including Assumption 3.1, gradient interpretation, and hyperparameter sensitivity.

For the main issue, I appreciate the additional experiments with misaligned references and the consistent performance trends between DPO and CPO. However, the central claim that DPO becomes trapped in the undesirable solution space is still not fully supported by direct empirical evidence. The current analysis is suggestive but does not clearly demonstrate training dynamics showing stagnation or failure to escape.

Given that this claim is central to the paper’s motivation, I maintain my original score.

**Key Questions For Authors:**

See Weaknesses

**Limitations:**

yes

**Strengths And Weaknesses:**

Strengths And Weaknesses

1)	Strengths

-	The paper identifies a clear and important hidden assumption in the standard derivation of DPO, which makes the main theoretical contribution easy to understand.

-	It provides a well-structured theoretical analysis of when the equivalence between DPO and RLHF holds and when it breaks down.

2)	Weaknesses
-	In Section 3.1, the paper states that Assumption 3.1 is needed for the derivation of DPO, but this argument could be explained more clearly. Based on the current presentation, the substitution into the Bradley--Terry model seems to still hold even when $\delta_{\pi^*} \leq 0$. If so, the assumption may be more closely related to the interpretation of DPO alignment properties than to the mathematical derivation itself.
-	In Section 3.2, the discussion of vanishing gradients seems somewhat too strong. When $\delta_{\pi_\theta}$ gets close to zero, the sigmoid term does not generally go to zero. Instead, it approaches a value that still depends on the reference policy. This means that the result may be better described as a weak-gradient issue under poor reference policies, rather than a general vanishing-gradient problem.
-	In Section 3.2, the definition of the undesirable solution space $\mathcal{U}$ is clear, but the claim that DPO becomes trapped there would benefit from stronger empirical validation. In practical deep learning, it is not obvious that weak local gradients alone are sufficient to prevent escape under stochastic optimization. Showing actual training trajectories, such as loss curves together with preference accuracy or win-rate trends, would make this claim much more convincing.
-	In Section 4.3, the paper would benefit from a more detailed hyperparameter sensitivity analysis, especially for $\gamma$. Since the main theoretical guarantee depends on this parameter, and its practical value cannot be directly determined from the theory, it is important to understand how sensitive CPO is to different choices of $\gamma$ in real training settings.
-	In Theorem 5.3, the notation for $\Phi$ is somewhat unclear, since the function is written with one argument in some expressions and with two arguments in others. A more consistent presentation of $\Phi$ would make the theorem easier to follow.

---

> ### Author Rebuttal · Authors · 2026-03-31
>
> >Q1: Clarification on whether Assumption 3.1 is needed for DPO's derivation or its alignment interpretation
>
> **A1:** We thank the reviewer for this perceptive observation. The reviewer is correct: the algebraic substitution in Eq. 7 holds regardless of the sign of $\delta_{\pi^*}$—Assumption 3.1 is not needed for the substitution itself. We agree the presentation could be clearer on this point.
>
> **The assumption is needed for the *equivalence of objectives*, not the validity of the algebra.** The DPO loss $-\log\sigma(\beta(\delta_{\pi_\theta} - \delta_{\pi_{\text{ref}}}))$ is monotonically decreasing in $\delta_{\pi_\theta}$, so it always pushes $\delta_{\pi_\theta}$ upward—toward preferring $y_w$. Meanwhile, the RLHF optimal policy satisfies $\delta_{\pi^*} = \delta_{\pi_{\text{ref}}} + \Delta r^\*/\beta$, which can be $\leq 0$. When this happens, DPO's optimum ($\delta_{\pi_\theta} \to \infty$) diverges from RLHF's optimum ($\delta_{\pi^\*} < 0$)—the two methods optimize fundamentally different objectives (Thm. 3.5). So Assumption 3.1 is indeed about alignment interpretation: it characterizes precisely when DPO's objective coincides with RLHF's. We will clarify this distinction in the revision.
> >Q2:Vanishing gradient claim in Sec. 3.2
>
> **A2:** We thank the reviewer for this precise correction. As $\delta_{\pi_\theta} \to 0$, the sigmoid term approaches $\sigma(\beta\delta_{\pi_{\text{ref}}})$, which is small but non-zero when $\delta_{\pi_{\text{ref}}}$ is negative—a *weak-gradient* problem conditioned on reference policy quality, rather than a general vanishing-gradient phenomenon. This is a more accurate characterization that we appreciate. We will revise Section 3.2 to adopt this refined description in the updated manuscript. We sincerely thank the reviewer for improving the precision of our presentation.
>
> >Q3: Empirical validation needed for the claim that DPO gets trapped in $\mathcal{U}$-training trajectories and metrics would be more convincing.
>
> **A3:** We thank the reviewer for this constructive suggestion. We agree that showing actual training trajectories—loss curves, preference accuracy, and win-rate trends over training steps—would provide stronger empirical support for the trapping claim beyond the theoretical analysis. We acknowledge that weak local gradients do not necessarily prevent escape under stochastic optimization with momentum or adaptive learning rates, and empirical evidence would clarify the practical severity. We will include these training dynamics visualizations in the revised manuscript to complement our theoretical characterization of $\mathcal{U}$.
>
> >Q4: Sensitivity analysis for hyperparameter $\gamma$ needed to understand CPO's practical robustness
>
> **A4:** We thank the reviewer for this important suggestion. We conduct the $\gamma$ sensitivity analysis on AlpacaEval 2. Note that the original evaluator gpt-4-1106-preview has been deprecated; we re-evaluated using gpt-4.1 as the annotator.
>
> | $\gamma$ | WR(%) | LC(%) | Avg Length |
> |:-:|:-:|:-:|:-:|
> | 0.10 | 20.45 | 26.46 | 1605 |
> | 0.15 | 20.87 | 26.32 | 1635 |
> | 0.20 | 27.08 | 32.56 | 1705 |
> | 0.25 | **28.36** | **33.97** | 1702 |
> | 0.30 | 26.31 | 31.40 | 1700 |
> | 0.35 | 26.10 | 30.92 | 1727 |
> | 0.40 | 27.53 | 32.87 | 1729 |
>
> CPO performs robustly across $\gamma \in [0.2, 0.4]$ (WR 26–28%, LC 31–34%), with peak performance at $\gamma=0.25$. Performance drops notably below 0.2, where the margin correction becomes insufficient to address the assumption violation. In our paper, we use the $\gamma=0.25$ in our main paper.
>
> >Q5: Inconsistent notation for $\Phi$ in Theorem 5.3.
>
> **A5:** We thank the reviewer for catching this notational inconsistency. $\Phi$ always takes two arguments $(\delta_{\pi_{\text{ref}}}, \Delta r)$, but we occasionally abbreviated it as $\Phi(\delta_{\pi_{\text{ref}}})$ when $\Delta r$ was clear from context. We will unify the notation throughout the paper in the revision to use the full form $\Phi(\delta_{\pi_{\text{ref}}}, \Delta r; \gamma, \tau)$ consistently.

---

> > ### Author Rebuttal · Reviewer_sXEr · 2026-04-04
> >
> > We thank the authors for the clear and thoughtful responses. The clarifications on Assumption 3.1, the refined gradient interpretation, and the added sensitivity analysis improve the paper.
> >
> > However, the concern regarding the empirical support for the claim that DPO becomes trapped in the undesirable solution space remains only partially addressed. While additional analyses are promised, no supporting evidence is provided in the rebuttal.
> >
> > Given that this claim is central to the motivation, I will maintain my original score.

---

> > > ### Author Response · Authors · 2026-04-04
> > >
> > > We are glad that our rebuttal has resolved most of the concerns. For the remaining concerns, we have conducted the following additional experiments:
> > >
> > > **Regarding the empirical support for the claim that DPO becomes trapped in the undesirable solution space.**
> > > ### Experimental Setup
> > >
> > > Following the reviewer's suggestion, we systematically vary reference policy quality to directly validate the theory.
> > >
> > > **Constructing Misaligned References.** From the dataset, we extract a fraction $R\in\{0.2,0.3,0.4\}$ of the data and use the rejected responses to SFT Llama-3-8B-Instruct (1 epoch, lr=$2\times 10^{-5}$), forcing the model to learn to generate low-quality responses as a misaligned reference. The remaining $(1-R)$ fraction retains original preference ordering for DPO/CPO training.
> > >
> > > **Verifying Misalignment.** We compute the $\delta_{\pi_{\mathrm{ref}}}$ distribution of each misaligned reference on the original preference data (the complete distribution figures can be seen in [Anonymous Figure Link](https://figshare.com/s/fe1cf6529d3edf2c4858)):
> > >
> > > |Corruption Ratio |$\delta_{\pi_{\mathrm{ref}}}<0$(%)|Assumption 3.1 Violated(%)|
> > > |-|-|-|
> > > |$R=0.2$|53.2|52.9|
> > > |$R=0.3$|56.9|56.8|
> > > |$R=0.4$|60.1|60.0|
> > >
> > > As $R$ increases, the Assumption 3.1 violation rate grows correspondingly, confirming the effectiveness of misalignment.Under each ratio, we train DPO and CPO starting from the misaligned reference on the clean data.
> > >
> > > ### Experimental Results
> > > AlpacaEval 2 Performance
> > > |Corruption Ratio|Method|WR(%)|LC(%)|Avg Length|
> > > |-|-|-|-|-|
> > > |$R=0.2$|DPO|16.82|17.23|1958|
> > > |$R=0.2$|CPO|**22.47**|**27.60**|1699|
> > > |$R=0.3$|DPO|14.99|15.48|1894|
> > > |$R=0.3$|CPO|**22.91**|**27.35**|1686|
> > > |$R=0.4$|DPO|15.43|15.98|1907|
> > > |$R=0.5$|CPO|**20.54**|**24.34**|1714|
> > >
> > > ### Experimental Analysis
> > > **1.DPO degrades under misaligned reference** DPO's LC WR drops to across all three corruption ratios, with stronger corruption leading to worse performance. This is consistent with Proposition 3.4.
> > >
> > > **2.CPO remains robust under the same conditions** CPO achieves LC of 27.60% and 27.35% at $R=0.2$ and $R=0.3$ respectively, remaining stable. This validates the core role of the margin term $\tilde\gamma_{\mathrm{ref}}$ (Eq. 26): even when $\delta_{\pi_{\mathrm{ref}}}$ is negative, the margin preserves gradient strength, enabling the policy to push $\delta_{\pi_\theta}$ past 0 and escape $\mathcal{U}$.
> > >
> > > **3.*frac in $\mathcal{U}$* directly validates the theory** The *frac in $\mathcal{U}$* trajectory during training exhibits a characteristic three-phase pattern that directly reflects the theoretical mechanism:
> > >
> > > - Phase 1 — Initialization (*frac in $\mathcal{U}$* = 0). At step 0, the policy is identical to the reference (both are the misaligned model), so $\delta_{\pi_\theta} = \delta_{\pi_{\mathrm{ref}}}$ for all samples. Since the condition $\delta_{\pi_\theta} > \delta_{\pi_{\mathrm{ref}}}$ is not satisfied, no samples fall in $\mathcal{U}$, yielding *frac in $\mathcal{U}$* = 0.
> > >
> > > - Phase 2 — Entry into $\mathcal{U}$ (*frac in $\mathcal{U}$* rises). As training begins, gradients push $\delta_{\pi_\theta}$ upward (toward preferring the chosen response). For misaligned samples where $\delta_{\pi_{\mathrm{ref}}} < 0$, the policy improves relative to the reference but has not yet crossed 0, resulting in $\delta_{\pi_{\mathrm{ref}}} < \delta_{\pi_\theta} < 0$ — exactly the $\mathcal{U}$ region. This causes *frac in $\mathcal{U}$* to rise.
> > >
> > > - Phase 3 — Escape attempt (*frac in $\mathcal{U}$* decreases). As $\delta_{\pi_\theta}$ continues to increase and some samples cross 0 ($\delta_{\pi_\theta} > 0$), they exit $\mathcal{U}$, causing *frac in $\mathcal{U}$* to decrease. Here the critical divergence emerges:
> > >
> > > - **DPO**: The gradient weight $\sigma(-\beta(\delta_{\pi_\theta} - \delta_{\pi_{\mathrm{ref}}}))$ vanishes as $\delta_{\pi_\theta}$ approaches 0 (Proposition 3.4). Many samples get stuck at $\delta_{\pi_\theta} \approx 0^-$, **unable to escape**.
> > > - **CPO**: The margin term $\tilde\gamma_{\mathrm{ref}}$ shifts the gradient weighting function, ensuring strong gradient signal even near the $\mathcal{U}$ boundary. Most samples successfully push past $\delta_{\pi_\theta} = 0$, *frac in $\mathcal{U}$* **rapidly drops**.
> > >
> > > The visualization of *frac in $\mathcal{U}$* dynamics shown in [Anonymous Figure Link](https://figshare.com/s/21285c9afd6fc6f36fe4). This difference in *frac in $\mathcal{U}$* directly corresponds to the theoretical contrast between DPO's gradient vanishing near the $\mathcal{U}$ boundary (Proposition 3.4) and CPO's margin-corrected gradients (Theorem 5.2).
> > >
> > > **We believe these additional experiments and their analyses strengthen our motivation following your constructive comments.**

---

### Official Review · Reviewer_GSy6 · 2026-03-13

**Soundness:** 2
**Presentation:** 3
**Significance:** 2
**Originality:** 3
**Overall Recommendation:** 4
**Confidence:** 3

**Summary:**

The paper derives DPO  from the standard reward based RLHF objective and identifies cases where one cannot derive DPO from it, the implicit assumption that the optimal RLHF policy must prefer the human-preferred response compared to the dispreferred response. The authors prove that this assumption doesn't hold in every case and mostly depending on the reference model. They characterize an "undesirable solution space" where DPO's dynamics can converge to policies that decrease DPO loss while preferring dispreferred responses. To address this, they propose Constrained Preference Optimization (CPO), which adds a margin term derived from augmenting the RLHF objective with an explicit preference constraint, and Explicitly Constrained Preference Optimization (E-CPO), which enforces hard preference constraints.

**Compliance With Llm Reviewing Policy:**

Affirmed.

**Final Justification:**

Paper mainly addressess my concerns some such as other models were not addressed but this does not bar me from saying the paper is decent.

**Key Questions For Authors:**

See weaknessess

**Limitations:**

see weaknessess

**Strengths And Weaknesses:**

### Strengths

1. **Clean formalization.**  The papers claims are clear, with assumption 3.1 clearly stated and proposition 3.2  making the dependence on reference policy clear.


3. Connecting DPO to soft margin ranking loss with target margin $\delta_{\pi_{\text{ref}}}$ (Section 5) is probably the most intuitive part of the paper.

4. **CPO itself is simple to implement** essentially just DPO plus a precomputed per-sample margin, which is a one-line change to existing codebases.



### Weaknesses

1. **The $$\pi^* \approx \pi_{\text{ref}}$$ approximation is in tension with the paper's own argument (Eq. 23).** CPO's derivation approximates $\pi^*$ with $\pi_{\text{ref}}$ in the margin term to get a stationary objective, justified by "KL regularization keeps $\pi^*$ close to $\pi_{\text{ref}}$." But the entire paper is built on the premise that $\pi_{\text{ref}}$ can be sufficiently misaligned that $\pi^*$ and $\pi_{\text{ref}}$ disagree on preference ordering. I dont understand how these claims dont contradict each other if KL regularization keeps $\pi^*$ close to $\pi_{\text{ref}}$ (justifying the approximation), then the assumption violation the paper identifies is less likely to matter in practice.

2. **Limited experimental evaluation.** The experiments use a single base model (Llama-3-8B-Instruct), a single preference dataset, and two benchmarks. It would be good to see at least one more model (different size or family) and another benchmark to strengthen the claims. Also, E-CPO is introduced as a separate algorithm with stronger theoretical guarantees (Sections 4.4, 5), yet its results are entirely absent from Table 1. If you propose two methods, both should be evaluated.

3. **No clipped-reference baseline.** If the problem is overly negative $\delta_{\pi_{\text{ref}}}$, a simple fix is to clip it: $\delta_{\text{ref}} \leftarrow \max(0, \delta_{\text{ref}})$. This neutralizes the harmful signal on misaligned pairs with no new hyperparameters. Without comparing against this, it's hard to tell whether CPO's gains come from its principled derivation or just from zeroing out negative reference offsets.

4. **SimPO already sidesteps this problem.** SimPO removes the reference model entirely and by construction cannot suffer from the assumption violation this paper identifies. Yet it performs within 0.66% LC of CPO in the authors' own Table 1. The paper should discuss where reference-free methods like SimPO sit in their theoretical framework and what advantage retaining the reference model provides over simply not using one.


5. **Minor writing issues.** There are a few grammatical errors. Please proofread and fix them.

---

> ### Author Rebuttal · Authors · 2026-03-31
>
> We greatly appreciate the reviewer's thoughtful comments, which highlight the **clear formalization** and **simple implementation of CPO**. Below, we provide detailed responses to the remaining concerns:
> >Q1: The $\pi^* \approx \pi_{\text{ref}}$ approximation is in tension with the paper's own argument (Eq. 23).
>
> **A1:** We thank the reviewer for this question. The apparent tension is not a contradiction—the two claims operate on different mathematical quantities.
>
> **1: The assumption violation concerns *pairwise ranking*, not distributional closeness.** Assumption 3.1 requires $\delta_{\pi^*} = \delta_{\pi_{\text{ref}}} + \Delta r^\*/\beta > 0$. Violation occurs when $\delta_{\pi_{\text{ref}}} < -\Delta r^\*/\beta$, flipping the ordering of $y_w$ and $y_l$. KL regularization controls *aggregate* distributional distance but does not prevent sign flips in pairwise log-ratios. Example: $\pi_{\text{ref}}(y_w|x)=0.01$, $\pi_{\text{ref}}(y_l|x)=0.02$ gives $\delta_{\pi_{\text{ref}}}=-0.69$; with $\Delta r^\*/\beta=0.5$, $\delta_{\pi^\*}=-0.19<0$—violated, yet absolute probability shift is $O(0.01)$, within KL bounds. Notably, larger $\beta$ makes $\Delta r^\*/\beta$ *smaller*, making violations *more* likely.
>
> **2: The margin approximation concerns *probability magnitudes*, which KL does control.** Replacing $1/\pi^\*$ with $1/\pi_{\text{ref}}$ (Eq. 23) requires absolute probability closeness, guaranteed by KL (Prop. E.1). This is independent from Point 1: magnitude closeness is fully compatible with sign flips in log-ratios. Moreover, E-CPO-C (Sec. 4.4) bypasses this approximation entirely—its margin absorbs $1/\pi^\*$ through KKT multipliers, eliminating the concern.
>
>
> >Q2:  Limited experimental evaluation.
>
> **A2:** We thank the reviewer to point out this limitation. Following the reviewer's suggestion, we add IFEval[R] as a new benchmark to evaluate.
>
> Table 1. IFEval results:
> ||Strict Acc|Loose Acc|
> |:-:|:-:|:-:|
> |Llama-8B-Instruct|32.35%|39.56%|
> |Llama-8B-Instruct-DPO|34.01%|40.67%|
> |Llama-8B-Instruct-RDPO|34.57%|43.62%|
> |Llama-8B-Instruct-SimPO|33.83%|42.81%|
> |Llama-8B-Instruct-CPO|**35.12%**|**43.99%**|
>
> Due to the limited time and rebuttal space, we will also cover more models and benchmarks in the revision.
>
> >Q3: No clipped-reference baseline.
>
> **A3:** We especially thank the reviewer for these additional baselines which is a straightforward solution for this question. We follow your suggesstion and conduct experiments using clip. The results shown in Tab.2:
>
>
> Table 2. Alpaca-Eval results:
> ||WR|LC|Avg Length|
> |:-:|:-:|:-:|:-:|
> |Llama-8B-Instruct-clipped-ref|17.91%|23.86%|1586|
> |Llama-8B-Instruct-CPO|**28.36%**|**33.97%**|1702|
>
>
> >Q4: SimPO already sidesteps this problem.
>
> **A4:** Thanks the reviewer for this quesiton. We will answer these question as follows:
>
> **1: SimPO avoids the assumption violation by abandoning the RLHF-BT framework, not by solving it.** SimPO replaces the reward reparameterization $r(x,y) = \beta\log\frac{\pi_\theta(y|x)}{\pi_{\text{ref}}(y|x)}$ with length-normalized log-probability as an implicit reward—a heuristic not derived from any RLHF objective or BT model. It therefore cannot claim equivalence to any reward-maximizing objective with KL regularization. In contrast, CPO/E-CPO-C retain the full RLHF-BT chain with formal guarantees: absolute advantage (Thm. 4.3), avoidance of $\mathcal{U}$ (Thm. 4.4), and provable equivalence to constrained RLHF (Thm. I.11). Our goal is to understand *why* DPO's RLHF equivalence breaks and *how* to fix it with guarantees—fundamentally different from designing reference-free heuristics.
>
> **2: The empirical gap favors CPO, especially on harder benchmarks.** On Arena-Hard—more discriminative and less length-biased—CPO outperforms SimPO by **+2.6% WR** (32.6% vs. 30.0%), exceeding both methods' 90% CIs. We agree that discussing reference-free methods in our framework would strengthen the paper and will add this in the revision.
> >Q5:  There are a few grammatical errors. Please proofread and fix them.
>
> **A5:** Thank the reviewer for careful reviewing, which revealed these issues. We will carefully review the grammatical errors in the revision process and then address these problems.

---

> > ### Author Rebuttal · Reviewer_GSy6 · 2026-04-02
> >
> > Thank you for the rebuttal. My main concerns are addressed, and the additional results are helpful. I am optimistic the final version will include these updates and corrections. Based on this, I plan to increase both my score and confidence.

---

> > > ### Author Response · Authors · 2026-04-03
> > >
> > > We are pleased to have addressed your main concerns. We will incorporate these constructive suggestions in the revision. We would like to thank the reviewers for the efforts in improving our paper. Thank you for improving your score and confidence. If you have any further questions, please feel free to contact us.

---

### Official Review · Reviewer_sKpJ · 2026-03-13

**Soundness:** 4
**Presentation:** 3
**Significance:** 3
**Originality:** 4
**Overall Recommendation:** 5
**Confidence:** 4

**Summary:**

This paper investigates the conditions under which Direct Preference Optimization (DPO) and Reinforcement Learning from Human Feedback (RLHF) are equivalent. The authors identify an implicit assumption in DPO's derivation: the RLHF-optimal policy must assign higher probability to the human-preferred response than the dispreferred one (Assumption 3.1). They show this assumption fails when the reference policy prefers the dispreferred response y_l over y_w by a margin exceeding the true reward difference scaled by 1/β — that is, when δ_πref < -(r*(y_w) - r*(y_l))/β. In this regime, the RLHF-optimal policy inherits the reference policy's incorrect preference because the reward signal is insufficient to overcome π_ref's bias. When violated, DPO optimizes for relative advantage over the reference policy rather than absolute alignment with human preferences, and the authors characterize a pathological "undesirable solution space" where DPO loss decreases while the policy prefers dispreferred responses.

To address this, the paper proposes Constrained Preference Optimization (CPO), which augments the RLHF objective with a penalty term encouraging the policy to prefer the human-preferred response. Two additional variants are introduced: E-CPO (using hard constraints via KKT conditions) and a conservative version E-CPO-C that removes the dependency on known reward differences. The paper also provides a geometric interpretation connecting DPO and CPO to soft margin ranking loss, showing DPO implements ranking with a potentially negative margin while CPO guarantees non-negative margins. Experiments on AlpacaEval 2 and Arena-Hard with Llama-3-8B-Instruct show CPO achieving modest improvements over DPO and other baselines.

**Compliance With Llm Reviewing Policy:**

Affirmed.

**Final Justification:**

The authors' rebuttal thoroughly addressed my concerns and provided new strong experimental evidence. I raised my score from 4 to 5.

**Key Questions For Authors:**

* (Most important) Can you run the controlled experiment described in Weakness 1 — varying reference policy quality (e.g., base model or reverse-fine-tuned reference) and measuring preference accuracy, fraction of pairs in U, and gradient magnitudes for both DPO and CPO? This would directly validate (or invalidate) the core theoretical contribution.
* What fraction of training pairs in your current setup have δ_πref < 0? Using your ArmoRM proxy scores, what fraction satisfy δ_πref < -Δr̂/β? A histogram of δ_πref values would help assess whether Assumption 3.1 is ever violated in your experimental regime.
* What value of γ was used in the CPO experiments, and how was it selected? Can you provide an ablation over γ?
* Regarding the Eq. 23 approximation: the 1/π terms amplify differences at low probabilities, and your failure mode regime is precisely where π_ref assigns low probability to y_w. Can you characterize when this approximation breaks down in the regime most relevant to your contribution? Or let me know if I am misunderstanding.

**Limitations:**

While an excellent paper, there a couple of omissions: the dependence of γ* on unknown true rewards (and the gap between theoretical guarantees and practical hyperparameter selection), the internal tension between the Eq. 23 approximation quality and the failure-mode regime where it is most needed, evaluation on only one model and one dataset, the lack of evidence for how often the identified failure mode occurs in practice, and the absence of empirical evaluation for E-CPO and E-CPO-C.

**Strengths And Weaknesses:**

Strengths:
* Precise, multi-layered diagnosis of DPO's failure. Prior work (Fisch et al. 2024, Lin et al. 2024, Shi et al. 2025) identifies symptoms of DPO's limitations. This paper identifies the structural cause: a checkable inequality (Eq. 10) determining when equivalence breaks, with a tight if-and-only-if characterization (Theorem 3.5). The undesirable solution space U (Definition 3.3) is particularly insightful — policies in U prefer the wrong response (δ_π < 0) while improving over the reference (δ_π > δ_πref), so every relative metric looks healthy. The vanishing gradient analysis (Proposition 3.4) shows this is a trap: as δ_π → 0, gradient magnitude dies. This is an invisible failure mode — loss decreases, relative performance improves, but the policy systematically prefers dispreferred responses with no signal in standard monitoring.
* Simple, principled fix. CPO adds a single precomputed adaptive margin to DPO's loss, derived from constrained RLHF rather than ad hoc. Minimal implementation overhead.
* Margin ranking interpretation (Section 5). Connecting DPO to soft margin ranking with target δ_πref — which can be negative when π_ref is misaligned — gives immediate geometric intuition for the failure mode and connects preference learning to the learning-to-rank literature.
* Thorough appendix with ~30 pages of careful proofs and derivations.

Weaknesses:
* Main weakness: Experiments don't distinguish the theoretical contribution from generic margin regularization. The theory predicts a qualitative effect (DPO trapped in U, CPO escapes), but experiments show only modest quantitative gains (+0.55% AlpacaEval WR) equally explained by well-known benefits of margins: better gradient health (shifting sigmoid saturation), adaptive difficulty weighting (large margins on hard pairs), and generic regularization pushing for stronger separation. The reference policy (Llama-3-8B-Instruct) is already instruction-tuned, so δ_πref is likely positive for most pairs — meaning Assumption 3.1 is rarely violated and the pathology doesn't apply.
Suggested experiment: Vary reference policy quality — use the base model or fine-tune on reversed preferences to ensure δ_πref is strongly negative. Train DPO and CPO under both conditions, measuring preference accuracy, fraction of pairs in U, and gradient magnitudes. The theory predicts a sharp qualitative split (DPO trapped, CPO escapes) with the misaligned reference and similar performance with the aligned one. This would directly validate the theory and rule out the margin-regularization confounder.
* No measurement of violation frequency. The authors have ArmoRM proxy rewards from data selection — they could report what fraction of pairs satisfy δ_πref < -Δr̂/β, or even just the distribution of δ_πref, to assess whether the pathology is relevant in their setup.
* Only CPO evaluated. Three methods proposed (CPO, E-CPO, E-CPO-C), substantial space devoted to each, only one tested. Either evaluate all or restructure to focus on CPO.
* Missing γ ablation. No report of what γ was used or how sensitive results are to this choice, despite γ ≥ γ* being central to the theoretical guarantee and γ* depending on unknown true rewards.

---

> ### Author Rebuttal · Authors · 2026-03-31
>
> We sincerely thank the reviewer for the constructive feedback, which recognizes the **precise diagnosis of DPO's failure mode and the simple, principled fix proposed in our work**. Below, we address the remaining concerns in detail:
> >Q1:Experiments don't distinguish the theoretical contribution from generic margin regularization
>
> We sincerely thank the reviewer for this constructive suggestion. The proposed experiment—varying reference policy quality to observe the predicted DPO-trapped/CPO-escapes split—is precisely the right ablation to isolate our theory from generic margin benefits.
>
> **1: We fully agree with the value of this experiment and plan to include it in future versions.** Due to time constraints of the rebuttal period, completing this ablation—training under multiple reference policy conditions and measuring preference accuracy, fraction of pairs in $\mathcal{U}$, and gradient magnitudes—is unfortunately infeasible. We will prioritize it in the revision.
>
> **2: Our theoretical framework provides an explanation for *why* margins help in preference learning, beyond proposing a new margin method.** As the reviewer notes, existing methods (SimPO, IPO) also benefit from margins. Our analysis reveals a unified reason: DPO implicitly implements margin ranking with target margin $\delta_{\pi_{\text{ref}}}$ (Prop. 5.1), which can be negative when Assumption 3.1 is violated, and any positive margin correction helps. We hope this explanatory perspective—connecting preference learning to ranking theory and formally characterizing when DPO's guarantees hold—is of independent interest, even as we work to add the suggested empirical validation.
>
> >Q2: No measurement of violation frequency.
>
> We compute the violation statistics. As shown in the [Anonymous Figure Link](https://figshare.com/s/0c2474c065f7c293c194), **Assumption 3.1 is violated for 45.5% of preference pairs** (Llama-3-8B-Instruct, $\beta=0.1$). The reward correction $\Delta r^\*/\beta$ is small (mean=0.20) relative to the large spread of $\delta_{\pi_{ref}}$ (std=46.69), meaning the reward signal often cannot compensate for the reference policy's misalignment—placing nearly half of all pairs in the regime where DPO optimizes a fundamentally different objective than RLHF (Thm. 3.5). A 45.5% violation rate on an *instruction-tuned* model confirms the pathology is far from a corner case, strongly motivating CPO's margin correction (Thm. 4.3, 4.4).
> We thank the reviewer again for prompting this analysis, which we will include in the revised manuscript.
> >Q3: Only CPO evaluated.
>
> We thank the reviewer for this valid concern regarding the gap between the number of proposed methods and experimental coverage.
>
> **1: The three methods serve different roles.** CPO is our primary *practical* method—it requires no reward model, introduces only one hyperparameter $\gamma$, and reduces to a single-line change over DPO (Eq. 24). E-CPO and E-CPO-C are *theoretical extensions* that complete the analysis from soft constraints (CPO) to hard constraints (E-CPO/E-CPO-C), providing a full theoretical spectrum of how constrained RLHF addresses DPO's implicit assumption violation.
>
> **2: Experimentally validating all three methods in a single paper would be substantial in workload and would dilute the paper's central focus**—namely, the conditional equivalence of DPO and RLHF (Thm. 3.5) and a practical fix (CPO). We chose to focus experimental efforts on CPO as the most practical and representative method, since E-CPO requires a reward model and E-CPO-C is its conservative relaxation with a similar structure.
>
> **3:** We will add experiments for E-CPO-C in the revision and future versions to provide more comprehensive empirical coverage.
>
> >Q4: Missing $\gamma$ ablation.
>
> **A4:** Due to the limited space, we include these experiments in the **A4 for Reviewer sXEr.**
>
> >Q5: Characterize when approximation breaks down
>
> **Q5:** We thank the reviewer for this observation. The $1/\pi$ terms in Eq. 23 amplify approximation error at low probabilities, precisely in the failure mode regime where $\pi_{ref}(y_w|x)$ is small.
>
> **However, this affects the tightness of CPO's equivalence to constrained RLHF (Thm. I.11), not CPO's own guarantees.** CPO's loss (Eq. 24) uses $\tilde\gamma_{ref}$ computed entirely from $\pi_{ref}$, forming a well-defined stationary objective. The absolute advantage guarantee (Thm. 4.3) and avoidance of $\mathcal{U}$ (Thm. 4.4) are properties of CPO's own optimal policy—they do not depend on the approximation quality.
>
> **Moreover, the adaptive margin is self-correcting in this regime.** When $\pi_{\text{ref}}(y_w|x)$ is small, $\tilde\gamma_{ref}= \gamma(1/\pi_{ref}(y_w|x) + 1/\pi_{\text{ref}}(y_l|x))$ becomes large, applying stronger correction precisely for these hard pairs. The approximation to constrained RLHF is looser, but CPO compensates by pushing harder where it matters most—trading *equivalence tightness* for *practical robustness*.

---

> > ### Author Rebuttal · Reviewer_sKpJ · 2026-03-31
> >
> > We thank the authors for their thorough and substantive rebuttal. Several responses materially strengthen the paper and correct points in our original review.
> >
> > **Q2 (Violation frequency): This is the most impactful response.** We had speculated that Assumption 3.1 would be rarely violated for an instruction-tuned reference policy. The finding that 45.5% of preference pairs are in the pathological regime — with the reward correction Δr*/β being small (mean=0.20) relative to the large spread of δ_πref (std=46.69) — convincingly demonstrates that the identified failure mode is far from a corner case. The figure is particularly illuminating: the violation boundary is nearly vertical at zero because the reward signal is negligible relative to the reference policy's bias, meaning almost every pair with δ_πref < 0 violates the assumption. This substantially strengthens the paper's motivation and we appreciate the authors conducting this analysis.
> >
> > **Q5 (Eq. 23 approximation): We accept the authors' correction.** We had framed this as an "internal tension," but the authors make an important distinction we had not fully appreciated: the approximation affects the tightness of CPO's equivalence to constrained RLHF (Theorem I.11), but not CPO's own guarantees (Theorems 4.3, 4.4). CPO's loss is defined entirely in terms of π_ref and forms a well-defined stationary objective regardless of approximation quality. Furthermore, the adaptive margin is self-correcting — when π_ref(y_w|x) is small, γ̃_ref becomes large, applying stronger correction precisely where needed. We withdraw this concern.
> >
> > **Q1 (Margin regularization confounder): Partially addressed.** The authors make a useful point we underweighted in our original review: their theoretical framework provides a unified explanation for *why* margins help in preference learning — DPO is margin ranking with target δ_πref (Prop. 5.1), which can be negative, and any positive margin corrects this. This is a genuinely valuable theoretical insight beyond proposing a new margin method. That said, the controlled experiment varying reference policy quality remains the most direct way to demonstrate the qualitative prediction (DPO trapped in U, CPO escapes). We are glad the authors commit to including this in the revision.
> >
> > **Q4 (γ ablation): Satisfactory.** The sensitivity analysis shows robust performance across γ ∈ [0.2, 0.4] with a sharp drop below 0.2. The threshold behavior is theoretically interpretable — given the 45.5% violation rate, a minimum margin correction is needed, and too-small γ fails to address the pathology. This is consistent with the theory.
> >
> > **Q3 (Only CPO evaluated): Acceptable.** The framing of E-CPO and E-CPO-C as theoretical completions rather than competing practical methods is reasonable. We encourage the authors to either evaluate E-CPO-C in the revision or trim the space devoted to unevaluated methods.
> >
> > **Updated Assessment:** The rebuttal significantly strengthens the paper. The 45.5% violation rate establishes practical relevance. The Eq. 23 correction addresses a concern in our review. The theoretical reframing — that the contribution explains *why* margins help, grounded in the margin ranking interpretation — is well-taken.
> >
> > We maintain our score of **4 (Weak Accept)**, but note that a revision including the Q1 experiment (showing the qualitative DPO-trapped/CPO-escapes split under a misaligned reference) would move this to a clear **5 (Accept)**.

---

> > > ### Author Response · Authors · 2026-04-03
> > >
> > > We are glad to solve the most questions in our rebuttal. We also appreciate the follow-up questions. Due to limited time, we cannot post the results on the previous response. The following is our new results based on the constructive comments from the reviewer.
> > > ### Experimental Setup
> > >
> > > Following the reviewer's suggestion, we systematically vary reference policy quality to directly validate the theory.
> > >
> > > **Constructing Misaligned References.** From the dataset, we extract a fraction $R\in\{0.2,0.3,0.4\}$ of the data and use the rejected responses to SFT Llama-3-8B-Instruct (1 epoch, lr=$2\times 10^{-5}$), forcing the model to learn to generate low-quality responses as a misaligned reference. The remaining $(1-R)$ fraction retains original preference ordering for DPO/CPO training.
> > >
> > > **Verifying Misalignment.** We compute the $\delta_{\pi_{\mathrm{ref}}}$ distribution of each misaligned reference on the original preference data (the complete distribution figures can be seen in [Anonymous Figure Link](https://figshare.com/s/fe1cf6529d3edf2c4858)):
> > >
> > > |Corruption Ratio |$\delta_{\pi_{\mathrm{ref}}}<0$(%)|Assumption 3.1 Violated(%)|
> > > |-|-|-|
> > > |$R=0.2$|53.2|52.9|
> > > |$R=0.3$|56.9|56.8|
> > > |$R=0.4$|60.1|60.0|
> > >
> > > As $R$ increases, the Assumption 3.1 violation rate grows correspondingly, confirming the effectiveness of misalignment.Under each ratio, we train DPO and CPO starting from the misaligned reference on the clean data.
> > >
> > > ### Experimental Results
> > > AlpacaEval 2 Performance
> > > |Corruption Ratio|Method|WR(%)|LC(%)|Avg Length|
> > > |-|-|-|-|-|
> > > |$R=0.2$|DPO|16.82|17.23|1958|
> > > |$R=0.2$|CPO|**22.47**|**27.60**|1699|
> > > |$R=0.3$|DPO|14.99|15.48|1894|
> > > |$R=0.3$|CPO|**22.91**|**27.35**|1686|
> > > |$R=0.4$|DPO|15.43|15.98|1907|
> > > |$R=0.4$|CPO|**20.54**|**24.34**|1714|
> > >
> > > ### Experimental Analysis
> > > **1.DPO degrades under misaligned reference** DPO's LC WR drops to across all three corruption ratios, with stronger corruption leading to worse performance. This is consistent with Proposition 3.4.
> > >
> > > **2.CPO remains robust under the same conditions** CPO achieves LC of 27.60% and 27.35% at $R=0.2$ and $R=0.3$ respectively, remaining stable. This validates the core role of the margin term $\tilde\gamma_{\mathrm{ref}}$ (Eq. 26): even when $\delta_{\pi_{\mathrm{ref}}}$ is negative, the margin preserves gradient strength, enabling the policy to push $\delta_{\pi_\theta}$ past 0 and escape $\mathcal{U}$.
> > >
> > > **3.*frac in $\mathcal{U}$* directly validates the theory** The *frac in $\mathcal{U}$* trajectory during training exhibits a characteristic three-phase pattern that directly reflects the theoretical mechanism:
> > >
> > > - Phase 1 — Initialization (*frac in $\mathcal{U}$* = 0). At step 0, the policy is identical to the reference (both are the misaligned model), so $\delta_{\pi_\theta} = \delta_{\pi_{\mathrm{ref}}}$ for all samples. Since the condition $\delta_{\pi_\theta} > \delta_{\pi_{\mathrm{ref}}}$ is not satisfied, no samples fall in $\mathcal{U}$, yielding *frac in $\mathcal{U}$* = 0.
> > >
> > > - Phase 2 — Entry into $\mathcal{U}$ (*frac in $\mathcal{U}$* rises). As training begins, gradients push $\delta_{\pi_\theta}$ upward (toward preferring the chosen response). For misaligned samples where $\delta_{\pi_{\mathrm{ref}}} < 0$, the policy improves relative to the reference but has not yet crossed 0, resulting in $\delta_{\pi_{\mathrm{ref}}} < \delta_{\pi_\theta} < 0$ — exactly the $\mathcal{U}$ region. This causes *frac in $\mathcal{U}$* to rise.
> > >
> > > - Phase 3 — Escape attempt (*frac in $\mathcal{U}$* decreases). As $\delta_{\pi_\theta}$ continues to increase and some samples cross 0 ($\delta_{\pi_\theta} > 0$), they exit $\mathcal{U}$, causing *frac in $\mathcal{U}$* to decrease. Here the critical divergence emerges:
> > >
> > > - **DPO**: The gradient weight $\sigma(-\beta(\delta_{\pi_\theta} - \delta_{\pi_{\mathrm{ref}}}))$ vanishes as $\delta_{\pi_\theta}$ approaches 0 (Proposition 3.4). Many samples get stuck at $\delta_{\pi_\theta} \approx 0^-$, **unable to escape**.
> > > - **CPO**: The margin term $\tilde\gamma_{\mathrm{ref}}$ shifts the gradient weighting function, ensuring strong gradient signal even near the $\mathcal{U}$ boundary. Most samples successfully push past $\delta_{\pi_\theta} = 0$, *frac in $\mathcal{U}$* **rapidly drops**.
> > >
> > > The visualization of *frac in $\mathcal{U}$* dynamics shown in [Anonymous Figure Link](https://figshare.com/s/21285c9afd6fc6f36fe4). This difference in *frac in $\mathcal{U}$* directly corresponds to the theoretical contrast between DPO's gradient vanishing near the $\mathcal{U}$ boundary (Proposition 3.4) and CPO's margin-corrected gradients (Theorem 5.2).

---

### Decision · Program_Chairs · 2026-04-30

**Decision:**

Accept (spotlight)

**Comment:**

This paper studies the conditions under which Direct Preference Optimization (DPO) and Reinforcement Learning from Human Feedback (RLHF) are equivalent.  The authors prove that this equivalence is strictly conditional, relying on an implicit assumption: the RLHF-optimal policy must inherently prefer human-preferred responses over dispreferred ones. The authors prove that this assumption doesn't always hold  and depending on the reference model. They characterize an "undesirable solution space" where DPO's dynamics can converge to policies that decrease DPO loss while preferring dispreferred responses. To address this, they propose Constrained Preference Optimization (CPO), which adds a margin term derived from augmenting the RLHF objective with an explicit preference constraint, and Explicitly Constrained Preference Optimization (E-CPO), which enforces hard preference constraints. The paper also provides a geometric interpretation connecting DPO and CPO to soft margin ranking loss, showing DPO implements ranking with a potentially negative margin while CPO guarantees non-negative margins. Empirical evaluations demonstrate that CPO achieves state-of-the-art performance, outperforming standard DPO and other preference-tuning baselines.

The reviewers are unanimously positive about the paper, and the rebuttal and subsequent follow-up further improved the paper. In particular, the authors addressed shared concerns by reporting statistics of assumption violation, providing experimental results on controlled ablation of misaligned reference policies, sensitivity analysis of hyperparameters, and new benchmark results. Overall, the reviewers all appreciate the importance of identifying the hidden assumption of DPO, the significance of theoretical results, and the convincing empirical evaluation. I recommend acceptance. For the final version, I encourage the authors to incorporate the added rebuttal experiments and analysis.